# Entropy-Based Aggregation for Fair and Effective Federated Learning

## Abstract

Federated Learning (FL) enables collaborative model training across distributed devices while preserving data privacy. Nonetheless, the heterogeneity of edge devices often leads to inconsistent performance of the globally trained models, resulting in unfair outcomes among users. Existing federated fairness algorithms strive to enhance fairness but often fall short in maintaining the overall performance of the global model, typically measured by the average accuracy across all clients. To address this issue, we propose a novel algorithm that leverages entropy-based aggregation combined with model and gradient alignments to simultaneously optimize fairness and global model performance. Our method employs a bi-level optimization framework, where we derive an analytic solution to the aggregation probability in the inner loop, making the optimization process computationally efficient. Additionally, we introduce an innovative alignment update and an adaptive strategy in the outer loop to further balance global model's performance and fairness. Theoretical analysis indicates that our approach guarantees convergence even in non-convex FL settings and demonstrates significant fairness improvements in generalized regression and strongly convex models. Empirically, our approach surpasses state-of-the-art federated fairness algorithms, ensuring consistent performance among clients while improving the overall performance of the global model.

## 1 Introduction

Federated Learning (FL) is a distributed learning paradigm that allows clients to collaborate with a central server to train a model (McMahan et al., 2017). To learn models without transferring data, clients process data locally and only periodically transmit model updates to the server, aggregating these updates into a global model. Due to data heterogeneity, intermittent client participation, and system heterogeneity, even the well-trained global model will perform better on some clients than others, which leads to performance unfairness (Shi et al., 2021). Achieving fairness is vital to prevent problems like performance discrimination, client disengagement, and legal and ethical concerns (Caton & Haas, 2020).

To address performance unfairness and ensure consistent performance in FL, several approaches have been explored with promising results (Li et al., 2019a; Kanaparthy et al., 2022; Zhao & Joshi, 2022; Kanaparthy et al., 2022; Pan et al., 2023; Papadaki et al., 2022). However, these methods often suffer from slow convergence and high communication and computation overheads (Wang et al., 2021; Huang et al., 2022; Chu et al., 2023). More critically, existing solutions tend to either sacrifice global model performance for fairness (Li et al., 2019a; Mohri et al., 2019; Zhang et al., 2023; Li et al., 2020a), while training an effective global model remains the core goal of FL (Kairouz et al., 2019). Although some efforts aim to balance fairness without degrading global performance (Lin et al., 2022; Li et al., 2021), they fail to model the problem directly and achieve suboptimal performance.

To overcome these limitations, we propose a novel algorithm, *FedEBA+*. It simultaneously optimizes fairness and global model performance through a bi-level optimization framework, leveraging Entropy-Based Aggregation plus model and gradient alignment. FedEBA+ assigns

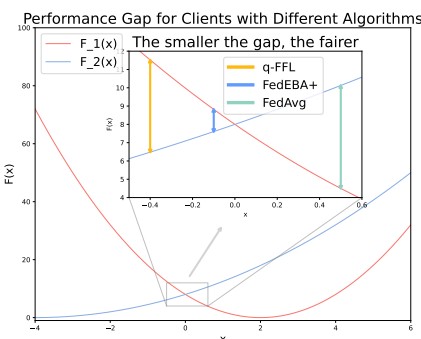

Figure 1: **Illustration of fairness improvement of FedEBA+ over q-FFL and FedAvg.** The performance gap means the performance difference between two clients, i.e., $\|F_1(x) - F_2(x)\|$. A smaller performance gap implies a smaller variance, resulting in a fairer method. For clients $F_1(x) = 2(x-2)^2$ and $F_2(x) = \frac{1}{2}(x+4)^2$ with global model $x^t = 0$ at round $t$, q-FFL, FedEBA+, and FedAvg produce $x^{t+1}$ of $-0.4$, $-0.1$, and $0.5$, respectively. The yellow, blue, and green double-arrow lines indicate the performance gap between the clients using different methods. FedEBA+ is the fairest method with the smallest loss gap, thus the smallest performance variance. Computational details are outlined in Appendix I.1.

higher aggregation weights to underperforming clients, providing an analytical solution that minimizes communication costs while improving both global performance and fairness.

In particular, the objective is based on a constrained entropy model for aggregation in FL. While entropy models have successfully promoted fairness in areas like data preprocessing (Singh & Vishnoi, 2014) and resource allocation (Johansson & Sternad, 2005), applying entropy to FL presents unique challenges. In FL, fairness requires equitable performance across diverse clients with heterogeneous data (Shi et al., 2021; Donahue & Kleinberg, 2021), not just uniform resource distribution. To address this, *FedEBA+* formulates entropy over aggregation distribution, constraining the distance between aggregated and ideal objectives (see Section 4.1), leading to an aggregation distribution proportional to loss. Compared with typical fair aggregation methods, like FedAvg (McMahan et al., 2017) and q-FFL (Li et al., 2019a), FedEBA+ ensures more uniform client performance (Figure 1). The maximum entropy model efficiently provides an analytic solution at each computation step, making the bi-level optimization problem computationally efficient without requiring cyclic updates.

**Our major contributions can be summarized as below:**

- We propose a bi-level optimization framework, involving a well-designed objective function capturing both the global model performance and the entropy-based fair aggregation, aimed at simultaneously enhancing fairness and the overall performance of FL. In the inner loop of the optimization framework, we derive the analytical solution to the inner variable, i.e., aggregation probability, ensuring computational efficiency and improving fairness. In the outer loop, we introduce an innovative alignment update and an adaptive strategy to dynamically balance the global model's performance and fairness.

- We propose *FedEBA+*, a novel FL algorithm for advocating fairness while improving the global model performance, embedding the analytical fair aggregation solution and the innovative model and gradient alignment update strategy. To alleviate the communication burdens, we further present a practical algorithm *Prac-FedEBA+*, achieving competitive performance with communication costs comparable to FedAvg.

- Theoretically, we provide the convergence guarantee for *FedEBA+* under a nonconvex setting. In addition, we establish the fairness of *FedEBA+* through performance variance analysis using both the generalized linear regression model and the strongly convex model.

- Empirical results on Fashion-MNIST, CIFAR-10, CIFAR-100, and Tiny-ImageNet demonstrate that *FedEBA+* surpasses existing fairness FL algorithms in both fairness and global model performance. Additionally, experiments highlight the efficiency of *Prac-FedEBA+*, showing its robustness to noisy labels and the enhancement for privacy protection.

## 2 RELATED WORK

There have been encouraging efforts to address fairness in Federated Learning, including function-based approaches like q-FFL (Li et al., 2019a) and AFL (Deng et al., 2020), gradient-based methods such as FedFV Wang et al. (2021) and MGDA (Hu et al., 2022; Pan et al., 2023), and personalized methods (Li et al., 2021; Lin et al., 2022). While these improve

fairness, they suffer from slow convergence (Li et al., 2019a; Deng et al., 2020) and high communication and computation overheads (Hu et al., 2022; Pan et al., 2023). Crucially, to the best of our knowledge, none of these methods simultaneously optimize fairness and global model performance or explicitly model the goal of balancing both, which is a key challenge in fair FL.

To this end, we propose a computationally efficient bi-level optimization algorithm designed to enhance global model performance while ensuring fairness among clients. Our approach effectively addresses key challenges in this research area. A more comprehensive discussion of the related work and fairness concepts can be found in Appendix A and Appendix B.

## 3 PRELIMINARIES AND METRICS

**Notations.** Let $m$ be the number of clients and $|S_t| = n$ be the number of selected clients for round $t$. We denote $K$ as the number of local steps and $T$ as the total number of communication rounds. We use $F_i(x)$ and $f(x)$ to represent the local and global loss of client $i$ with model $x$, respectively. Specifically, $x_{t,k}^i$ and $g_{t,k}^i = \nabla F_i(x_{t,k}^i, \xi_{t,k}^i)$ represents the model parameter and local gradient of the $k$-th local step in the $i$-th worker after the $t$-th communication, respectively. $x$ is the global model and $x_t$ is global model at round $t$. The global model update is denoted as $\Delta_t = 1/\eta(x_{t+1} - x_t)$, while the local model update is represented as $\Delta_t^i = x_{t,k}^i - x_{t,0}^i$. Here, $\eta$ and $\eta_L$ correspond to the global and local learning rates, respectively.

**Problem formulation.** The typically FL objective can be formulated as follows:

$$\min_x f(x) = \sum_{i=1}^m p_i F_i(x), \tag{1}$$

where $F_i(x) = \mathbb{E}_{\xi_i \sim D_i} F_i(x, \xi_i)$ is the local objective function of client $i$ over data distribution $D_i$, $\xi_i$ means the sampled data of client $i$ and $p_i$ represents the aggregation weight of client $i$.

In this paper, our goal is to improve the performance of the global model, specifically by minimizing the objective loss function, while also reducing performance variance. This motivates us to establish the following optimization objective as our *final objective*:

$$x^* = \arg\min_x f(x) = \arg\min_x \left\{ \sum_{i=1}^m p_i F_i(x) + \beta \Phi(x) \right\}, \tag{2}$$

where $x^*$ is the optimal model parameter, $F_i(x)$ is the local loss on client $i$, and $f(x)$ represents the global model's loss, aimed at improving the global model's performance. $\beta > 0$ is the penalty coefficient of the fairness regularization, while $\Phi(x)$ is the regularization term that aims to improve fairness. Thus, optimizing this objective entails simultaneously enhancing the global model's performance and reducing variance. We explicitly formulate $\Phi(x)$ in Section 4.2, building on the fair aggregation optimization in Section 4.1, and rewrite (2) as a bi-level optimization Problem (6).

**Metrics.** This paper aims to *1) promote fairness* in FL while *2) enhance the global model's performance*. Typically, the global model's performance is evaluated based on its accuracy or loss. Regarding the fairness metric, we adhere to the definition proposed by (Li et al., 2019a), which employs the variance of clients' performance as the fairness metric:

**Definition 3.1** (Fairness via variance). *A model $x_1$ is more fair than $x_2$ if the test performance distribution of $x_1$ across the network with $m$ clients is more uniform than that of $x_2$, i.e. $\text{var}\{F_i(x_1)\}_{i \in [m]} < \text{var}\{F_i(x_2)\}_{i \in [m]}$, where $F_i(\cdot)$ denotes the test loss of client $i \in [m]$ and $\text{var}\{F_i(x)\} = \frac{1}{m}\sum_{i=1}^m \left[F_i(x) - \frac{1}{m}\sum_{i=1}^m F_i(x)\right]^2$ denotes the variance.*

Ensuring the global model's performance is the fundamental goal of FL. However, fairness-targeted algorithms may compromise high-performing clients to mitigate variance (Shi et al., 2021). Our evaluation of fairness algorithms extends beyond global accuracy, considering the accuracy of the best 5% and worst 5% clients. This analysis, also viewed as a form of robustness in some studies (Yu et al., 2023; Li et al., 2021), provides insights into potential compromises.

---

**Algorithm 1** FedEBA+

---

1: **Input:** Number of clients $m$, global learning rate $\eta$, local learning rate $\eta_l$, number of local epoch $K$, total training rounds $T$, threshold $\theta$.
2: **Output:** Final model parameter $x_T$.
3: **Initialize:** model $x_0$, guidance vector $\mathbf{r} = [1, \cdots, 1]$.
4: **for** round $t = 1, \ldots, T$ **do**
5:     Server selects a set of clients $|S_t|$ and broadcast model $x_t$;
6:     Server collects selected clients' loss $\mathbf{L} = [F_1(x_t), \ldots, F_{|S_t|}(x_t)]$;
7:     **if** $\arccos(\frac{\mathbf{L}, \mathbf{r}}{\|\mathbf{L}\| \cdot \|\mathbf{r}\|}) > \theta$ **then**
8:         Sever receives $\nabla F_i(x_t)$, calculates the fair gradient and broadcast to clients: $\tilde{g}^{b,t} = \sum_{i \in S_t} \frac{\exp[F_i(x_t)/\tau)]}{\sum_{j \in S_t} \exp[F_j(x_t)/\tau]} \nabla F_i(x_t)$;
9:         **for** Client $i \in S_t$ in parallel **do**
10:           **for** $k = 0, \cdots, K - 1$ **do**
11:             $h_{t,k}^i \leftarrow (1 - \alpha)\nabla F_i(x_{t,k}^i; \xi_i) + \alpha \tilde{g}^{b,t}$;
12:           **end for**
13:           $\Delta_t^i = x_{t,K}^i - x_{t,0}^i = -\eta_L \sum_{k=0}^{K-1} h_{t,k}^i$;
14:         **end for**
15:         Aggregation: $\Delta_t = \sum_{i \in S_t} p_i \Delta_t^i$, where $p_i = \frac{\exp[F_i(x_{t,K}^i)/\tau)]}{\sum_{i \in S_t} \exp[F_i(x_{t,K}^i)/\tau]}$;
16:     **else**
17:         **for** each worker $i \in S_t$, in parallel **do**
18:           **for** $k = 0, \cdots, K - 1$ **do**
19:             $x_{t,k+1}^i = x_{t,k}^i - \eta_L \nabla F_i(x_{t,k}^i; \xi_i)$;
20:           **end for**
21:           Let $\Delta_t^i = x_{t,K}^i - x_{t,0}^i = -\eta_L \sum_{k=0}^{K-1} \nabla F_i(x_{t,k}^i; \xi_i)$ and $\tilde{\Delta}_t^{a,i} = x_{t,1}^i - x_{t,0}^i$;
22:         **end for**
23:         Server aggregates model update by Eq. (8);
24:     **end if**
25:     Server update: $x_{t+1} = x_t + \eta \Delta_t$;
26: **end for**

---

# 4 FEDEBA+: AN EFFECTIVE FAIR ALGORITHM

In this section, we first define the constrained maximum entropy for aggregation probability and derive a fair aggregation strategy (Sec 4.1). We then introduce a bi-level optimization objective for fair FL (Sec 4.2), which enhances the global model's performance through model alignment and improves fairness through gradient alignment (Sec 4.3). The complete algorithm, covering entropy-based aggregation, model alignment, and gradient alignment, is presented in Algorithm 1.

## 4.1 FAIR AGGREGATION: EBA

Inspired by the Shannon entropy to fairness (Jaynes, 1957), which ensures unbiased probability distribution by maximizing neutrality towards unobserved information and eliminating inherent bias (Hubbard et al., 1990; Sampat & Zavala, 2019), we formulate the following optimization problem with designed constraints on FL aggregation:

$$\max_{p_i, \forall i \in [m]} \mathbb{H}(p_i) := -\sum_{i=1}^m p_i \log(p_i) \quad s.t. \quad \sum_{i=1}^m p_i = 1, \ p_i \geq 0, \ \sum_{i=1}^m p_i F_i(x_i) = \tilde{f}(x). \tag{3}$$

$\mathbb{H}(p_i)$ denotes the entropy of aggregation probability $p_i$, and $\tilde{f}(x)$ signifies the ideal loss, representing the global model's performance under ideal training setting, which is unknown but whose gradient can be approximately formulated and utilized as shown in Eq. (8) and Eq. (10), detailed in the next section. The classical entropy model reduces prior distribution knowledge and avoids bias from subjective influences. Compared to the existing entropy model of fairness Johansson & Sternad (2005), we first incorporate the FL constraints $\sum_{i=1}^m p_i F_i(x_i) = \tilde{f}(x)$ to force aggregation into the fair regularization region, specifically improving fairness. Maximizing constrained entropy implies greater fairness, as shown in the toy example in Appendix I.1.

**Proposition 4.1.** *By solving the constrained maximum entropy problem, we propose an aggregation strategy called **EBA** to enhance fairness in FL, expressed as follows:*

$$p_i = \frac{\exp[F_i(x_i)/\tau)]}{\sum_{j=1}^N \exp[F_j(x_j)/\tau]}, \tag{4}$$

*where $\tau > 0$ is the temperature, and the derivation of $\tau$ is related to $\tilde{f}(x)$.*

Details for deriving the above proposition and the proof of the uniqueness of the solution for the constrained maximum entropy model are provided in Appendix C.1 and K, respectively.

Proposition 4.1 shows that assigning higher aggregation weights to underperforming clients directs the aggregated global model's focus toward these users, enhancing their performance and reducing the gap with top performers, ultimately promoting fairness, as shown in the toy case of Figure 1 and experiments in Table 15. It is worth noting that the aggregation probability can be solved in closed form, relying solely on the loss of the local model, making it computationally efficient.

When taking into account the prior distribution of aggregation probability $p_i$, which is typically expressed as the relative data ratio $q_i = n_i/\sum_{i \in S_t} n_i$ where $n_i$ is the number of data in client $i$, the expression of fair aggregation probability becomes $p_i = \frac{q_i \exp[F_i(x)/\tau)]}{\sum_{j=1}^N q_j \exp[F_j(x)/\tau]}$. Without loss of generality, we utilize Eq. (4) to represent entropy-based aggregation in this paper. The derivations for fair aggregation probability expression w/o prior distribution are given in Appendix C.1.

**Remark 4.2** (The effectiveness of $\tau$ on fairness). *$\tau$ controls the fairness level as it decides the spreading of weights assigned to each client. A higher $\tau$ results in uniform weights for aggregation, while a lower $\tau$ yields concentrated weights. This aggregation algorithm degenerates to FedAvg(McMahan et al., 2017) or AFL (Mohri et al., 2019) when $\tau$ is extremely large or small. We further discuss the effectiveness of $\tau$ in Appendix M.6.*

**Remark 4.3** (Robustness of EBA). *Typical aggregation methods focusing on fairness or heterogeneity often suffer significant performance degradation in scenarios with noisy labels (Pillutla et al., 2019; Yang et al., 2022; Xu et al., 2022). We demonstrate that our aggregation method maintains robustness to noisy labels by extending the local loss $F_i(x)$ to a robust loss $F_i^r(x)$. The aggregation then becomes:*

$$p_i = \frac{\exp\left(F_i^r(x)/\tau\right)}{\sum_j \exp\left(F_j^r(x)/\tau\right)}, \qquad F_i^r(x) = \mathbb{E}_{\xi_i}\left[F_i^{cls}(x;\xi_i) + \gamma F_i^{reg}(x; Augment(\xi_i))\right], \tag{5}$$

*where $F_i^{cls}(x;\xi_i)$ represents the cross-entropy loss, and $F_i^{reg}(x; Augment(\xi_i))$ denotes the self-distillation loss with augmented data. The robust loss mitigates model output discrepancies between original and mildly augmented instances, addressing noisy label scenarios and enhancing robustness. The detailed LSR implementation algorithm is presented in Algorithm 2 of Appendix D.*

### 4.2 BI-LEVEL OPTIMIZATION FORMULATION AND ALIGNMENT UPDATE

Recall the *final objective* (2) to develop an objective function that simultaneously improves fairness and global model performance. Based on the proposed maximum entropy model, we define $\Phi = -\left[\sum_{i=1}^N p_i \log p_i + \lambda_0 \left(\sum_{i=1}^N p_i - 1\right) + \frac{1}{\tau}\left(\tilde{f}(x) - \sum_{i=1}^N p_i F_i(x)\right)\right]$. Maximizing $\Phi$ with respect to $p_i$ ensures the same fair aggregation result as proposition 4.1. Thus, we develop *final objective* into a bi-level optimization objective that enhances model performance during updates while maintaining aggregation fairness, formulated as below:

$$\min_x \max_{p_i} L(x, p_i) := \sum_{i=1}^N p_i F_i(x) - \beta \left[\sum_{i=1}^N p_i \log p_i \right.$$
$$\left. + \lambda_0 \left(\sum_{i=1}^N p_i - 1\right) + \frac{1}{\tau}\left(\tilde{f}(x) - \sum_{i=1}^N p_i F_i(x)\right)\right], \tag{6}$$

For the inner loop of Problem (6), maximizing the objective $L(x, p_i)$ over the inner variable $p_i$ results in the same analytical solution as the aggregation probability in Eq. (4). For

the outer loop of Problem (6), minimizing the objective $L(x, p_i)$ with respect to the outer variable $x$ introduces the following model update formula:

$$\frac{\partial L(x, p_i)}{\partial x} = (1 - \alpha) \sum_{i=1}^{m} p_i \nabla F_i(x) + \alpha \nabla \tilde{f}(x), \tag{7}$$

where $\alpha = \beta / \tau \geq 0$ is a constant. Then the global model is updated by $\Delta_t = -\eta_L \frac{\partial L(x, p_i)}{\partial x} = -\eta_L (1 - \alpha) \sum_{i=1}^{m} p_i \nabla F_i(x) - \alpha \eta_L \nabla \tilde{f}(x)$.

The proposed update formulation integrates the traditional federated learning (FL) update with the *ideal gradient* $\nabla \tilde{f}(x)$ to align model updates. The choice of approximation for the ideal loss gradient, $\nabla \tilde{f}(x)$, influences the extent of performance improvement. Specifically, $\nabla \tilde{f}(x)$ can represent either the *ideal global gradient* $\nabla \tilde{f}^a(x_t)$ to enhance global model performance or the *ideal fair gradient* $\nabla \tilde{f}^b(x_t)$ to improve fairness, as detailed in the subsequent section.

### 4.3 Adaptive Balance between Fairness and Global Performance Improvement

Our approach leverages an alignment update strategy, derived from the outer optimization loop, to simultaneously enhance global model performance and fairness through entropy-based aggregation. This process is dynamically adjusted to prioritize either fairness or global performance based on the current state of the system. When local updates diverge significantly from fairness, improving fairness also mitigates local shifts, thereby boosting global performance (Karimireddy et al., 2020b). Conversely, when fairness is within an acceptable range, we focus on enhancing global performance through server-side alignment updates, formulated using a momentum-like method.

To achieve this adaptive balance, we employ an arccos-based scheme. If the arccos value of the clients' performance vector $\mathbf{L} = [F_1(x_t), \ldots, F_{|S_t|}(x_t)]$ and the guidance vector (an all-ones vector of length $|S_t|$) exceeds a predefined threshold (fair angle $\theta$), the system is deemed unfair, and gradient alignment for fairness is applied. Otherwise, if the arccos value is below the threshold, the system is considered to be within the tolerable fairness range, as illustrated in Figure 2.

**Model Alignment for Improving Global Accuracy.** Based on the proposed model update formula (7), we propose an server-side model update approach to improve the global model performance. The ideal global gradient $\nabla \tilde{f}(x) := \nabla \tilde{f}^a(x_t) = \tilde{\Delta}_t^a$ aligns the aggregated model to facilitate updates towards the global optimum. Unable to directly obtain the ideal global gradient, we estimate it by averaging local one-step gradients and align the model update. Utilizing local SGD with $x_{t+1} = x_t - \eta \frac{\partial L(x)}{\partial x}$ and $x_{t+1} = x_t - \eta \Delta_t$, we have

$$\Delta_t = (1 - \alpha) \sum_{i \in S_t} p_i \sum_{k=0}^{K-1} \nabla F_i(x_{t,k}^i; \xi_{t,k}^i) + \alpha \nabla \tilde{f}^a(x) = (1 - \alpha) \sum_{i \in S_t} p_i \Delta_t^i + \alpha \tilde{\Delta}_t^a, \tag{8}$$

where $p_i$ follows the proposed aggregation probability, i.e., $p_i = \frac{\exp[F_i(x_{t,K}^i) / \tau)]}{\sum_{i \in S_t} \exp[F_i(x_{t,K}^i) / \tau]}$. Here, $\tilde{\Delta}_t^a$ denotes the aggregation of one-step local updates, defined as follows:

$$\tilde{\Delta}_t^a = \frac{1}{|S_t|} \sum_{i \in S_t} \tilde{\Delta}_t^{a,i} = \frac{1}{|S_t|} \sum_{i \in S_t} (x_{t,1}^i - x_{t,0}^i). \tag{9}$$

When the client's dataset size $n_i$ varies, the expression of $\tilde{\Delta}_t^a$ should be $\tilde{\Delta}_t^a = \sum_{i \in S_t} \frac{n_i}{\sum_{j \in S_t} n_j} \tilde{\Delta}_t^{a,i}$. The model alignment update is outlined in Algorithm 1 (Steps 17-23). The rationale for utilizing the above equation to estimate the ideal global model is twofold: 1) a single local update corresponds to an unshifted update on local data, whereas multiple local updates introduce model bias in heterogeneous FL (Karimireddy et al., 2020b); 2) the expectation of sampled clients' data over rounds represents the global data due to unbiased random sampling (Wang et al., 2022).

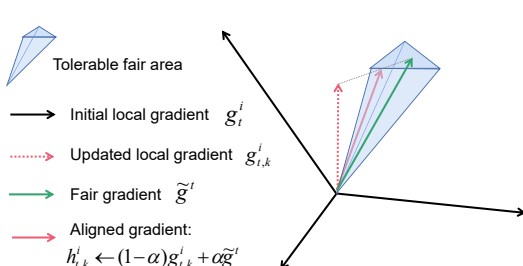

Figure 2: **Gradient Alignment improves fairness.** Gradient alignment ensures that each local step's gradient stays on track and does not deviate too far from the fair direction. It achieves this by constraining the aligned gradient, denoted by $h^i_{k,t}$, to fall within the tolerable fair area. The gradient $g^i_t$ represents the gradient of global model for each client in round $t$, while $\tilde{g}^t = \nabla F_i(x_t)$ denotes the ideal fair gradient for model $x_t$. The gradient $g^i_{k,t} = \nabla F_i(x^i_{t,k}; \xi_i)$ is the gradient of client $i$ at round $t$ and local epoch $k$.

In the figure legend:
- Tolerable fair area
- Initial local gradient $g^i_t$
- Updated local gradient $g^i_{t,k}$
- Fair gradient $\tilde{g}^t$
- Aligned gradient:
  $h^i_{t,k} \leftarrow (1-\alpha)g^i_{t,k} + \alpha\tilde{g}^t$

**Gradient Alignment for Fairness.** To enhance fairness, we define $\nabla\tilde{f}(x) := \nabla\tilde{f}^b(x_t) = \sum_{i\in S_t} p_i \sum_{k=0}^{K-1} \nabla\tilde{f}^b(x^i_{t,k})$ as the ideal fair gradient to align the local model updates. To align gradients, the server receives $\nabla F_i(x_t)$ and $F_i(x_t)$ from clients, utilizing entropy-based aggregation to assess each client's importance. The fair update is denoted as $\Delta_t = (1-\alpha)\sum_i p_i \sum_{k=0}^{K-1} \nabla F_i(x^i_{t,k}; \xi^i_{t,k}) + \alpha\nabla\tilde{f}^b(x) = \sum_i p_i \sum_{k=0}^{K-1} \left[(1-\alpha)\nabla F_i(x^i_{t,k}; \xi^i_{t,k}) + \alpha\nabla\tilde{f}^b(x^i_{t,k})\right]$. Subsequently, the ideal fair gradient $\nabla\tilde{f}^b(x^i_{t,k})$ is estimated by:

$$\nabla\tilde{f}^b(x^i_{t,k}) = \tilde{g}^{b,t} = \sum_{i\in S_t} \tilde{p}_i \nabla F_i(x_t), \tag{10}$$

where $\tilde{p}_i = \exp[F_i(x_t)/\tau] / \sum_{j\in S_t} \exp[F_j(x_t)/\tau]$, $\tilde{g}^{b,t}$ represents the fair gradient of the selected clients, obtained using the global model's performance on these clients without local shift (i.e., one local update). In particular, for each local epoch $k$, we use the same fair gradient that is regardless of $k$. Therefore, the aligned gradient of model $x^i_{t,k}$ can be expressed as:

$$h^i_{t,k} \leftarrow (1-\alpha)\nabla F_i(x^i_{t,k}; \xi_i) + \alpha\tilde{g}^{b,t}. \tag{11}$$

The fairness alignment is depicted in Algorithm 1, Steps 8-15.

## 4.4 PRACTICAL GRADIENT ALIGNMENT TO REDUCE COMMUNICATION.

Note that in the above discussion, the server needs to obtain the one local update to calculate the aligned gradient $\tilde{g}^{b,t}$ and sends it back to clients for local update. Considering the communication burden of FL, we propose a practical version of the gradient alignment method:

**Proposition 4.4.** *For approximating the aligned gradient and overcoming the communication overhead issue, we use the average of multiple local updates to approximate the one-step gradient. Then, the fair gradient is approximated by:*

$$\tilde{g}^{b,t} = \sum_{i\in S_t} \frac{\exp[F_i(x_t)/\tau]}{\sum_{j\in S_t} \exp[F_j(x_t)/\tau]} \frac{1}{K} \sum_{k=0}^{K-1} \nabla F_i(x^i_{t,k}; \xi_i). \tag{12}$$

*In this way, the client only needs to communicate the model once to the server, same as FedAvg. The complete practical algorithm, named Prac-FedEBA+, is presented in Algorithm 3.*

# 5 ANALYSIS OF CONVERGENCE AND FAIRNESS

In this section, we analyze convergence under a nonconvex setting and evaluate fairness using variance and Pareto-optimality.

## 5.1 CONVERGENCE ANALYSIS OF FEDEBA+

To facilitate the theoretical analysis, we adopt common assumptions for nonconvex federated learning: L-smoothness, unbiased local gradient estimators, and bounded gradient dissimilarity. See Appendix G for assumptions' details.

**Theorem 5.1.** *Under Assumption 1–3, and let constant local and global learning rate $\eta_L$ and $\eta$ be chosen such that $\eta_L < \min\left(1/(8LK), C\right)$, where $C$ is obtained from the condition that $\frac{1}{2} - 10L^2 \frac{1}{m}\sum_{i-1}^m K^2 \eta_L^2 (A^2 + 1)(\chi_{p\|w}^2 A^2 + 1) > C > 0$, and $\eta \leq 1/(\eta_L L)$. In particular, let $\eta_L = \mathcal{O}\left(\frac{1}{\sqrt{TKL}}\right)$ and $\eta = \mathcal{O}\left(\sqrt{Km}\right)$, the convergence rate of Algorithm 1 (FedEBA+) with $\alpha = 0$ is:*

$$\min_{t \in [T]} \mathbb{E}\left\|\nabla f\left(\boldsymbol{x}_t\right)\right\|^2 \leq \mathcal{O}\left(\frac{(f^0 - f^*) + {}^m/_2 \sum_i w_i^2 \sigma_L^2}{\sqrt{mKT}}\right) + \mathcal{O}\left(\frac{5(\sigma_L^2 + 4K\sigma_G^2) + 40K(A^2+1)\chi_{\boldsymbol{w}\|\boldsymbol{p}}^2 \sigma_G^2}{2KT}\right).$$

(13)

Here, $A \geq 0$ is a constant defined in Assumption 3, and $\boldsymbol{w}$ is the prior aggregation distribution detailed in Lemma H.1. The proof details of Theorem 5.1 are provided in Appendix H.

**Remark 5.2.** *According to the property of unified probability, we know $\frac{1}{m} \leq \sum_{i=1}^m w_i^2 \leq 1$, where the right inequality comes from $\sum_i w_i^2 \leq \sum_i w_i$ and the left inequality comes from Cauchy-Schwarz inequality. Therefore, the worst case of the convergence rate will be $\mathcal{O}(\frac{\sqrt{m}}{\sqrt{KT}} + \frac{1}{T})$.*

**Remark 5.3.** *When $\alpha \neq 0$, the convergence rate of FedEBA+ is: $\min_{t \in [T]} \mathbb{E}\left\|\nabla f\left(\boldsymbol{x}_t\right)\right\|^2 \leq \mathcal{O}(\frac{(1-\alpha)^2 \sum_i w_i^2 \sqrt{m}\sigma_L^2 + \alpha^2 \sqrt{K}\rho^2}{\sqrt{KT}} + \frac{1}{T})$, where $\sigma_L \sim \rho$ by Assumption 4, thus a larger $\alpha$ indicating a tighter convergence upper bound than only using reweight aggregation with $\alpha = 0$. $K$ represents the local epoch times (in each communication round) and $m$ represents the client numbers, usually client numbers are larger than the local epoch in the cross-device FL. In addition, when $w_i = \frac{1}{m}$, i.e., uniform aggregation, the rate is $\mathcal{O}(\frac{(1-\alpha)^2 \sigma_L^2 + \alpha^2 \sqrt{K/m}\rho^2}{\sqrt{mKT}} + \frac{1}{T})$. When $\sqrt{K/m} << 1$, using the proposed alignment update results in a faster convergence rate than FedAvg. The proof details are provided in Appendix H.2.*

### 5.2 Fairness Analysis of FedEBA+

**Variance analysis.** We analyze the performance variance of clients of FedEBA+ using both the generalized linear regression model and the strongly convex model.

**Theorem 5.4.** *Under Algorithm 1, FedEBA+ exhibits smaller performance variance than FedAvg:*

*(1) For the generalized regression model, as per the setup in Li et al. (2020a), it is formulated as $f(\mathbf{x}; \xi) = T(\xi)^\top \mathbf{x} - A(\xi)$, where $T(\xi)$ represents the generalized regression coefficient and $A(\xi)$ denotes the Gaussian noise term. We then derive the test variance of FedEBA+ and compare it with FedAvg:*

$$\operatorname{var}\left(F_i^{test}\left(\boldsymbol{x}_{EBA+}\right)\right) = \frac{\tilde{b}^2}{4}\operatorname{var}\left(\|\tilde{\mathbf{w}} - \mathbf{w}_i\|_2^2\right)$$

(14)

$$\operatorname{var}\{F_i^{test}(\boldsymbol{x}_{EBA+})\}_{i \in m} \leq \operatorname{var}\{F_i^{test}(\boldsymbol{x}_{Avg})\}_{i \in m}$$

(15)

*where $\tilde{\mathbf{w}} = \sum_{i=1}^m p_i \mathbf{w}_i$, $\mathbf{w}_i$ represents the true parameter on client $i$, , and $\tilde{b}$ is a constant that approximates $b_i$ in $\boldsymbol{\Xi}_i^\top \boldsymbol{\Xi}_i = mb_i\mathbf{I}_d$, where $\Xi_i = [T(\xi_{i,1}), \ldots, T(\xi_{i,n})]$. The data heterogeneity is reflected in the heterogeneity of $\mathbf{w}_i$.*

*(2) For the strongly convex setting, we assume the client's loss to be smooth and strongly convex, following the setting in (Chu et al., 2023). By assuming the existence of an outlier, we derive the test variance of FedEBA+ and compare it with FedAvg:*

$$\operatorname{var}\left(F_i^{test}\left(\boldsymbol{x}_{EBA+}\right)\right) = \frac{1}{N}\sum_{i=1}^N \tilde{L}_i^2 - \left(\frac{1}{N}\sum_{i=1}^N \tilde{L}_i\right)^2,$$

(16)

$$\operatorname{var}\{F_i^{test}(\boldsymbol{x}_{EBA+})\}_{i \in m} \leq \operatorname{var}\{F_i^{test}(\boldsymbol{x}_{Avg})\}_{i \in m},$$

(17)

*where $\tilde{L}_i$ is the test loss of FedEBA+ on client $i$, distinguishing from training loss $F_i(x)$.*

Details regarding the setting of the linear regression model, smooth and strongly convex assumptions, and the derivation details are presented in Appendix I.2 and Appendix I.3.

In addition to analyzing fairness variance in federated learning, we demonstrate that our algorithm, FedEBA+, satisfies Pareto-optimality and uniqueness as per Property 1 of (Sampat & Zavala, 2019). This supports the fairness effectiveness of our algorithm, with further details provided in Appendix J and Appendix K.

Table 1: **Performance of algorithms on FashionMNIST and CIFAR-10.** We report the accuracy of global model, variance fairness, worst 5%, and best 5% accuracy. The data is divided into 100 clients, with 10 clients sampled in each round. All experiments are running over 2000 rounds for a single local epoch ($K = 10$) with local batch size $= 50$, and learning rate $\eta = 0.1$. The reported results are averaged over 5 runs with different random seeds. We highlight the best and the second-best results by using **bold font** and blue text.

| Algorithm | FashionMNIST | | | | CIFAR-10 | | | |
|---|---|---|---|---|---|---|---|---|
| | Global Acc. ↑ | Var. ↓ | Worst 5% ↑ | Best 5% ↑ | Global Acc. ↑ | Var. ↓ | Worst 5% ↑ | Best 5% ↑ |
| FedAvg | 86.49 ±0.09 | 62.44 ±4.55 | 71.27 ±1.14 | 95.84 ±0.35 | 67.79 ±0.35 | 103.83 ±10.46 | 45.00 ±2.83 | 85.13 ±0.82 |
| FedSGD | 83.79 ±0.28 | 81.72 ±0.26 | 61.19 ±0.30 | 96.60 ±0.20 | 67.48 ±0.37 | 95.79 ±4.03 | 48.70 ±0.9 | 84.20 ±0.40 |
| q-FFL | 86.57 ±0.19 | 54.91 ±2.82 | 70.88 ±0.98 | 95.06 ±0.17 | 68.76 ±0.22 | 97.81 ±2.18 | 48.33 ±0.84 | 84.51 ±1.33 |
| FedMGDA+ | 84.64 ±0.25 | 57.89 ±6.21 | **73.49 ±1.17** | 93.22 ±0.20 | 65.19 ±0.87 | 89.78 ±5.87 | 48.84 ±1.12 | 81.94 ±0.67 |
| Ditto | 86.37 ±0.13 | 55.56 ±5.43 | 69.20 ±0.37 | 95.79 ±0.38 | 60.11 ±4.41 | 85.99 ±7.13 | 42.20 ±2.20 | 77.90 ±4.90 |
| PropFair | 85.51 ±0.28 | 75.27 ±5.38 | 63.60 ±0.53 | 97.60 ±0.19 | 65.79 ±0.53 | 79.67 ±5.71 | 49.88 ±0.93 | 82.40 ±0.40 |
| TERM | 84.31 ±0.38 | 73.46 ±2.06 | 68.23 ±0.10 | 94.16 ±0.16 | 65.41 ±0.37 | 91.99 ±2.69 | 49.08 ±0.66 | 81.98 ±0.19 |
| FOCUS | 86.24 ±0.18 | 61.15 ±1.17 | 68.15 ±0.25 | **98.50 ±0.10** | 59.60 ±1.52 | 455.14 ±11.19 | 9.54 ±0.18 | **87.72 ±0.12** |
| lp-proj | 86.21 ±0.02 | 56.71 ±2.25 | 68.47 ±0.37 | 97.86 ±0.52 | 68.86 ±0.51 | 78.65 ±7.01 | 49.53 ±1.11 | 83.33 ±1.23 |
| Rank-Core-Fed | 85.54 ±0.33 | 58.19 ±2.83 | 67.80 ±0.55 | 96.60 ±0.40 | 67.15 ±1.12 | 87.02 ±2.46 | 45.41 ±0.62 | 85.82 ±0.20 |
| Prac-FedEBA+ | 86.62 ±0.07 | 46.41 ±0.88 | 71.40 ±0.15 | 96.1 ±0.46 | 69.83 ±0.34 | 74.16 ±1.66 | 52.40 ±0.50 | 84.10 ±0.39 |
| FedEBA+ | **87.50 ±0.19** | **43.41 ±4.34** | 72.07 ±1.47 | 95.91 ±0.19 | **72.75 ±0.25** | **68.71 ±4.39** | **55.80 ±1.28** | 86.93 ±0.52 |

Table 2: **Performance of algorithms on CIFAR-100 and Tiny-ImageNet.** We include FedFV (Wang et al., 2021) and FedProx (Li et al., 2020b) to compare the performance.

| Algorithm | CIFAR-100 | | | | Tiny-ImageNet | | | |
|---|---|---|---|---|---|---|---|---|
| | Global Acc. ↑ | Std. ↓ | Worst 5% ↑ | Best 5% ↑ | Global Acc. ↑ | Var. ↓ | Worst 5% ↑ | Best 5% ↑ |
| FedAvg | 30.94 ±0.04 | 17.24 ±0.08 | 0.20 ±0.00 | 65.90 ±1.48 | 61.99 ±0.17 | 19.62 ±1.12 | 53.60 ±0.06 | **71.18 ±0.13** |
| q-FFL | 24.97 ±0.46 | 14.54 ±0.21 | 0.00 ±0.00 | 45.04 ±0.53 | 62.42 ±0.46 | 15.44 ±1.89 | 54.13 ±0.11 | 70.01 ±0.09 |
| AFL | 20.84 ±0.43 | **11.32 ±0.20** | **4.03 ±0.14** | 50.83 ±0.30 | 62.09 ±0.53 | 16.47 ±0.88 | 54.65 ±0.64 | 68.83 ±1.30 |
| FedProx | 31.50 ±0.04 | 17.50 ±0.09 | 0.41 ±0.00 | 64.50 ±0.11 | 62.05 ±0.04 | 16.21 ±1.13 | 54.41 ±0.47 | 69.92 ±0.26 |
| FedFV | 31.23 ±0.04 | 17.50 ±0.02 | 0.20 ±0.00 | 66.05 ±0.11 | 62.13 ±0.08 | 15.69 ±0.58 | 53.92 ±0.30 | 69.60 ±0.31 |
| FedMGDA+ | 31.34 ±0.12 | 16.61 ±0.09 | 0.74 ±0.12 | 65.21 ±1.15 | 62.33 ±0.26 | 17.49 ±0.31 | 53.77 ±0.16 | 70.04 ±0.30 |
| PropFair | 30.85 ±0.07 | 16.52 ±0.24 | 0.29 ±0.04 | 64.33 ±0.71 | 62.01 ±0.17 | 16.81 ±0.28 | 53.83 ±0.42 | 69.95 ±0.18 |
| TERM | 28.98 ±0.45 | 17.19 ±0.13 | 0.37 ±0.02 | 63.85 ±0.40 | 61.29 ±0.37 | 19.36 ±0.94 | 52.92 ±0.65 | 69.82 ±0.44 |
| Prac-FedEBA+ | 31.95 ±0.12 | 15.23 ±0.09 | 1.05 ±0.25 | 67.20 ±0.03 | 63.43 ±0.56 | 15.13 ±0.48 | 54.38 ±0.67 | 70.15 ±0.33 |
| FedEBA+ | **31.98 ±0.30** | 13.75 ±0.16 | 1.12 ±0.05 | **67.94 ±0.54** | **63.75 ±0.09** | **13.89 ±0.72** | **55.64 ±0.18** | 70.93 ±0.22 |

## 6 NUMERICAL RESULTS

**Metrics and Baselines.** We use **variance**, worst 5% accuracy, and best 5% accuracy as performance metrics for fairness evaluation, and **global accuracy** to evaluate the global model's performance. Additionally, the **coefficient of variation** ($C_v = \frac{std}{acc}$) (Jain et al., 1984), the ratio of standard deviation and accuracy, is used to capture the fairness and global performance simultaneously. We compare FedEBA+ with FedAvg, FedSGD (McMahan et al., 2016), and fair FL algorithms, including AFL (Mohri et al., 2019), q-FFL (Li et al., 2019a), FedMGDA+(Hu et al., 2022), PropFair (Zhang et al., 2023), TERM (Li et al., 2020a), FOCUS (Chu et al., 2023), Ditto (Li et al., 2021) and lp-proj (Lin et al., 2022). Additional implementation details, such as models and hyperparameters, are available in Appendix L.

**FedEBA+ can significantly improve both fairness and global accuracy simultaneously.** In Table 1 and Table 2, we compare FedBEA+'s performance with other fairness FL algorithms on diverse datasets and models. The result reveals the following insights: 1) *FedEBA+ significantly reduces performance variance and improves global accuracy simultaneously.* The variance improvement is **3×** on FashionMNIST and **1.5×** on CIFAR-10 compared to the best-performing baseline. Accuracy improves by **4%** on CIFAR-10 and **3%** on CIFAR-100 and Tiny-ImageNet. 2) *Other baselines face an accuracy-variance trade-off,* showing either lower global accuracy or limited improvement compared to FedAvg. 3) *With the same communication cost as FedAvg, Prac-FedEBA+ surpasses other baselines.* Moreover, Figure 3(a) clearly shows FedEBA+'s superiority in both fairness and global accuracy. Similarly, Table 18 in Appendix M shows that FedEBA+ achieves nearly **4×** better performance in $C_v$, capturing both fairness and accuracy simultaneously.

**Fast convergence and stability to hyperparameters of FedEBA+.** Figure 3(b) shows that FedEBA+ converges faster and achieves better accuracy than others. Figure 5(a) indicates that increasing $\alpha$ improves fairness but decreases accuracy. Figure 5(b) demonstrates that decreasing $\tau$ enhances fairness, with $\tau > 1$ generally leading to better global accuracy.

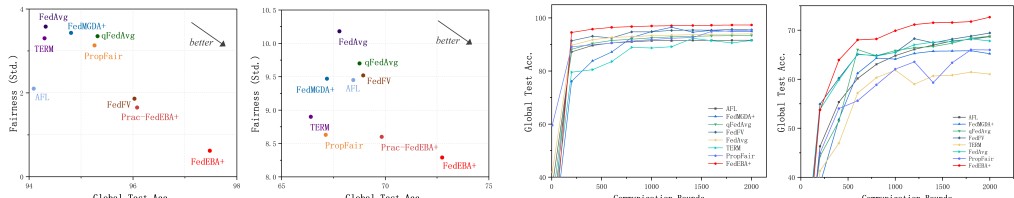

(a) **Performance of variance and accuracy**  (b) **Performance of convergence**

Figure 3: Performance of algorithms on (a) left: variance and accuracy on MNIST, (a) right: variance and accuracy on CIFAR-10, (b) left: convergence on MNIST, (b) right: convergence on CIFAR-10.

Table 3: **Ablation study for $\theta$ of FedEBA+.**

| FedEBA+$_{\theta=}$ | FashionMNIST (MLP) | | | CIFAR-10 (CNN) | | |
|---|---|---|---|---|---|---|
| | Global Acc. | Var. | Additional cost | Global Acc. | Var. | Additional cost |
| $\theta = 0°$ | $87.50 \pm 0.19$ | $43.41 \pm 4.34$ | 50.0% | $72.75 \pm 0.25$ | $68.71 \pm 4.39$ | 50.0% |
| $\theta = 15°$ | $87.14 \pm 0.12$ | $43.95 \pm 5.12$ | 48.6% | $71.92 \pm 0.33$ | $75.95 \pm 4.72$ | 26.2% |
| $\theta = 30°$ | $86.96 \pm 0.06$ | $46.82 \pm 1.21$ | 37.7% | $70.91 \pm 0.46$ | $70.97 \pm 4.88$ | 12.7% |
| $\theta = 45°$ | $86.94 \pm 0.26$ | $46.63 \pm 4.38$ | 4.2% | $70.24 \pm 0.08$ | $79.51 \pm 2.88$ | 0.2% |
| $\theta = 90°$ | $86.78 \pm 0.47$ | $48.91 \pm 3.62$ | 0% | $70.14 \pm 0.27$ | $79.43 \pm 1.45$ | 0% |

Table 3 shows our schedule of using the fair angle $\theta$ to control the gradient alignment times is effective, as it largely reduces the communication rounds with larger angles. In addition, compared with the results of baseline in Table 1, the results illustrate that our algorithm remains effective when we increase the fair angle. The communication cost of communicating the MLP model is 7.8MB/round, the CNN model is 30.4MB/round. If the communication cost is affordable, $\theta = 0$ should be chosen for optimal performance. Otherwise, we recommend using the Prac-FedEBA+ algorithm with the default $\theta = 15°$, which requires no additional communication cost but with better performance than SOTA baselines.

**Robustness and Privacy Evaluation.** Table 13 demonstrates that FedEBA+ keeps robust to noisy label scenarios; Figure 8 indicates that FedEBA+ is compatible with differential privacy methods without significant performance degradation. Additional details are provided in Appendix M.

**All the components of FedEBA+ are necessary.** In Table 15 of Appendix M, we conduct the ablation study on FedEBA+, showing that each step of FedEBA+ is beneficial. Even the aggregation alone improves global performance and fairness.

**Additional results in Appendix M consistently demonstrate the superiority of FedEBA+**, including: 1) Performance table with full hyperparameter choices for algorithms (Table 7 for baselines and Table 16 for FedEBA+). 2) Performance of fairness algorithms integrated with advanced optimization methods like momentum (Table 10) and VARP (Table 11). 3) Performance results under cosine similarity and entropy metrics (Table 19). 4) Ablation studies on the fair angle $\theta$, Dirichlet parameter (non-iid-ness), and annealing strategies of $\tau$, as detailed in Table 8, Figure 14, and Figure 9, respectively. 5) Scalability of FedEBA+ in Table 21 and 22.

## 7 Conclusions, Limitations and Future Works

In this paper, we introduced FedEBA+, a novel federated learning algorithm that enhances fairness and global model performance through a computationally efficient bi-level optimization framework. We propose an innovative entropy-based fair aggregation method for the inner loop and develop adaptive alignment strategies to optimize global performance and fairness in the outer loop. Our theoretical analysis confirms that FedEBA+ converges effectively in non-convex federated learning settings, and empirical results demonstrate its superiority over state-of-the-art fairness algorithms, ensuring consistent performance across diverse clients and improving overall global model accuracy.

While FedEBA+ exhibits resilience to noisy label scenarios, ensuring its efficacy in the face of backdoor or Byzantine attacks remains an open challenge. Malicious attackers may upload high losses to divert server's focus, thereby diminishing model performance. Developing a Byzantine-robust version of FedEBA+ is left for future investigation.

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

Contents of Appendix

# A An Expanded Version of The Related Work

**Fairness-Aware Federated Learning.** Various fairness concepts have been proposed in FL, including performance fairness (Li et al., 2019a; 2021; Wang et al., 2021; Zhao & Joshi, 2022; Kanaparthy et al., 2022; Huang et al., 2022), group fairness (Du et al., 2021; Ray Chaudhury et al., 2022), selection fairness (Zhou et al., 2021), and contribution fairness (Cong et al., 2020), among others (Shi et al., 2021; Wu et al., 2022; Chen et al., 2023). These concepts address specific aspects and stakeholder interests, making direct comparisons inappropriate. This paper specifically focuses on performance fairness, the most commonly used metric in FL, which serves client interests while improving model performance. We list and compare the commonly used fairness metrics of FL in the next section, i.e., Section B.

Some works propose objective function-based approaches to enhance performance fairness for FL. In (Li et al., 2019a), q-FFL uses $\alpha$-fair allocation for balancing fairness and efficiency, but specific $\alpha$ choices may introduce bias. In contrast, FedEBA+ employs maximum entropy aggregation to accommodate diverse preferences. Additionally, FedEBA+ introduces a novel fair FL objective with dual-variable optimization, enhancing global model performance and variance. Besides, Deng et al. (2020) achieves fairness by defining a min-max optimization problem in FL. In the gradient-based approach, FedFV (Wang et al., 2021) mitigates gradient conflicts among FL clients to promote fairness, but it consumes much computational and storage resources. Efforts have been made to connect fairness and personalized FL to enhance robustness (Li et al., 2021; Lin et al., 2022), different from our goal of learning a valid global model to guarantee fairness. FOCUS (Chu et al., 2023) introduces the *Fairness via Agent-Awareness* (FAA) metric, quantifying the maximum discrepancy in excess loss across agents. Utilizing an Expectation Maximization (EM) algorithm, FOCUS achieves soft clustering of clients. However, it involves communication between all clients and the server, with each client requiring all cluster models, resulting in elevated communication and computation costs. Although addressing FAA is not our primary focus, we illustrate that FedEBA+ remains effective and outperforms FOCUS in both variance and FAA in our experimental setting, as detailed in Table 1 and Table 17. Notably, our method operates without imposing data distribution or model class assumptions, distinguishing it from existing work (Chu et al., 2023) that relies on the distance disparity of local loss and ideal loss as a fairness measure. The use of variance in performance fairness naturally aligns with the goal of ensuring uniform performance across clients. Recently, reweighting methods encourage a uniform performance by up-reweighting the importance of underperforming clients (Zhao & Joshi, 2022; Mollanejad et al., 2024). However, these methods enhance fairness at the expense of the performance of the global model (Kanaparthy et al., 2022; Huang et al., 2022). In contrast, we propose FedEBA+ as a solution that significantly promotes fairness while improving the global model performance. Notably, FedEBA+ is orthogonal to existing optimization methods like momentum (Karimireddy et al., 2020a) and VARP (Jhunjhunwala et al., 2022), allowing seamless integration, as shown in Table 10 and Table 11.

Recently, several federated learning studies have explored a diverse range of fairness objectives, such as Proportionality (Chaudhury et al., 2024; Ray Chaudhury et al., 2022), Disparity (Hamman & Dutta), Stability (Gao et al.), and fairness in vertical FL (Fan et al.; Qi et al., 2022). Chaudhury et al. (2024) provides explainable proportional fairness guarantees to the agents in general settings in which the error rates of the agents are proportional to the size of their local data, and Ray Chaudhury et al. (2022) proposes a core-stability as fairness metric that is more resilient to noisy data from certain clients. The used fairness is sensitive to data, while ours focuses on performance fairness for clients, regarding the data distribution, thus the objective is different. Hamman & Dutta offers an information-theoretic perspective on group fairness trade-offs in federated learning, utilizing partial information decomposition to identify unfairness. Gao et al. mainly focus on establishing a theoretical bound for showing the influence of clients' altruistic behaviors and the configuration of the friend-relationship network on the achievable egalitarian fairness. These works aim to establish the theoretical bound for analyzing the fairness and trade-offs, from an information

perspective and game theory, instead of providing a fair algorithm. Fan et al.; Qi et al. (2022) discuss fairness in vertical FL by learning fair and unified representations, where feature fields are decentralized across different platforms. In contrast, our work focuses on horizontal FL and compares our results with state-of-the-art horizontal FL fairness algorithms.

**Aggregation in Federated Optimization.** FL employs aggregation algorithms to combine decentralized data for training a global model (Kairouz et al., 2019). Approaches include federated averaging (FedAvg) McMahan et al. (2017), robust federated weighted averaging Pillutla et al. (2019); Laguel et al. (2021); Pillutla et al. (2023), importance aggregation Wang et al. (2022), and federated dropout Zheng et al. (2022). However, these algorithms can be sensitive to the number and quality of participating clients, causing fairness issues (Li et al., 2019b; Balakrishnan et al., 2021; Shi et al., 2021). To the best of our knowledge, we are the first to analyze the aggregation from the view of entropy. Unlike heuristics that assign weights proportional to client loss (Zhao & Joshi, 2022; Kanaparthy et al., 2022), our method has physical meanings, i.e., the aggregation probability ensures that known constraints are as certain as possible while retaining maximum uncertainty for unknowns. By selecting the maximum entropy solution with constraints, we actually choose the solution that fits our information with the least deviation (Jaynes, 1957), thus achieving fairness.

Our proposed aggregation method differs from existing approaches in several key aspects. First, the aggregation formulation is novel, with probabilities $p_i = e^{\frac{F_i(x)/\tau}{Z}}$ proportional to the exponential of client loss and regulated by a controllable parameter $\tau$. Unlike heuristic methods that assign weights directly proportional to client loss $p_i \propto F_i(x)$ (Mollanejad et al., 2024; Zhao & Joshi, 2022; Kanaparthy et al., 2022), our approach is derived from a constrained optimization framework. Second, the objective is fundamentally different. Existing entropy-based aggregation methods (Huang et al., 2022; Herath et al., 2024) and softmax-based reweighting approaches (Zhao & Joshi, 2022; Kanaparthy et al., 2022) aim to enhance model accuracy without addressing fairness, whereas our approach focuses explicitly on improving fairness. Third, our method introduces a novel constrained entropy model, the first of its kind in the FL fairness community, which prioritizes underperforming clients to achieve weighted fair aggregation. Furthermore, our approach offers practical advantages, such as its exponent form and control parameter $\tau$, which effectively mitigate extreme unfairness and allow flexibility in recovering existing aggregation methods like FedAvg, AFL, and q-FFL. Empirically, our entropy-based aggregation (FedEBA+ with $\alpha = 0$ ) outperforms state-of-the-art methods like q-FFL and TERM, achieving superior results in both fairness and accuracy.

**FL others.** In addition to fairness algorithms, FL faces other challenges such as privacy preservation (Wang et al., 2023; Zhou et al., 2023; Chen et al., 2023) and communication efficiency (Chai et al., 2023; Almanifi et al., 2023; Paragliola & Coronato, 2022). Given the widespread adoption of FL, our primary focus in this work is on designing a high-performance fairness algorithm. Nonetheless, we acknowledge the significance of other aspects in FL, such as privacy preservation. Hence, we provide experimental results demonstrating the compatibility of our algorithm with existing privacy protection methods and its robustness to external noise scenarios.

## B    DISCUSSION OF FAIRNESS METRICS

In this section, we summarize the commonly used definitions of fairness metrics and comment on their advantages and disadvantages.

Euclidean Distance and person correlation coefficient are usually used for contribution fairness, and risk difference and Jain's fairness Index are usually used for group fairness, which is a different target from performance fairness in this paper. In particular, cosine similarity and entropy play roles similar to variance, used to measure the performance distribution among clients. The more uniform the distribution, the smaller the variance and the more similar to vector 1. The larger the entropy of the normalized performance, the more similar to vector 1. Thus, for performance fairness, we only need one of them. We use variance, which is the most widely used metric in related works.

The detailed discussion of each metric is shown below:

- **Variance**, applied in accuracy parity and performance fairness scenarios, is valued for its simplicity and straightforward implementation, focusing on a common performance metric. However, it has a limitation as it only measures relative fairness, making it sensitive to outliers (Zafar et al., 2017; Li et al., 2019a; 2021; Hu et al., 2022; Shi et al., 2021).

- **Cosine similarity**, sharing applications with variance, is known for its similarity to variance and the ease with which it captures linear relationships (Li et al., 2019a). Nevertheless, it falls short when it comes to capturing magnitude differences and is sensitive to zero vectors (Selbst et al., 2019; Hardt et al., 2016).

- Also utilized in scenarios akin to variance, **entropy** offers simplicity but has dependencies on normalization and sensitivity to the number of clients involved in the computation, making it less robust in certain situations (Li et al., 2019a; Selbst et al., 2019; Hardt et al., 2016).

- Applied in contribution fairness, **Euclidean distance** provides a straightforward interpretation and is sensitive to magnitude differences. However, it lacks consideration for the direction of the differences, limiting its overall effectiveness.

- In contribution fairness scenarios, the **Pearson correlation coefficient** is appreciated for its scale invariance and ability to capture linear relationships (Jia et al., 2019). Yet, it may be sensitive to outliers and may not accurately capture magnitude differences, assuming a linear relationship between the data variables (Wang et al., 2019).

- Commonly used in group fairness contexts, **risk difference** is sensitive to group disparities and offers interpretability (Du et al., 2021). However, it lacks normalization, which can impact its effectiveness in certain scenarios (Dwork et al., 2012).

- **Jain's Fairness Index** finds application in various fairness aspects, including group fairness, selection fairness, performance fairness, and contribution fairness. It boasts normalization across groups and flexibility in handling various metrics. Nevertheless, it is sensitive to metric choice and introduces complexity in interpretability (Chiu, 1984; Liu et al., 2022).

# C  Entropy Analysis

## C.1  Derivation of Proposition 4.1

In this section, we derive the maximum entropy distribution for the aggregation strategy employed in FedEBA+.

The choice of an exponential formula treatment for the loss function, represented as $p_i \propto e^{F_i(x)/\tau}$, is motivated by our adherence to a maximum entropy distribution. This approach is favored over alternatives such as $p_i \propto F_i(x)$ because our aggregation strategy is designed to achieve maximum entropy.

Maximizing entropy minimizes the incorporation of prior information into the distribution, ensuring that the selected probability distribution is free from subjective influences and biases (Bian et al., 2021; Sampat & Zavala, 2019). Simultaneously, this aligns with the tendency of many physical systems to evolve towards configurations with maximal entropy over time (Jaynes, 1957).

In the following we will give a derivation to show that $p_i \propto e^{F_i(x_i)/\tau}$ is indeed the maximum entropy distribution for FL. The derivation below is closely following (Jaynes, 1957) for statistical mechanics. Suppose the loss function of the user corresponding to the aggregation probability $p_i$ is $F_i(x_i)$. We would like to maximize the entropy $\mathbb{H}(p_i) = -\sum_{i=1}^{m} p_i \log p_i$, subject to FL constrains that $\sum_{i=1}^{m} p_i = 1, p_i \geq 0$, $\sum_i p_i F_i(x_i) = \tilde{f}(x)$, which means we constrain the reweighted clients' performance to be close to ideal model's performance, such as ideal global model performance or the ideal fair performance.

*Proof.*

$$L\left(p, \lambda_0; \frac{1}{\tau}\right) := -\left[\sum_{i=1}^{N} p_i \log p_i + \lambda_0 \left(\sum_{i=1}^{N} p_i - 1\right) + \frac{1}{\tau}\left(\mu - \sum_{i=1}^{N} p_i F_i(x_i)\right)\right], \quad (18)$$

where $\mu = \tilde{f}(x)$.

By setting

$$\frac{\partial L\left(p, \lambda_0; \frac{1}{\tau}\right)}{\partial p_i} = -\left[\log p_i + 1 + \lambda_0 - \frac{1}{\tau}F_i(x_i)\right] = 0, \quad (19)$$

we get:

$$p_i = \exp\left[-\left(\lambda_0 + 1 - \frac{1}{\tau}F_i(x_i)\right)\right]. \quad (20)$$

According to $\sum_i p_i = 1$, we have:

$$\lambda_0 + 1 = \log \sum_{i=1}^{N} \exp\left(\frac{1}{\tau}F_i(x_i)\right) =: \log Z, \quad (21)$$

which is the log-partition function.

Thus, we reach the exponential form of $p_i$ as:

$$p_i = \frac{\exp[F_i(x_i)/\tau]}{\sum_{j=1}^{N} \exp(F_j(x_j)/\tau)}. \quad (22)$$

$\square$

When taking into account the prior distribution of aggregation probability (Li et al., 2020b; Balakrishnan et al., 2021), which is typically expressed as $q_i = n_i / \sum_{i \in S_t} n_i$, the original entropy formula can be extended to include the prior distribution as follows:

$$H(p_i) = \sum_{i=1}^{m} p_i \log(\frac{q_i}{p_i}). \quad (23)$$

Thus, the solution of the original problem under this prior distribution becomes:

$$p_i = \frac{q_i \exp[F_i(x_i)/\tau]}{\sum_{j=1}^{N} q_j \exp[F_j(x_i)/\tau]}. \quad (24)$$

*Proof.*

$$L\left(p, \lambda_0; \frac{1}{\tau}\right) := -\sum_{i=1}^{N} p_i \log \frac{q_i}{p_i} + \lambda_0 \left(\sum_{i=1}^{N} p_i - 1\right) + \frac{1}{\tau}\left(\mu - \sum_{i=1}^{N} p_i F_i(x_i)\right). \quad (25)$$

Following similar derivation steps, let

$$\frac{\partial L\left(p, \lambda_0; \frac{1}{\tau}\right)}{\partial p_i} = -\log(q_i) + \log(p_i) + 1 + \lambda_0 - \frac{1}{\tau}F_i(x_i) = 0, \quad (26)$$

we get:

$$p_i = \exp\left[-\left(\lambda_0 + 1 - \log(q_i) - \frac{1}{\tau}F_i(x_i)\right)\right]. \quad (27)$$

According to $\sum_i p_i = 1$, we have:

$$\sum_i p_i = \sum_i \exp\left[-\left(\lambda_0 + 1 - \log(q_i) - \frac{1}{\tau}F_i(x_i)\right)\right] = 1. \quad (28)$$

Therefore, we get:

$$\lambda_0 + 1 = \log \sum_{i=1}^{N} q_i \exp\left(\frac{1}{\tau} F_i(x)\right) =: \log(Z). \tag{29}$$

Then substituting $\lambda_0 + 1 = \log(Z)$ back to $p_i = \exp\left[-\left(\lambda_0 + 1 - \log(q_i) - \frac{1}{\tau} F_i(x_i)\right)\right]$, we obtain (24):

$$p_i = \frac{q_i \exp[F_i(x_i)/\tau]}{\sum_{j=1}^{N} q_j \exp[F_j(x_i)/\tau]}. \tag{30}$$

$\square$

## D  ENHANCING ROBUSTNESS IN FEDEBA+ THROUGH LOCAL SELF-REGULARIZATION

In this section, we introduce Local Self-Regularization (LSR) for FedEBA+ as a robustness solver. The method is primarily based on the work of Jiang et al. (2022). For the sake of completeness in this paper, we restate the LSR algorithm here. The LSR algorithm effectively regulates the local training process by implicitly preventing the model from memorizing noisy labels. Additionally, it explicitly narrows the model output discrepancy between original and augmented instances through self-distillation.

---

**Algorithm 2** Local Self-Regularization

---

1: **for** client $i$ in parallel **do**
2:     **Input:** client $i$, global model $x_t$, parameter $\gamma$, $\lambda \sim Beta(1,1)$.
3:     **Output:** local trained model $x_i^{t+1}$.
4:     **Initialize:** $x_i^{t,0} \leftarrow x_t$.
5:     **for** $k = 0, \cdots, K-1$ **do**
6:         $p_1, p_2 = Softmax(F_i(x_i^{t,k}; \xi_i)), Softmax(F(x_i^{t,k}; Augment(\xi_i)));$
7:         $p = \lambda p_1 + (1-\lambda)p_2;$
8:         $p_{s,c} = \frac{p_c^{1/T_s}}{\sum_j p_j^{1/T_s}}$, where $c$ denotes the $c$-th class, and $T_s$ is the sharpening temperature;
9:         $F^{cls} = CorssEntropy(p_s, y);$
10:        $F^{reg} = SelfDistillation(F(x_i^{t,k}; \xi_i), F(x_i^{t,k}; Augment(\xi_i)));$
11:        $F_i^r = F^{cls} + \gamma F^{reg};$
12:        Update $x_{t+1}^i$ with $F_i^r;$
13:     **end for**
14: **end for**

---

For the regression loss, self-distillation is performed on the network. We use the two output logits $\xi_i$ and $Augment(\xi_i)$ to conduct instance-level self-distillation. First, apply a softmax function with a distillation parameter $T_d$ to the output as:

$$q_{1,i}, q_{2,i} = \frac{\exp([F(x_i^{t,k}; \xi_i)]_c/T_d)}{\sum_j \exp([F(x_i^{t,k}; \xi_i)]_j/T_d)}, \frac{\exp([F(x_i^{t,k}; Augment(\xi_i))]_c/T_d)}{\sum_j \exp([F(x_i^{t,k}; Augment(\xi_i))]_j/T_d)}, \tag{31}$$

where $c$ and $j$ denote the output logits for the $c$-th and $j$-th class, respectively. The self-distillation loss term is formulated as:

$$F^{reg} = \frac{1}{2}(\text{KL}(q_1 \| U) + \frac{1}{2}(\text{KL}(q_2 \| U)), \tag{32}$$

where KL means Kullback-Leibler divergence and $U = \frac{1}{2}(q_1 + q_2)$.

In this way, we can express the *robust EBA* method by:

$$p_i = \frac{\exp(F_i^r(x)/\tau)}{\sum_j \exp(F_j^r(x)/\tau)}, \qquad F_i^r(x) = \mathbb{E}_{\xi_i}\left[F_i^{cls}(x; \xi_i) + \gamma F_i^{reg}(x; Augment(\xi_i))\right]. \tag{33}$$

We experimentally demonstrate the robustness of EBA in Table 13.

---

**Algorithm 3** Prac-FedEBA+

---

1: **Input:** Number of clients $m$, global learning rate $\eta$, local learning rate $\eta_l$, number of local epoch $K$, total training rounds $T$, threshold $\theta$.
2: **Output:** Final model parameter $x_T$.
3: **Initialize:** model $x_0$, guidance vector $\mathbf{r} = [1, \cdots, 1]$.
4: **for** round $t = 1, \ldots, T$ **do**
5:    Server selects a set of clients $|S_t|$ and broadcast model $x_t$.
6:    **for** each worker $i \in S_t$,in parallel **do**
7:      **for** $k = 0, \cdots, K-1$ **do**
8:        $x_{t,k+1}^i = x_{t,k}^i - \eta_L \nabla F_i(x_{t,k}^i; \xi_i)$;
9:      **end for**
10:     $\Delta_t^i = x_{t,K}^i - x_{t,0}^i = -\eta_L \sum_{k=0}^{K-1} \nabla F_i(x_{t,k}^i; \xi_i)$;
11:    **end for**
12:    Server receive model updates $\Delta_t^i$ and clients' loss $\mathbf{L} = [F_1(x_t), \ldots, F_{|S_t|}(x_t)]$;
13:    **if** $arccos(\frac{\mathbf{L},\mathbf{r}}{\|\mathbf{L}\| \cdot \|\mathbf{r}\|}) > \theta$ **then**
14:      Approximate fair gradient: $\tilde{g}^t = \sum_{i \in S_t} \frac{\exp[F_i(x_t)/\tau)]}{\sum_{i \in S_t} \exp[F_i(x_t)/\tau]} \frac{1}{K} \sum_{k=0}^{K-1} \nabla F_i(x_{t,k}^i; \xi_i)$;
15:      Align model: $\hat{\Delta}_i^t = (1-\alpha)\Delta_i^t - \alpha \eta_L K \tilde{g}^t$;
16:      Aggregation: $\Delta_t = \sum_{i \in S_t} p_i \hat{\Delta}_t^i$, where $p_i = \frac{\exp[F_i(x_{t,K}^i)/\tau)]}{\sum_{i \in S_t} \exp[F_i(x_{t,K}^i)/\tau]}$ ;
17:    **else**
18:      Approximate global update for participating client: $\tilde{\Delta}_t^i = \frac{1}{K}(x_{t,K-1}^i - x_{t,0}^i)$;
19:      Server aggregates model update by (8);
20:    **end if**
21:    Server update: $x_{t+1} = x_t + \eta \Delta_t$;
22: **end for**

---

## D.1   Toy example of extremal case

In this subsection, we examine an extreme case as an illustrative example. Consider two clients: client 1 with noisy data and client 2 with separable data. Assume the test accuracy on client 1 is consistently zero or the loss is always high, denoted as $H_1$.

After local updates on each client, the model adjusts its parameters to minimize the noise. However, in the absence of an underlying pattern, the weights do not capture any meaningful relationship between features and labels. Consequently, the loss can be assumed to be $H_1$, and the model parameter as $x_1^t = x_i^{t+1}$ without loss of generality, as the model has no convergence point.

In contrast, assume client 2's model is $y = \frac{1}{2}x^2$, and starting from $x_2^t = 2$, it converges to $x_2^{t+1} = 0$. Thus, for FedEBA+, the updated model is $\tilde{x} = 0 + x_1^t \cdot e^{\frac{H_1}{H_1+0}}$. For FedAvg, the updated model is $\hat{x} = \frac{1}{2}x_1^t$. Since $|e \cdot x_1| \geq |\frac{1}{2}x_1|$, we have $y(\tilde{x}) \leq y(\hat{x})$. Consequently, we can assert that the disparity between client 1 and client 2 using EBA+ is smaller than with FedAvg.

Hence, we assert that even in the extreme case, FedEBA+ effectively reduces performance variance through the entropy-based aggregation method.

# E   Practical Algorithm with effective communication.

To achieve the same communication costs to FedAvg, we introduce a practical adaptation of FedEBA+ termed Prac-FedEBA+. Specifically, Prac-FedEBA+ leverages the last round's gradient to approximate current round information, reducing the need for extensive communication between the server and clients, as outlined in Algorithm 3.

Table 4: Convergence rate comparison of FedEBA+ with existing works.

| Algorithm | Convergence Upper Bound | Rate Order |
|---|---|---|
| FedAvg (Yang et al., 2021) | $\frac{1}{c}\left(\frac{f^0-f^*}{\sqrt{nKT}} + \frac{\sigma_L^2+3K\sigma_G^2}{2\sqrt{nKT}} + \frac{5(\sigma_L^2+6K\sigma_G^2)^2}{2KT} + \frac{15(\sigma_L^2+6K\sigma_G^2)}{2\sqrt{nKT^3}}\right)$ | $\mathcal{O}(\frac{1}{\sqrt{nKT}} + \frac{1}{T} + \frac{1}{\sqrt{nKT^3}})$ |
| FedIS (Chen et al., 2020) | $\frac{1}{c}\left(\frac{(f^0-f^*)B^2}{\sqrt{nKT}} + \frac{2F\sigma_L^2+2F(1-n/m)K\sigma_G^2}{2\sqrt{nKT}} + \frac{B^2F}{T} + \frac{F^{2/3}\sigma_G}{T^{2/3}}\right)$ | $\mathcal{O}(\frac{1}{\sqrt{nKT}} + \frac{1}{T} + \frac{1}{\sqrt{T^3}})$ |
| FedNova (Wang et al., 2020) | $\frac{1}{c}\left(\frac{(f^0-f^*)}{\sqrt{nKT}} + \frac{A\sigma_L^2+\overline{\tau}/\tau_{eff}}{2\sqrt{nKT}} + \frac{mC\sigma_G^2}{\overline{\tau}T}\right)$ | $\mathcal{O}(\frac{1}{\sqrt{nKT}} + \frac{1}{T})$ |
| FedEBA+ | $\frac{1}{c}\left(\frac{f^0-f^*}{\sqrt{nKT}} + \frac{(1-\alpha)^2\sum_{i=1}^m w_i^2\sqrt{m}\sigma_L^2 + \alpha^2 K^{-1/2}\sqrt{m}\rho^2}{2\sqrt{nKT}} + \frac{5(1-\alpha)^2(\sigma_L^2+6K\sigma_G^2)+15(1-\alpha)^2\alpha^2 K\rho^2}{2KT}\right)$ | $\mathcal{O}(\frac{\sqrt{K/n}}{\sqrt{nKT}} + \frac{1}{T})$ |

## F  ANALYSIS COMPARISON WITH EXISTING WORKS

In this paper, the fairness and global model performance are analyzed via variance and convergence, respectively. The comprehensive analysis significantly improves upon existing research.

- For the variance analysis, all existing fairness works are typically evaluated by comparing them with FedAvg. However, our analysis expands beyond linear models to include the strongly convex setting.

- For the convergence analysis, beyond the strongly convex and convex settings, we demonstrate that our algorithms converge in nonconvex settings with a convergence rate no worse than the state-of-the-art FedAvg algorithm, as shown in the Table 4.

To explicitly demonstrate the importance of the paper's theoretical merit, we provide the following table to illustrate its contributions compared with other fairness works:

Table 5: Analysis Comparison of Different Fairness Algorithms

| Algorithm | Variance analysis | Convergence analysis |
|---|---|---|
| q-FFL | ✓ | × |
| FedMGDA+ | × | ✓ Strongly convex |
| TERM | ✓ Linear model | ✓ Strongly convex |
| AFL | × | ✓ Convex |
| PropFair | × | ✓ Nonconvex |
| lp-proj | ✓ Linear model | ✓ Nonconvex |
| FedEBA+ | ✓ Linear model & Strongly convex | ✓ Nonconvex |

The above comparison reveals that, among existing work, only FedEBA+ and lp-proj offer simultaneous variance and convergence analysis. In contrast to lp-proj:

- FedEBA+ expands fairness analysis from generalized linear regression models to strongly convex models.

- Moreover, lp-proj is a personalized FL algorithm, markedly distinct from ours, as this paper focuses on achieving a fair global model. Consequently, the convergence analysis and fairness analysis are distinct. Only FedEBA+ aims to improve the global model's performance and variance simultaneously, employing variance and convergence analyses, respectively.

# G   ASSUMPTIONS FOR CONVERGENCE ANALYSIS

To facilitate the convergence analysis, we adopt the following commonly used assumptions in FL.

**Assumption 1** (L-Smooth). *There exists a constant $L > 0$, such that $\|\nabla F_i(x) - \nabla F_i(y)\| \leq L\|x - y\|, \forall x, y \in \mathbb{R}^d$, and $i = 1, 2, \ldots, m$.*

**Assumption 2** (Unbiased Local Gradient Estimator and Local Variance). *Let $\xi_t^i$ be a random local data sample in the round $t$ at client $i$: $\mathbb{E}\left[\nabla F_i(x_t, \xi_t^i)\right] = \nabla F_i(x_t), \forall i \in [m]$. There exists a constant bound $\sigma_L > 0$, satisfying $\mathbb{E}\|\nabla F_i(x_t, \xi_t^i) - \nabla F_i(x_t)\|^2 \leq \sigma_L^2$.*

**Assumption 3** (Bound Gradient Dissimilarity). *For any set of weights $\{w_i \geq 0\}_{i=1}^m$ with $\sum_{i=1}^m w_i = 1$, there exist constants $\sigma_G^2 \geq 0$ and $A \geq 0$ such that $\sum_{i=1}^m w_i \|\nabla F_i(x)\|^2 \leq (A^2 + 1) \|\sum_{i=1}^m w_i \nabla F_i(x)\|^2 + \sigma_G^2$.*

These assumptions are commonly used in both non-convex optimization and FL literature, see e.g. (Karimireddy et al., 2020b; Yang et al., 2021; Wang et al., 2020). For Assumption 3, if all local loss functions are identical, then $A = 0$ and $\sigma_G = 0$.

# H   CONVERGENCE ANALYSIS OF FEDEBA+

In this section, we give the proof of Theorem 5.1.

Before going to the details of our convergence analysis, we first state the key lemmas used in our proof, which helps us to obtain the advanced convergence result.

**Lemma H.1.** *To make this paper self-contained, we restate the Lemma 3 in (Wang et al., 2020):*

*For any model parameter $\boldsymbol{x}$, the difference between the gradients of $f_{avg}(\boldsymbol{x})$ and $f(\boldsymbol{x})$ can be bounded as follows:*

$$\|\nabla f_{avg}(\boldsymbol{x}) - \nabla f(\boldsymbol{x})\|^2 \leq \chi_{\boldsymbol{w}\|\boldsymbol{p}}^2 \left[A^2 \|\nabla f(\boldsymbol{x})\|^2 + \chi_{\boldsymbol{w}\|\boldsymbol{p}}^2\right], \tag{34}$$

*where $\chi_{\boldsymbol{w}\|\boldsymbol{p}}^2$ denotes the chi-square distance between $\boldsymbol{w}$ and $\boldsymbol{p}$, i.e., $\chi_{\boldsymbol{w}\|\boldsymbol{p}}^2 = \sum_{i=1}^m (w_i - p_i)^2 / p_i$. $f(x)$ is the global objective with $f(x) = \sum_{i=1}^m w_i f_i(x)$ where $\boldsymbol{w}$ is usually the data ratio of clients, i.e., $\boldsymbol{w} = [\frac{n_i}{N}, \cdots, \frac{n_i}{N}]$. $f(x) = \sum_{i=1}^m p_i f_i(x)$ is the objective function of FedEBA+ with the reweight aggregation probability $\boldsymbol{p}$.*

*Proof.*

$$\nabla f_{avg}(x) - \nabla f(\boldsymbol{x}) = \sum_{i=1}^m (w_i - p_i) \nabla f_i^{avg}(\boldsymbol{x})$$

$$= \sum_{i=1}^m (w_i - p_i) (\nabla f_i^{avg}(\boldsymbol{x}) - \nabla f(\boldsymbol{x})) \tag{35}$$

$$= \sum_{i=1}^m \frac{w_i - p_i}{\sqrt{p_i}} \cdot \sqrt{p_i} (\nabla f_i^{avg}(\boldsymbol{x}) - \nabla f(\boldsymbol{x})) .$$

Applying Cauchy-Schwarz inequality, it follows that

$$\|\nabla f_{avg}(x) - \nabla f(\boldsymbol{x})\|^2 \leq \left[\sum_{i=1}^m \frac{(w_i - p_i)^2}{p_i}\right] \left[\sum_{i=1}^m p_i \|\nabla f_i^{avg}(x) - \nabla f(\boldsymbol{x})\|^2\right]$$

$$\leq \chi_{\boldsymbol{w}\|\boldsymbol{p}}^2 \left[A^2 \|\nabla f(\boldsymbol{x})\|^2 + \sigma_G^2\right], \tag{36}$$

where the last inequality uses Assumption 3. Note that

$$\|\nabla f_{avg}(\boldsymbol{x})\|^2 \leq 2\|\nabla f_{avg}(\boldsymbol{x}) - \nabla f(\boldsymbol{x})\|^2 + 2\|\nabla f(\boldsymbol{x})\|^2$$

$$\leq 2\left[\chi_{\boldsymbol{w}\|\boldsymbol{p}}^2 A^2 + 1\right] \|\nabla f(\boldsymbol{x})\|^2 + 2\chi_{\boldsymbol{p}\|\boldsymbol{w}}^2 \sigma_G^2 . \tag{37}$$

As a result, we obtain

$$\min_{t\in[T]} \|\nabla f_{avg}(\boldsymbol{x}_t)\|^2 \leq \frac{1}{T}\sum_{t=0}^{T-1}\|\nabla f_{avg}(\boldsymbol{x}_t)\|^2 \tag{38}$$

$$\leq 2\left[\chi^2_{\boldsymbol{w}\|\boldsymbol{p}}A^2+1\right]\frac{1}{T}\sum_{t=0}^{T-1}\|\nabla f(\boldsymbol{x}_t)\|^2 + 2\chi^2_{\boldsymbol{w}\|\boldsymbol{p}}\sigma_G^2 \tag{39}$$

$$\leq 2\left[\chi^2_{\boldsymbol{w}\|\boldsymbol{p}}A^2+1\right]\epsilon_{\text{opt}} + 2\chi^2_{\boldsymbol{w}\|\boldsymbol{p}}\sigma_G^2\,, \tag{40}$$

where $\epsilon_{\text{opt}} = \frac{1}{T}\sum_{t=0}^{T-1}\|\nabla f(\boldsymbol{x}_t)\|^2$ denotes the optimization error.

$\square$

## H.1 ANALYSIS WITH $\alpha=0$.

**Lemma H.2** (Local updates bound.). *For any step-size satisfying $\eta_L \leq \frac{1}{8LK}$, we can have the following results:*

$$\mathbb{E}\|x_{t,k}^i - x_t\|^2 \leq 5K(\eta_L^2\sigma_L^2 + 4K\eta_L^2\sigma_G^2) + 20K^2(A^2+1)\eta_L^2\|\nabla f(x_t)\|^2\,. \tag{41}$$

*Proof.*

$$\mathbb{E}_t\|x_{t,k}^i - x_t\|^2 \tag{42}$$

$$= \mathbb{E}_t\|x_{t,k-1}^i - x_t - \eta_L g_{t,k-1}^t\|^2 \tag{43}$$

$$= \mathbb{E}_t\|x_{t,k-1}^i - x_t - \eta_L(g_{t,k-1}^t - \nabla F_i(x_{t,k-1}^i) + \nabla F_i(x_{t,k-1}^i) - \nabla F_i(x_t) + \nabla F_i(x_t))\|^2 \tag{44}$$

$$\leq (1+\frac{1}{2K-1})\mathbb{E}_t\|x_{t,k-1}^i - x_t\|^2 + \mathbb{E}_t\|\eta_L(g_{t,k-1}^t - \nabla F_i(x_{t,k}^i))\|^2$$
$$+ 4K\mathbb{E}_t[\|\eta_L(\nabla F_i(x_{t,K-1}^i) - \nabla F_i(x_t))\|^2] + 4K\eta_L^2\mathbb{E}_t\|\nabla F_i(x_t)\|^2 \tag{45}$$

$$\leq (1+\frac{1}{2K-1})\mathbb{E}_t\|x_{t,k-1}^i - x_t\|^2 + \eta_L^2\sigma_L^2 + 4K\eta_L^2 L^2\mathbb{E}_t\|x_{t,k-1}^i - x_t\|^2$$
$$+ 4K\eta_L^2\sigma_G^2 + 4K\eta_L^2(A^2+1)\|\nabla f(x_t)\|^2 \tag{46}$$

$$\leq (1+\frac{1}{K-1})\mathbb{E}\|x_{t,k-1}^i - x_t\|^2 + \eta_L^2\sigma_L^2 + 4K\eta_L^2\sigma_G^2 + 4K(A^2+1)\|\eta_L\nabla f(x_t)\|^2\,. \tag{47}$$

Unrolling the recursion, we obtain:

$$\mathbb{E}_t\|x_{t,k}^i - x_t\|^2 \tag{48}$$

$$\leq \sum_{p=0}^{k-1}(1+\frac{1}{K-1})^p\left[\eta_L^2\sigma_L^2 + 4K\eta_L^2\sigma_G^2 + 4K(A^2+1)\|\eta_L\nabla f(x_t)\|^2\right] \tag{49}$$

$$\leq (K-1)\left[(1+\frac{1}{K-1})^K - 1\right]\left[\eta_L^2\sigma_L^2 + 4K\eta_L^2\sigma_G^2 + 4K(A^2+1)\|\eta_L\nabla f(x_t)\|^2\right] \tag{50}$$

$$\leq 5K(\eta_L^2\sigma_L^2 + 4K\eta_L^2\sigma_G^2) + 20K^2(A^2+1)\eta_L^2\|\nabla f(x_t)\|^2\,. \tag{51}$$

$\square$

Thus, we can have the following convergence rate of FedEBA+:

**Theorem H.3.** *Under Assumption 1–3, and let constant local and global learning rate $\eta_L$ and $\eta$ be chosen such that $\eta_L < \min(1/(8LK), C)$, where $C$ is obtained from the condition that $\frac{1}{2} - 10L^2\frac{1}{m}\sum_{i-1}^m K^2\eta_L^2(A^2+1)(\chi^2_{\boldsymbol{w}\|\boldsymbol{p}}A^2+1) > c > 0$ ,and $\eta \leq 1/(\eta_L L)$, the expected gradient norm of FedEBA+ with $\alpha=0$, i.e., only using aggregation strategy 4, is bounded as follows:*

$$\min_{t\in[T]}\mathbb{E}\|\nabla f(x_t)\|^2 \leq \frac{f_0 - f_*}{c\eta\eta_L KT} + \Phi\,, \tag{52}$$

*where*

$$\Phi = \frac{1}{c}\big[\frac{5\eta_L^2 K L^2}{2}(\sigma_L^2 + 4K\sigma_G^2) + \frac{\eta\eta_L L}{2}\sigma_L^2 + 20L^2 K^2(A^2+1)\eta_L^2\chi_{\boldsymbol{w}\|\boldsymbol{p}}^2\sigma_G^2\big]. \tag{53}$$

*where $c$ is a constant, $\chi_{\boldsymbol{w}\|\boldsymbol{p}}^2 = \sum_{i=1}^m (w_i - p_i)^2 /p_i$ represents the chi-square divergence between vectors $\boldsymbol{p} = [p_1, \ldots, p_m]$ and $\boldsymbol{w} = [w_1, \ldots, w_m]$. For common FL algorithms with uniform aggregation or with data ratio as aggregation probability, $w_i = \frac{1}{m}$ or $w_i = \frac{n_i}{N}$.*

*Proof.* Based on Lemma H.1, we first focus on analyzing the optimization error $\epsilon_{opt}$:

$$\mathbb{E}_t[f(x_{t+1})] \tag{54}$$

$$\overset{(a1)}{\leq} f(x_t) + \langle \nabla f(x_t), \mathbb{E}_t[x_{t+1} - x_t]\rangle + \frac{L}{2}\mathbb{E}_t[\|x_{t+1} - x_t\|^2] \tag{55}$$

$$= f(x_t) + \langle \nabla f(x_t), \mathbb{E}_t[\eta\Delta_t + \eta\eta_L K\nabla f(x_t) - \eta\eta_L K\nabla f(x_t)]\rangle + \frac{L}{2}\eta^2\mathbb{E}_t[\|\Delta_t\|^2] \tag{56}$$

$$= f(x_t) - \eta\eta_L K\|\nabla f(x_t)\|^2 + \eta\underbrace{\langle \nabla f(x_t), \mathbb{E}_t[\Delta_t + \eta_L K\nabla f(x_t)]\rangle}_{A_1} + \frac{L}{2}\eta^2\underbrace{\mathbb{E}_t\|\Delta_t\|^2}_{A_2}, \tag{57}$$

where (a1) follows from the Lipschitz continuity condition. Here, the expectation is over the local data SGD and the filtration of $x_t$. However, in the next analysis, the expectation is over all randomness, including client sampling. This is achieved by taking expectation on both sides of the above equation over client sampling.

To begin with, we consider $A_1$:

$$A_1 \tag{58}$$

$$= \langle \nabla f(x_t), \mathbb{E}_t[\Delta_t + \eta_L K\nabla f(x_t)]\rangle \tag{59}$$

$$= \left\langle \nabla f(x_t), \mathbb{E}_t[-\sum_{i=1}^m w_i \sum_{k=0}^{K-1} \eta_L g_{t,k}^i + \eta_L K\nabla f(x_t)]\right\rangle \tag{60}$$

$$\overset{(a2)}{=} \left\langle \nabla f(x_t), \mathbb{E}_t[-\sum_{i=1}^m w_i \sum_{k=0}^{K-1} \eta_L \nabla F_i(x_{t,k}^i) + \eta_L K\nabla f(x_t)]\right\rangle \tag{61}$$

$$= \left\langle \sqrt{\eta_L K}\nabla f(x_t), -\frac{\sqrt{\eta_L}}{\sqrt{K}}\mathbb{E}_t[\sum_{i=1}^m w_i \sum_{k=0}^{K-1} (\nabla F_i(x_{t,k}^i) - \nabla F_i(x_t))]\right\rangle \tag{62}$$

$$\overset{(a3)}{=} \frac{\eta_L K}{2}\|\nabla f(x_t)\|^2 + \frac{\eta_L}{2K}\mathbb{E}_t\left\|\sum_{i=1}^m w_i \sum_{k=0}^{K-1} (\nabla F_i(x_{t,k}^i) - \nabla F_i(x_t))\right\|^2$$

$$- \frac{\eta_L}{2K}\mathbb{E}_t\|\sum_{i=1}^m w_i \sum_{k=0}^{K-1} \nabla F_i(x_{t,k}^i)\|^2. \tag{63}$$

The use Jensen's Inequality:

$$A_1 \tag{64}$$

$$\overset{(a4)}{\leq} \frac{\eta_L K}{2} \|\nabla f(x_t)\|^2 + \frac{\eta_L}{2} \sum_{k=0}^{K-1} \sum_{i=1}^{m} w_i \mathbb{E}_t \left\| \nabla F_i(x_{t,k}^i) - \nabla F_i(x_t) \right\|^2$$

$$- \frac{\eta_L}{2K} \mathbb{E}_t \| \sum_{i=1}^{m} w_i \sum_{k=0}^{K-1} \nabla F_i(x_{t,k}^i) \|^2 \tag{65}$$

$$\overset{(a5)}{\leq} \frac{\eta_L K}{2} \|\nabla f(x_t)\|^2 + \frac{\eta_L L^2}{2m} \sum_{i=1}^{m} \sum_{k=0}^{K-1} \mathbb{E}_t \left\| x_{t,k}^i - x_t \right\|^2 - \frac{\eta_L}{2K} \mathbb{E}_t \| \sum_{i=1}^{m} w_i \sum_{k=0}^{K-1} \nabla F_i(x_{t,k}^i) \|^2 \tag{66}$$

$$\leq \left( \frac{\eta_L K}{2} + 10K^3 L^2 \eta_L^3 (A^2 + 1) \right) \|\nabla f(x_t)\|^2 + \frac{5L^2 \eta_L^3}{2} K^2 \sigma_L^2 + 10\eta_L^3 L^2 K^3 \sigma_G^2$$

$$- \frac{\eta_L}{2K} \mathbb{E}_t \| \sum_{i=1}^{m} w_i \sum_{k=0}^{K-1} \nabla F_i(x_{t,k}^i) \|^2 , \tag{67}$$

where (a2) follows from Assumption 2. (a3) is due to $\langle x, y \rangle = \frac{1}{2} \left[ \|x\|^2 + \|y\|^2 - \|x - y\|^2 \right]$ and (a4) uses Jensen's Inequality: $\|\sum_{i=1}^{m} w_i z_i\|^2 \leq \sum_{i=1}^{m} w_i \|z_i\|^2$, (a5) comes from Assumption 1. Then we consider $A_2$:

$$A_2 \tag{68}$$

$$= \mathbb{E}_t \|\Delta_t\|^2 = \mathbb{E}_t \left\| \eta_L \sum_{i=1}^{m} w_i \sum_{k=0}^{K-1} g_{t,k}^i \right\|^2 \tag{69}$$

$$= \eta_L^2 \mathbb{E}_t \left\| \sum_{i=1}^{m} w_i \sum_{k=0}^{K-1} g_{t,k}^i - \sum_{i=1}^{m} w_i \sum_{k=0}^{K-1} \nabla F_i(x_{t,k}^i) \right\|^2 + \eta_L^2 \mathbb{E}_t \left\| \sum_{i=1}^{m} w_i \sum_{k=0}^{K-1} \nabla F_i(x_{t,k}^i) \right\|^2 \tag{70}$$

$$\overset{(a6)}{\leq} \eta_L^2 \sum_{i=1}^{m} w_i^2 \sum_{k=0}^{K-1} \mathbb{E} \|g_i(x_{t,k}^i) - \nabla F_i(x_{t,k}^i)\|^2 + \eta_L^2 \mathbb{E}_t \| \sum_{i=1}^{m} w_i \sum_{k=0}^{K-1} \nabla F_i(x_{t,k}^i) \|^2 \tag{71}$$

$$\leq \sum_{i=1}^{m} w_i^2 \eta_L^2 K \sigma_L^2 + \eta_L^2 \mathbb{E}_t \| \sum_{i=1}^{m} w_i \sum_{k=0}^{K-1} \nabla F_i(x_{t,k}^i) \|^2 \tag{72}$$

where (a6) follows from $\| \sum_i w_i a_i \|^2 = \sum_i w_i^2 \|a_i\|^2$ where $a_i$ is an unbiased estimator.

Now we take expectation over iteration on both sides of expression:

$$f(x_{t+1}) \tag{73}$$

$$\leq f(x_t) - \eta\eta_L K \mathbb{E}_t \|\nabla f(x_t)\|^2 + \eta\mathbb{E}_t \langle \nabla f(x_t), \Delta_t + \eta_L K \nabla f(x_t)\rangle + \frac{L}{2}\eta^2 \mathbb{E}_t \|\Delta_t\|^2 \tag{74}$$

$$\overset{(a7)}{\leq} f(x_t) - \eta\eta_L K \left(\frac{1}{2} - 20L^2 K^2 \eta_L^2 (A^2+1)(\chi_{\boldsymbol{w}\|\boldsymbol{p}}^2 A^2 + 1)\right) \mathbb{E}_t \|\nabla f(x_t)\|^2$$

$$+ \frac{5\eta\eta_L^3 L^2 K^2}{2}(\sigma_L^2 + 4K\sigma_G^2) + \frac{\sum_i w_i^2 \eta^2 \eta_L^2 KL}{2}\sigma_L^2 + 20L^2 K^3 (A^2+1)\eta\eta_L^3 \chi_{\boldsymbol{w}\|\boldsymbol{p}}^2 \sigma_G^2$$

$$- \left(\frac{\eta\eta_L}{2K} - \frac{L\eta^2\eta_L^2}{2}\right) \mathbb{E}_t \left\|\frac{1}{m}\sum_{i=1}^m \sum_{k=0}^{K-1} \nabla F_i(x_{t,k}^i)\right\|^2 \tag{75}$$

$$\overset{(a8)}{\leq} f(x_t) - c\eta\eta_L K \mathbb{E}\|\nabla f(x_t)\|^2 + \frac{5\eta\eta_L^3 L^2 K^2}{2}(\sigma_L^2 + 4K\sigma_G^2) \tag{76}$$

$$+ \frac{\sum_i w_i^2 \eta^2 \eta_L^2 KL}{2}\sigma_L^2 + 20L^2 K^3(A^2+1)\eta\eta_L^3 \chi_{\boldsymbol{w}\|\boldsymbol{p}}^2 \sigma_G^2$$

$$- \left(\frac{\eta\eta_L}{2K} - \frac{L\eta^2\eta_L^2}{2}\right) \mathbb{E}_t \left\|\frac{1}{m}\sum_{i=1}^m \sum_{k=0}^{K-1} \nabla F_i(x_{t,k}^i)\right\|^2 \tag{77}$$

$$\overset{(a9)}{\leq} f(x_t) - c\eta\eta_L K \mathbb{E}_t\|\nabla f(x_t)\|^2 + \frac{5\eta\eta_L^3 L^2 K^2}{2}(\sigma_L^2 + 4K\sigma_G^2)$$

$$+ \frac{\sum_i w\eta^2 \eta_L^2 KL}{2}\sigma_L^2 + 20L^2 K^3 (A^2+1)\eta\eta_L^3 \chi_{\boldsymbol{w}\|\boldsymbol{p}}^2 \sigma_G^2 \,, \tag{78}$$

where (a7) is due to Lemma H.1, (a8) holds because there exists a constant $c > 0$ (for some $\eta_L$) satisfying $\frac{1}{2} - 10L^2 \frac{1}{m}\sum_{i-1}^m K^2 \eta_L^2 (A^2+1)(\chi_{\boldsymbol{w}\|\boldsymbol{p}}^2 A^2 + 1) > c > 0$, and the (a9) follows from $\left(\frac{\eta\eta_L}{2K} - \frac{L\eta^2\eta_L^2}{2}\right) \geq 0$ if $\eta\eta_l \leq \frac{1}{KL}$.

Rearranging and summing from $t = 0, \ldots, T-1$, we have:

$$\sum_{t=1}^{T-1} c\eta\eta_L K \mathbb{E}\|\nabla f(x_t)\|^2 \leq f(x_0) - f(x_T) + T(\eta\eta_L K)\Phi \,. \tag{79}$$

Which implies:

$$\frac{1}{T}\sum_{t=1}^{T-1} \mathbb{E}\|\nabla f(x_t)\|^2 \leq \frac{f_0 - f_*}{c\eta\eta_L KT} + \Phi \,, \tag{80}$$

where

$$\Phi = \frac{1}{c}\left[\frac{5\eta_L^2 KL^2}{2}(\sigma_L^2 + 4K\sigma_G^2) + \frac{\eta\eta_L L \sum_i w_i^2}{2}\sigma_L^2 + 20L^2 K^2 (A^2+1)\eta_L^2 \chi_{\boldsymbol{w}\|\boldsymbol{p}}^2 \sigma_G^2\right]. \tag{81}$$

**Corollary H.4.** *Suppose $\eta_L$ and $\eta$ are $\eta_L = \mathcal{O}\left(\frac{1}{\sqrt{T}KL}\right)$ and $\eta = \mathcal{O}\left(\sqrt{Km}\right)$ such that the conditions mentioned above are satisfied. Then for sufficiently large $T$, the iterates of FedEBA+ with $\alpha = 0$ satisfy:*

$$\min_{t\in[T]} \|\nabla f(\boldsymbol{x}_t)\|^2 \leq \mathcal{O}\left(\frac{(f^0 - f^*)}{\sqrt{mKT}}\right) + \mathcal{O}\left(\frac{\sqrt{m}\sum_i w_i^2 \sigma_L^2}{2\sqrt{KT}}\right) + \mathcal{O}\left(\frac{5(\sigma_L^2 + 4K\sigma_G^2)}{2KT}\right)$$

$$+ \mathcal{O}\left(\frac{20(A^2+1)\chi_{\boldsymbol{w}\|\boldsymbol{p}}^2 \sigma_G^2}{T}\right). \tag{82}$$

*According to the property of unified probability, we know $\frac{1}{m} \leq \sum_{i=1}^m w_i^2 \leq 1$, where the upper comes from $\sum_i w_i^2 \leq \sum_i w_i$ and lower comes from Cauchy-Schwarz inquality. Therefore, the convergence rate upper bound lies between $\mathcal{O}(\frac{1}{\sqrt{mKT}} + \frac{1}{T})$ and $\mathcal{O}(\frac{\sqrt{m}}{\sqrt{KT}} + \frac{1}{T})$.*

$\square$

## H.2 Analysis with $\alpha \neq 0$

To derivate the convergence rate of FedEBA+ with $\alpha \neq 0$, we need the following assumption:

**Assumption 4** (Error bound between practical global gradient and ideal gradient)**.** *In each round, we assume the aligned gradient $\nabla\overline{f}(x_t)$ and the gradient $\nabla f(x_t)$ is bounded: $\mathbb{E}\|\nabla\overline{f}(x_t) - \nabla f(x_t)\|^2 \leq \rho^2, \forall i, t$. For simplicity of analysis, let $\rho$ is comparable to $\sigma_L$, i.e., $\rho \sim \sigma_L$, since they are both constant bounds.*

To simplify the notation, we define $h_{t,k}^i = (1 - \alpha)\nabla F_i(x_{t,k}^i) + \alpha\nabla\overline{f}(x_t)$.

**Lemma H.5.** *For any step-size satisfying $\eta_L \leq \frac{1}{8LK}$, we can have the following results:*

$$
\mathbb{E}\|x_{t,k}^i - x_t\|^2 \leq 5K(1-\alpha)^2(\eta_L^2\sigma_L^2 + 6K\eta_L^2\sigma_G^2) + + 30K^2\eta_L^2\alpha^2\rho^2
$$
$$
+ 30K^2\eta_L^2(1 + A^2(1-\alpha)^2)\|\nabla f(x_t)\|^2. \tag{83}
$$

*Proof.*

$$
\mathbb{E}_t\|x_{t,k}^i - x_t\|^2 \tag{84}
$$
$$
= \mathbb{E}_t\|x_{t,k-1}^i - x_t - \eta_L h_{t,k-1}^t\|^2 \tag{85}
$$
$$
= \mathbb{E}_t\|x_{t,k-1}^i - x_t - \eta_L((1-\alpha)g_{t,k-1}^t + \alpha\nabla\overline{f}(x_t) - (1-\alpha)\nabla F_i(x_{t,k-1}^i)
$$
$$
+ (1-\alpha)\nabla F_i(x_{t,k-1}^i) - (1-\alpha)\nabla F_i(x_t) + (1-\alpha)\nabla F_i(x_t) + \nabla f(x_t) - \nabla f(x_t))\|^2
$$
$$
\leq (1 + \frac{1}{2K-1})\mathbb{E}_t\|x_{t,k-1}^i - x_t\|^2 + (1-\alpha)^2\eta_L^2\sigma_L^2 + 6K\eta_L^2 L^2\mathbb{E}_t\|x_{t,k-1}^i - x_t\|^2
$$
$$
+ 6K\eta_L^2\alpha^2\mathbb{E}\|\nabla\overline{f}(x_t) - \nabla f(x_t)\|^2 + 6K\eta_L^2(1-\alpha)^2(\sigma_G^2 + A^2\|\nabla f(x_t)\|^2)
$$
$$
+ 6K\eta_L^2\|\nabla f(x_t)\|^2 \tag{86}
$$
$$
\leq (1 + \frac{1}{K-1})\mathbb{E}_t\|x_{t,k-1}^i - x_t\|^2 + (1-\alpha)^2\eta_L^2\sigma_L^2
$$
$$
+ 6K\eta_L^2\alpha^2\rho^2 + 6K\eta_L^2(1-\alpha)^2(\sigma_G^2 + A^2\|\nabla f(x_t)\|^2) + 6K\eta_L^2\|\nabla f(x_t)\|^2, \tag{87}
$$

Unrolling the recursion, we obtain:

$$
\mathbb{E}_t\|x_{t,k}^i - x_t\|^2 \tag{88}
$$
$$
\leq \sum_{p=0}^{k-1}(1 + \frac{1}{K-1})^p \left((1-\alpha)^2\eta_L^2\sigma_L^2 + 6K(1-\alpha)^2\eta_L^2\sigma_G^2 + 6K\alpha^2\eta_L^2\rho^2\right.
$$
$$
\left. + 6K\eta_L^2(A^2(1-\alpha)^2 + 1)\|\nabla f(x_t)\|^2\right) \tag{89}
$$
$$
\leq (K-1)\left[(1 + \frac{1}{K-1})^K - 1\right]\left[(1-\alpha)^2\eta_L^2\sigma_L^2\right.
$$
$$
\left. + 6K(1-\alpha)^2\eta_L^2\sigma_G^2 + 6K\alpha^2\eta_L^2\rho^2 + 6K\eta_L^2(A^2(1-\alpha)^2 + 1)\|\nabla f(x_t)\|^2\right] \tag{90}
$$
$$
\leq 5K\eta_L^2(1-\alpha)^2(\sigma_L^2 + 6K\sigma_G^2) + 30K^2\eta_L^2\alpha^2\rho^2 + 30K^2\eta_L^2(A^2(1-\alpha)^2 + 1)\|\nabla f(x_t)\|^2. \tag{91}
$$

Similarly, to get the convergence rate of objective $f(x_t)$, we first focus on $f(x_t)$:

$$
\mathbb{E}_t[f(x_{t+1})] \overset{(a1)}{\leq} f(x_t) + \langle\nabla f(x_t), \mathbb{E}_t[x_{t+1} - x_t]\rangle + \frac{L}{2}\mathbb{E}_t[\|x_{t+1} - x_t\|^2] \tag{92}
$$
$$
= f(x_t) + \langle\nabla f(x_t), \mathbb{E}_t[\eta\Delta_t + \eta\eta_L K\nabla f(x_t) - \eta\eta_L K\nabla f(x_t)]\rangle + \frac{L}{2}\eta^2\mathbb{E}_t[\|\Delta_t\|^2] \tag{93}
$$
$$
= f(x_t) - \eta\eta_L K\|\nabla f(x_t)\|^2 + \eta\underbrace{\langle\nabla f(x_t), \mathbb{E}_t[\Delta_t + \eta_L K\nabla f(x_t)]\rangle}_{A_1} + \frac{L}{2}\eta^2\underbrace{\mathbb{E}_t\|\Delta_t\|^2}_{A_2}, \tag{94}
$$

where (a1) follows from the Lipschitz continuity condition. Here, the expectation is over the local data SGD and the filtration of $x_t$. However, in the next analysis, the expectation is

over all randomness, including client sampling. This is achieved by taking expectation on both sides of the above equation over client sampling.

To begin with, we consider $A_1$:

$$A_1 \tag{95}$$
$$= \langle \nabla f(x_t), \mathbb{E}_t[\Delta_t + \eta_L K \nabla f(x_t)] \rangle \tag{96}$$
$$= \left\langle \nabla f(x_t), \mathbb{E}_t[-\sum_{i=1}^m w_i \sum_{k=0}^{K-1} \eta_L h_{t,k}^i + \eta_L K \nabla f(x_t)] \right\rangle \tag{97}$$
$$\stackrel{(a2)}{=} \left\langle \nabla f(x_t), \mathbb{E}_t[-\sum_{i=1}^m w_i \sum_{k=0}^{K-1} \eta_L[(1-\alpha)\nabla F_i(x_{t,k}^i) + \alpha\overline{f}(x_t)] + \eta_L K \nabla f(x_t)] \right\rangle. \tag{98}$$

For the above equation, we can separate the $\nabla f(x_t)$ into $(1-\alpha)\nabla f(x_t)$ and $\alpha\nabla f(x_t)$ two terms, thus, we have:

$$A_1 \tag{99}$$
$$= \left\langle \sqrt{\eta_L K}\nabla f(x_t), \right.$$
$$\left. -\frac{\sqrt{\eta_L}}{\sqrt{K}}\mathbb{E}_t\left(\sum_{i=1}^m w_i \sum_{k=0}^{K-1}(1-\alpha)[\nabla F_i(x_{t,k}^i) - \nabla f(x_t)] + \sum_{i=1}^m w_i \sum_{k=0}^{K-1}\alpha[\nabla\overline{f}(x_t) - \nabla f(x_t)]\right) \right\rangle \tag{100}$$
$$\stackrel{(a3)}{=} \frac{\eta_L K}{2}\|\nabla f(x_t)\|^2 - \frac{\eta_L}{2K}\mathbb{E}_t\|\sum_{i=1}^m w_i \sum_{k=0}^{K-1}[(1-\alpha)\nabla F_i(x_{t,k}^i) + \alpha\nabla\overline{f}(x_t)]\|^2$$
$$+ \frac{\eta_L}{2K}\mathbb{E}_t\left\|\sum_{i=1}^m w_i \sum_{k=0}^{K-1}\left((1-\alpha)[\nabla F_i(x_{t,k}^i) - \nabla f(x_t)] + \alpha[\nabla\overline{f}(x_t) - \nabla f(x_t)]\right)\right\|^2 \tag{101}$$
$$\stackrel{(a4)}{\leq} \frac{\eta_L K}{2}\|\nabla f(x_t)\|^2 + \frac{\eta_L(1-\alpha)^2}{2m}\sum_{k=0}^{K-1}\sum_{i=1}^m w_i\mathbb{E}_t\left\|\nabla F_i(x_{t,k}^i) - \nabla F_i(x_t)\right\|^2$$
$$+ \frac{\eta_L\alpha^2}{2m}\sum_{k=0}^{K-1}\sum_{i=1}^m w_i\mathbb{E}\|\nabla\overline{f}(x_t) - \nabla f(x_t)\|^2 - \frac{\eta_L}{2K}\mathbb{E}_t\|\sum_{i=1}^m w_i \sum_{k=0}^{K-1}[(1-\alpha)\nabla F_i(x_{t,k}^i) + \alpha\nabla\overline{f}(x_t)]\|^2 \tag{102}$$
$$\stackrel{(a5)}{\leq} \frac{\eta_L K}{2}\|\nabla f(x_t)\|^2 + \frac{\eta_L(1-\alpha)^2 L^2}{2m}\sum_{i=1}^m\sum_{k=0}^{K-1}\mathbb{E}_t\left\|x_{t,k}^i - x_t\right\|^2$$
$$+ \frac{\eta_L\alpha^2}{2m}\sum_{i=1}^m\sum_{k=0}^{K-1}\mathbb{E}\|\nabla\overline{f}(x_t) - \nabla f(x_t)\|^2 - \frac{\eta_L}{2K}\mathbb{E}_t\|\sum_{i=1}^m w_i \sum_{k=0}^{K-1}[(1-\alpha)\nabla F_i(x_{t,k}^i) + \alpha\nabla\overline{f}(x_t)]\|^2 \tag{103}$$
$$\leq \frac{\eta_L K}{2}\|\nabla f(x_t)\|^2 + \frac{\eta_L(1-\alpha)^2}{2m}\sum_{i=1}^m\sum_{k=0}^{K-1}\left(5K\eta_L(1-\alpha)^2(\sigma_L^2 + 6K\sigma_G^2) + 30K^2\eta_L^2[\alpha^2\rho^2\right.$$
$$\left. +(1 + A^2(1-\alpha)^2)\|\nabla f(x_t)\|^2]\right) + \frac{\eta_L^2\alpha^2}{2}K\rho^2 - \frac{\eta_L}{2K}\mathbb{E}\|\sum_{i=1}^m w_i \sum_{k=0}^{K-1}[(1-\alpha)\nabla F_i(x_{t,k}^i) + \alpha\nabla\overline{f}(x_t)]\|^2, \tag{104}$$

where (a2) follows from Assumption 2. (a3) is due to $\langle x, y \rangle = \frac{1}{2}\left[\|x\|^2 + \|y\|^2 - \|x-y\|^2\right]$ and (a4) uses Jensen's Inequality: $\|\sum_{i=1}^m w_i z_i\|^2 \leq \sum_{i=1}^m w_i \|z_i\|^2$, (a5) comes from Assumption 1.

Then we consider $A_2$:

$$A_2 \tag{105}$$

$$= \mathbb{E}_t \|\Delta_t\|^2 \tag{106}$$

$$= \mathbb{E}_t \left\| \eta_L \sum_{i=1}^{m} w_i \sum_{k=0}^{K-1} h_{t,k}^i \right\|^2 \tag{107}$$

$$= \eta_L^2 \mathbb{E}_t \left\| \sum_{i=1}^{m} w_i \sum_{k=0}^{K-1} \left[ (1-\alpha)\nabla F_i(x_{t,k}^i; \xi_t^i) + \alpha \overline{f}(x_t) \right] \right\|^2 \tag{108}$$

$$\leq \eta_L^2 \mathbb{E} \| \sum_{i=1}^{m} w_i \sum_{k=0}^{K-1} \left[ (1-\alpha)\nabla F_i(x_{t,k}^i; \xi_t^i) + \alpha \overline{f}(x_t) \right]$$

$$- (1-\alpha)\nabla F_i(x_{t,k}^i) + (1-\alpha)\nabla F_i(x_{t,k}^i) \|^2 \tag{109}$$

$$\overset{(a6)}{\leq} \sum_{i=1}^{m} w_i^2 \eta_L^2 K (1-\alpha)^2 \sigma_L^2 + \eta_L^2 \mathbb{E} \| \sum_{i=1}^{m} w_i \sum_{k=0}^{K-1} [(1-\alpha)\nabla F_i(x_{t,k}^i) + \alpha \nabla \overline{f}(x_t)] \|^2 \tag{110}$$

where (a6) follows from Assumption 2.

Now we substitute the expressions for $A_1$ and $A_2$ and take the expectation over the client sampling distribution on both sides. It should be noted that the derivation of $A_1$ and $A_2$ above is based on considering the expectation over the sampling distribution:

$$f(x_{t+1}) \tag{111}$$

$$\leq f(x_t) - \eta \eta_L K \mathbb{E}_t \|\nabla f(x_t)\|^2 + \eta \mathbb{E}_t \langle \nabla f(x_t), \Delta_t + \eta_L K \nabla f(x_t) \rangle + \frac{L}{2} \eta^2 \mathbb{E}_t \|\Delta_t\|^2 \tag{112}$$

$$\overset{(a7)}{\leq} f(x_t) - \eta \eta_L K \left( \frac{1}{2} - 30\alpha^2 L^2 K^2 \eta_L^2 ((1-\alpha)^2 A^2 + 1) \right) \mathbb{E} \|\nabla f(x_t)\|^2$$

$$+ \frac{5(1-\alpha)^2 \eta \eta_L^3 L^2 K^2}{2} \left[ 5(1-\alpha)^2 (\sigma_L^2 + 6K\sigma_G^2) + 30K\alpha^2 \rho^2 \right] + \frac{\eta \eta_L^2 \alpha^2}{2} K \rho^2$$

$$+ \frac{\sum_{i=1}^{m} w_i^2 L \eta^2 \eta_L^2}{2} (1-\alpha)^2 K \sigma_L^2$$

$$- \left( \frac{\eta \eta_L}{2K} - \frac{\eta^2 \eta_L^2 L}{2} \right) \mathbb{E} \left\| \sum_{i=1}^{m} w_i \sum_{k=0}^{K-1} [(1-\alpha)\nabla F_i(x_{t,k}^i) + \alpha \nabla \overline{f}(x_t)] \right\|^2 \tag{113}$$

where (a7) comes from $\frac{1}{2} - 15\alpha^2 L^2 K^2 \eta_L^2 ((1-\alpha)^2 A^2 + 1) > c > 0$ and $\frac{\eta \eta_L}{2K} - \frac{\eta^2 \eta_L^2 L}{2} \geq 0$.

Rearranging and summing from $t = 0, \ldots, T-1$, we have:

$$\sum_{t=1}^{T-1} c \eta \eta_L K \mathbb{E} \|\nabla f(x_t)\|^2 \leq f(x_0) - f(x_T) + T(\eta \eta_L K) \Phi. \tag{114}$$

Which implies:

$$\frac{1}{T} \sum_{t=1}^{T-1} \mathbb{E} \|\nabla f(x_t)\|^2 \leq \frac{f_0 - f_*}{c \eta \eta_L K T} + \tilde{\Phi}, \tag{115}$$

where

$$\tilde{\Phi} = \frac{1}{c} \left[ \frac{5\eta_L^2 K L^2 (1-\alpha)^4}{2} (\sigma_L^2 + 6K\sigma_G^2) + 15K^2 \eta_L^2 (1-\alpha)^2 \alpha^2 \rho^2 \right.$$

$$\left. + \frac{\sum_{i=1}^{m} w_i^2 \eta \eta_L L (1-\alpha)^2}{2} \sigma_L^2 + \frac{\eta_L \alpha^2 \rho^2}{2} \right]. \tag{116}$$

$$\square$$

**Corollary H.6.** *Suppose $\eta_L$ and $\eta$ are $\eta_L = \mathcal{O}\left(\frac{1}{\sqrt{T}KL}\right)$ and $\eta = \mathcal{O}\left(\sqrt{Km}\right)$ such that the conditions mentioned above are satisfied. Then for sufficiently large T, the iterates of FedEBA+ with $\alpha \neq 0$ satisfy:*

$$\min_{t \in [T]} \|\nabla f(\boldsymbol{x}_t)\|^2 \leq \mathcal{O}\left(\frac{(f^0 - f^*)}{\sqrt{mKT}}\right) + \mathcal{O}\left(\sum_{i=1}^{m} w_i^2 \frac{(1-\alpha)^2 \sqrt{m}\sigma_L^2}{2\sqrt{KT}}\right) + \mathcal{O}\left(\frac{5(1-\alpha)^2(\sigma_L^2 + 6K\sigma_G^2)}{2KT}\right)$$
$$+ \mathcal{O}\left(\frac{15(1-\alpha)^2\alpha^2\rho^2}{T}\right) + \mathcal{O}\left(\frac{\alpha^2\rho^2}{2\sqrt{T}K}\right). \tag{117}$$

*For the convergence rate of FedEBA+ with $\alpha \neq 0$, the convergence rate order can be represented as :$\mathcal{O}(\frac{(1-\alpha)^2 \sum_i w_i^2 \sqrt{m}\sigma_L^2 + \alpha^2\sqrt{K}\rho^2}{\sqrt{KT}} + \frac{1}{T})$, where $K << m$ and $\sigma_L \sim \rho$, thus a larger $\alpha$ indicating a tighter convergence upper bound than only using reweight aggregation. In addition, when $w_i = \frac{1}{m}$, i.e., uniform aggregation, it is $\mathcal{O}(\frac{(1-\alpha)^2\sigma_L^2 + \alpha^2\sqrt{K/m}\rho^2}{\sqrt{mKT}} + \frac{1}{T})$, since $\sqrt{K/m} << 1$, which indicating when using alignment update the convergence result will be faster than FedAvg.*

## I  FAIRNESS ANALYSIS VIA VARIANCE

To demonstrate the ability of FedEBA+ to enhance fairness in federated learning, we first employ a two-user toy example to demonstrate how FedEBA+ can achieve a more balanced performance between users in comparison to FedAvg and q-FedAvg, thus ensuring fairness. Furthermore, we use a general class of regression models and strongly convex cases to show how FedEBA+ reduces the variance among users and thus improves fairness.

### I.1  TOY CASE FOR ILLUSTRATING FAIRNESS

In Figure 1, the term "performance gap" refers to the performance disparity between two clients, calculated by $\|F_1(x) - F_2(x)\|$. The magnitude of this gap effectively reflects the variance among clients. Considering that $Var = \frac{|F_1(x) - F_2(x)|^2}{4}$, it can be inferred that a larger performance gap $|F_1(x) - F_2(x)|$ corresponds to a larger variance, thus indicating less fairness.

In this section, we examine the performance fairness of our algorithm. In particular, we consider two clients participating in training, each with a regression model: $f_1(x_t) = 2(x-2)^2$, $f_2(x_t) = \frac{1}{2}(x+4)^2$. Corresponding,

$$\nabla f_1(x_t) = 4(x-2), \tag{118}$$

$$\nabla f_2(x_t) = (x+4). \tag{119}$$

When the global model parameter $x_t = 0$ is sent to each client, each client will update the model by running gradient decent, here w.l.o.g, we consider one single-step gradient decent, and stepsize $\lambda = \frac{1}{4}$:

$$x_1^{t+1} = x_t - \lambda\nabla f_1(x_t) = 2, \tag{120}$$

$$x_2^{t+1} = x_t - \lambda\nabla f_2(x_t) = -1. \tag{121}$$

The aggregation weights for FedAvg and FedEBA+ can be concluded as:

$$(p_1, p_2)_{AVG} = (\frac{1}{2}, \frac{1}{2}); (p_1, p_2)_{EBA+} = (\frac{1}{1 + e^{9/2}}, \frac{e^{9/2}}{1 + e^{9/2}}). \tag{122}$$

Thus, for uniform aggregation, i.e., FedAvg:

$$x_{AVG}^{t+1} = \frac{1}{2}(x_1^{t+1} + x_2^{t+1}) = \frac{1}{2}. \tag{123}$$

While for FedEBA+:

$$x_{EBA+}^{t+1} = \frac{e^{f_1(x_1^{t+1})}}{e^{f_1(x_1^{t+1})} + e^{f_2(x_2^{t+1})}}x_1^{t+1} + \frac{e^{f_2(x_2^{t+1})}}{e^{f_1(x_1^{t+1})} + e^{f_2(x_2^{t+1})}}x_2^{t+1} \approx -0.1\,. \tag{124}$$

Therefore,

$$\text{Var}_{AVG} = \frac{1}{2}\sum_{i=1}^{2}\left(f_i(x_{AVG}^{t+1}) - \frac{1}{2}\sum_{i=1}^{2}(f_i(x_{AVG}^{t+1}))\right) = 2*(2.81)^2\,, \tag{125}$$

$$\text{Var}_{EBA+} = \frac{1}{2}\sum_{i=1}^{2}\left(f_i(x_{EBA+}^{t+1}) - \frac{1}{2}\sum_{i=1}^{2}(f_i(x_{EBA+}^{t+1}))\right) = 2*(0.6)^2\,. \tag{126}$$

Thus, we prove that FedEBA+ achieves a much smaller variance than uniform aggregation.

Furthermore, for q-FedAvg, we consider $q = 2$ that is also used in the proof of (Li et al., 2019a):

$$\nabla x_1^t = L(x^t - x_1^{t+1}) = -2\,, \tag{127}$$
$$\nabla x_2^t = L(x^t - x_2^{t+1}) = 1\,. \tag{128}$$

Thus, we have:

$$\Delta_1^t = f_1^q(x_t)\nabla x_1^t = 8*(-2) = -16\,, \tag{129}$$
$$h_1^t = qf_1^{q-1}(x_t)\|\nabla x_1^t\|^2 + Lf_1^q(x_t) = 1 \times 1 \times 2^2 + 8 = 12\,, \tag{130}$$
$$\Delta_2^t = f_2^q(x_t)\nabla x_2^t = 8*(1) = 8\,, \tag{131}$$
$$h_2^t = qf_2^{q-1}(x_t)\|\nabla x_2^t\|^2 + Lf_2^q(x_t) = 1 \times 1 \times 1^2 + 8 = 9\,. \tag{132}$$

The aggregation weights for q-FFL can be concluded as:

$$(p_1, p_2)_{q-FFL} = (\frac{4}{13}, \frac{4}{13}). \tag{133}$$

Finally, we can update the global parameter as:

$$x_{q-FFL}^{t+1} = x^t - \frac{\sum_i \Delta_i^t}{\sum_i h_i^t} \approx -0.4\,. \tag{134}$$

Then we can easily get:

$$\text{Var}_{q-FFL} = \frac{1}{2}\sum_{i=1}^{2}\left(f_i(x_{q-FFL}^{t+1}) - \frac{1}{2}\sum_{i=1}^{m}(f_i(x_{q-FFL}^{t+1}))\right) = 2*(2.52)^2$$

In conclusion, we prove that

$$\text{Var}_{EBA+} \leq \text{Var}_{q-FFL} \leq \text{Var}_{AVG}\,. \tag{135}$$

In this case, the normalized performance's entropy, after maxing the constrained entropy of aggregation probability, exhibits a relationship akin to variance (greater entropy corresponds to improved fairness).

$$\text{Entropy}\left(f\left(x_{EBA+}^{t+1}\right)\right) = -\sum_{i=1}^{2}\frac{f_i\left(x_{EBA+}^{t+1}\right)}{\sum_{j=1}^{2}f_j\left(x_{EBA+}^{t+1}\right)}\log\left(\frac{f_j\left(x_{EBA+}^{t+1}\right)}{\sum_{i=j}^{2}f_i\left(x_{EBA+}^{t+1}\right)}\right) \approx 0.996 \tag{136}$$

$$\text{Entropy}\left(f\left(x_{q-FFL}^{t+1}\right)\right) = -\sum_{i=1}^{2}\frac{f_i\left(x_{q-FFL}^{t+1}\right)}{\sum_{j=1}^{2}f_j\left(x_{q-FFL}^{t+1}\right)}\log\left(\frac{f_j\left(x_{q-FFL}^{t+1}\right)}{\sum_{i=j}^{2}f_i\left(x_{q-FFL}^{t+1}\right)}\right) \approx 0.942, \tag{137}$$

$$\text{Entropy}\left(f\left(x_{AVG}^{t+1}\right)\right) = -\sum_{i=1}^{2}\frac{f_i\left(x_{\text{avg}}^{t+1}\right)}{\sum_{j=1}^{2}f_j\left(x_{\text{AVG}}^{t+1}\right)}\log\left(\frac{f_j\left(x_{\text{AVG}}^{t+1}\right)}{\sum_{i=j}^{2}f_i\left(x_{\text{AVG}}^{t+1}\right)}\right) \approx 0.890 \tag{138}$$

where

$$f_1\left(x_{EBA+}^{t+1}\right) = 2*(2.1)^2, f_2\left(x_{EBA+}^{t+1}\right) = \frac{1}{2}*(3.9)^2, \tag{139}$$

$$f_1\left(x_{q-FFL}^{t+1}\right) = 2*(2.4)^2, f_2\left(x_{q-FFL}^{t+1}\right) = \frac{1}{2}*(3.6)^2, \tag{140}$$

$$f_1\left(x_{AVG}^{t+1}\right) = 2*(1.5)^2, f_2\left(x_{AVG}^{t+1}\right) = \frac{1}{2}*(4.5)^2. \tag{141}$$

Therefore, $Entropy(f(x_{EBA+}^{t+1})) > Entropy(f(x_{q-FFL}^{t+1})) > Entropy(f(x_{AVG}^{t+1}))$ and $Var_{EBA+} < Var_{q-FFL} < Var_{AVG}$.

### I.2  ANALYSIS FAIRNESS BY GENERALIZED LINEAR REGRESSION MODEL

**Our setting.** In this section, we consider a generalized linear regression setting, which follows from that in (Lin et al., 2022).

Suppose that the true parameter on client $i$ is $\mathbf{w}_i$, and there are $n$ samples on each client. The observations are generated by $\hat{y}_{i,k}(\mathbf{w}_i, \xi_{i,k}) = T(\xi_{i,k})^\top \mathbf{w}_i - A(\xi_{i,k})$, where the $A(\xi_{i,k})$ are i.i.d and distributed as $\mathcal{N}\left(0, \sigma_1^2\right)$. Then the loss on client $i$ is $F_i\left(\mathbf{x}_i\right) = \frac{1}{2n}\sum_{k=1}^n\left(T(\xi_{i,k})^\top \mathbf{x}_i - A(\xi_{i,k}) - \hat{y}_{i,k}\right)^2$.

We compare the performance of fairness of different aggregation methods. Recall Defination 3.1. We measure performance fairness in terms of the variance of the test accuracy/losses.

**Solutions of different methods** First, we derive the solutions of different methods. Let $\boldsymbol{\Xi}_i = \left(T(\xi_{i,1}), T(\xi_{i,2}), \ldots, T(\xi_{i,n})\right)^\top$, $\mathbf{A}_i = \left(A(\xi_{i,1}), A(\xi_{i,2}), \ldots, A(\xi_{i,n})\right)^\top$ and $\mathbf{y}_i = \left(y_{i,1}, y_{i,2}, \ldots, y_{i,n}\right)^\top$. Then the loss on client $i$ can be rewritten as $F_i\left(\mathbf{x}_i\right) = \frac{1}{2n}\left\|\boldsymbol{\Xi}_i\mathbf{x}_i - \mathbf{A}_i - \mathbf{y}_i\right\|_2^2$, where $\text{rank}\left(\boldsymbol{\Xi}_i\right) = d$. The least-square estimator of $\mathbf{w}_i$ is

$$\left(\boldsymbol{\Xi}_i^\top \boldsymbol{\Xi}_i\right)^{-1}\boldsymbol{\Xi}_i^\top\left(\mathbf{y}_i + \mathbf{A}_i\right). \tag{142}$$

*FedAvg:* For FedAvg, the solution is defined as $\mathbf{w}^{\text{Avg}} = \text{argmin}_{\mathbf{w}\in\mathbb{R}^d}\frac{1}{m}\sum_{i=1}^m F_i(\mathbf{w})$. One can check that $\mathbf{w}^{\text{Avg}} = \left(\sum_{i=1}^m \boldsymbol{\Xi}_i^\top \boldsymbol{\Xi}_i\right)^{-1}\sum_{i=1}^m\boldsymbol{\Xi}_i^\top\left(\mathbf{y}_i + \mathbf{A}_i\right) = \left(\sum_{i=1}^m \boldsymbol{\Xi}_i^\top \boldsymbol{\Xi}_i\right)^{-1}\sum_{i=1}^m\boldsymbol{\Xi}_i^\top \boldsymbol{\Xi}_i\hat{\mathbf{w}}_i + \Lambda$, where $\Lambda = \left(\sum_{i=1}^m \boldsymbol{\Xi}_i^\top \boldsymbol{\Xi}_i\right)^{-1}\sum_{i=1}^m\boldsymbol{\Xi}_i^\top A_i$ and $\hat{\mathbf{w}}_i = \text{argmin}_{\mathbf{x}\in\mathbb{R}^d} f_i(x_i)$ is the solution on client $i$.

*FedEBA+:* For our method FedEBA+, the solution of the global model is $\mathbf{w}^{\text{EBA+}} = \text{argmin}_{\mathbf{w}\in\mathbb{R}^d}\sum_{i=1}^m p_i F_i(\mathbf{w}) = \left(\sum_{i=1}^m p_i\boldsymbol{\Xi}_i^\top \boldsymbol{\Xi}_i\right)^{-1}\sum_{i=1}^m p_i\boldsymbol{\Xi}_i^\top \boldsymbol{\Xi}_i\hat{\mathbf{w}}_i + \hat{\Lambda}$, where $p_i \propto e^{F_i}(\mathbf{w_i})$, and $\hat{\Lambda} = \left(\sum_{i=1}^m p_i\boldsymbol{\Xi}_i^\top \boldsymbol{\Xi}_i\right)^{-1}\sum_{i=1}^m p_i\boldsymbol{\Xi}_i^\top A_i$

Following the setting of (Lin et al., 2022), to make the calculations clean, we assume $\boldsymbol{\Xi}_i^\top \boldsymbol{\Xi}_i = nb_i\boldsymbol{I}_d$. Then the solutions of different methods can be simplified as

- FedAvg: $\mathbf{w}^{\text{Avg}} = \frac{\sum_{i=1}^m b_i(\hat{\mathbf{w}}_i + A_i)}{\sum_{i=1}^m b_i}$.
- FedEBA+: $\mathbf{w}^{\text{Avg}} = \frac{\sum_{i=1}^m b_i p_i(\hat{\mathbf{w}}_i + A_i)}{\sum_{i=1}^m b_i p_i}$.

**Test Loss** We compute the test losses of different methods. In this part, we assume $b_i = b$ to make calculations clean. This is reasonable since we often normalize the data.

Recall that the dataset on client $i$ is $(\boldsymbol{\Xi}_i, \mathbf{y}_i)$, where $\boldsymbol{\Xi}_i$ is fixed and $\mathbf{y}_i$ follows Gaussian distribution $\mathcal{N}\left(\boldsymbol{\Xi}_i\mathbf{w}_i, \sigma_2^2\boldsymbol{I}_n\right)$. Then the data heterogeneity across clients only lies in the heterogeneity of $\mathbf{w}_i$. Besides, since distribution of $\Lambda$ also follows gaussian distribution $\mathcal{N}\left(0, \sigma_1^2\boldsymbol{I}_n\right)$, thus $\mathbf{w}_i + A_i$ follows from $\mathcal{N}\left(\boldsymbol{\Xi}_i\mathbf{w}_i, \sigma^2\boldsymbol{I}_n\right)$, where $\sigma^2 = \sigma_1^2 + \sigma_2^2$. Then, we can obtain the distribution of the solutions of different methods. Let $\overline{\mathbf{w}} = \frac{\sum_{i=1}^N \mathbf{w}_i}{N}$. We have

- FedAvg: $\mathbf{w}^{\mathrm{Avg}} \sim \mathcal{N}\left(\overline{\mathbf{w}}, \frac{\sigma^2}{bNn}\boldsymbol{I}_d\right)$.

- FedEBA+: $\mathbf{w}^{\mathrm{EBA+}} \sim \mathcal{N}\left(\tilde{\mathbf{w}}, \sum_{i=1}^{N} p_i^2 \frac{\sigma^2}{bn}\boldsymbol{I}_d\right)$, where $\tilde{\mathbf{w}} = \sum_{i=1}^{N} p_i w_i$.

Since $\boldsymbol{\Xi}_i$ is fixed, we assume the test data is $(\boldsymbol{\Xi}_i, \mathbf{y}_i')$ where $\mathbf{y}_i' = \boldsymbol{\Xi}_i \mathbf{w}_i + \mathbf{z}_i'$ with $\mathbf{z}_i' \sim \mathcal{N}\left(\mathbf{0}_n, \sigma_z^2 \boldsymbol{I}_n\right)$ independent of $\mathbf{z}_i$. Then the test loss on client $k$ is defined as:

$$F_i^{\mathrm{te}}\left(\mathbf{x}_i\right) = \frac{1}{2n}\mathbb{E}\left\|\boldsymbol{\Xi}_i \mathbf{x}_i + A_i - \mathbf{y}_i'\right\|_2^2 \tag{143}$$

$$= \frac{1}{2n}\mathbb{E}\left\|\boldsymbol{\Xi}_i \mathbf{x}_i + A_i - \left(\boldsymbol{\Xi}_i \mathbf{w}_i + \mathbf{z}_i'\right)\right\|_2^2 \tag{144}$$

$$= \frac{\tilde{\sigma}^2}{2} + \frac{1}{2n}\mathbb{E}\left\|\boldsymbol{\Xi}_i\left(\mathbf{x}_i - \mathbf{w}_i\right)\right\|_2^2 \tag{145}$$

$$= \frac{\tilde{\sigma}^2}{2} + \frac{b}{2}\mathbb{E}\left\|\mathbf{x}_i - \mathbf{w}_i\right\|_2^2 \tag{146}$$

$$= \frac{\tilde{\sigma}^2}{2} + \frac{b}{2}\operatorname{tr}\left(\operatorname{var}\left(\mathbf{x}_i\right)\right) + \frac{b}{2}\left\|\mathbb{E}\mathbf{x}_i - \mathbf{w}_i\right\|_2^2 . \tag{147}$$

where $\tilde{\sigma}$ is a Gaussian variance, which comes from the fact that both $A_i$ and $z_i'$ follow Gaussian distribution with mean 0.

Therefore, for different methods, we can compute that

$$F_i^{\mathrm{te}}\left(\mathbf{w}^{\mathrm{Avg}}\right) = \frac{\tilde{\sigma}^2}{2} + \frac{\tilde{\sigma}^2 d}{2Nn} + \frac{b}{2}\left\|\overline{\mathbf{w}} - \mathbf{w}_i\right\|_2^2 , \tag{148}$$

$$F_i^{\mathrm{te}}\left(\mathbf{w}^{\mathrm{EBA+}}\right) = \frac{\tilde{\sigma}^2}{2} + \sum_{k=1}^{N} p_i^2 \frac{\tilde{\sigma}^2 d}{2n} + \frac{b}{2}\left\|\tilde{\mathbf{w}} - \mathbf{w}_i\right\|_2^2 . \tag{149}$$

Define var as the variance operator. Then we give the formal version of Theorem 5.4.

The variance of test losses on different clients of different aggregation methods are as follows:

$$V^{\mathrm{Avg}} = \operatorname{var}\left(F_i^{te}\left(\mathbf{w}^{\mathrm{Avg}}\right)\right) = \frac{b^2}{4}\operatorname{var}\left(\left\|\overline{\mathbf{w}} - \mathbf{w}_i\right\|_2^2\right) , \tag{150}$$

$$V^{\mathrm{EBA+}} = \operatorname{var}\left(F_i^{te}\left(\mathbf{w}^{\mathrm{EBA+}}\right)\right) = \frac{b^2}{4}\operatorname{var}\left(\left\|\tilde{\mathbf{w}} - \mathbf{w}_i\right\|_2^2\right) . \tag{151}$$

Based on a simple fact: assign larger weights to smaller values and smaller weights to larger values, and give a detailed mathematical proof to show that the variance of such a distribution is smaller than the variance of a uniform distribution. Which means $V^{\mathrm{EBA+}} \leq V^{\mathrm{Avg}}$.

Formally, let $\|\tilde{\mathbf{w}} - \mathbf{w}_i\|^2 = A_i$. From equation (149), we know that $F_i^{te}(\mathbf{w}^{\mathrm{EBA+}}) \propto A_i$, and $p_i \propto F_i$. Thus, we know $p_i \propto A_i$.

Then, we consider the expression of $V^{\mathrm{EBA+}} = \frac{b^2}{4}\operatorname{var}(A_i)$. Assume $A_i = [A_1 > A_2 > \cdots > A_m]$, then the corresponding aggregation probability distribution is $[p_1 > p_2 > \cdots > p_m]$.

We show the analysis of variance with set size 2, while the analysis can be easily extended to the number $K$. For FedEBA+, we have

$$\text{var}(A_i) = \sum_{i=1}^{m} p_i \left( A_i - \sum_i p_i A_i \right)^2 \tag{152}$$

$$= p_1(A_1 - (p_1 A_1 + p_2 A_2))^2 + p_2(A_2 - (p_1 A_1 + p_2 A_2))^2 \tag{153}$$

$$= p_1(1 - p_1)^2 A_1^2 - 2(1 - p_1)p_1 p_2 A_1 A_2 + p_1 p_2^2 A_2^2 \tag{154}$$

$$+ p_2(1 - p_2)^2 A_2^2 - 2(1 - p_2)p_1 p_2 A_1 A_2 + p_1^2 p_2 A_1^2 \tag{155}$$

$$= (p_1 p_2^2 + p_1^2 p_2) A_1^2 - 2p_1 p_2(2 - p_1 - p_2) A_1 A_2 + (p_1 p_2^2 + p_1^2 p_2) A_2^2 \tag{156}$$

$$\overset{(a1)}{=} p_1 p_2(A_1^2 + A_2^2) - 2p_1 p_2 A_1 A_2 \tag{157}$$

$$= p_1 p_2(A_1 - A_2)^2 \,, \tag{158}$$

where $(a1)$ follows from the fact $\sum_i p_i = 1$.

According to our previous analysis, $p_1 > p_2$ while $A_1 > A_2$. According to Cauchy-Schwarz inequality, one can easily prove that $p_1 p_2 \leq \frac{1}{4}$, where $\frac{1}{4}$ comes from uniform aggregation.

Therefore, we prove that $V^{\text{EBA+}} \leq V^{\text{Avg}}$.

I.3    FAIRNESS ANALYSIS BY SMOOTH AND STRONGLY CONVEX LOSS FUNCTIONS.

In this section, we define the test loss on client $i$ as $L(x_i)$, to distinguish it from the training loss $F_i(x_i)$.

To extend the analysis to a more general case, we first introduce the following assumptions:

**Assumption 5** (Smooth and strongly convex loss functions)**.** *The loss function $L_i(x)$ for each client is L-smooth,*

$$\|\nabla L_i(x)\|_2 \leq L \,, \tag{159}$$

*and $\mu$-strongly convex:*

$$L(y) \geq L(x) + <\nabla L(x), y - x> + \frac{1}{2}\mu\|y - x\|^2 \,. \tag{160}$$

The variance of FedAvg with $N$ clients loss can be formulated as:

$$V_N^{Avg} = \frac{1}{N}\sum_{i=1}^{N} L_i^2(x) - \left(\frac{1}{N}\sum_{i=1}^{N} L_i(x)\right)^2. \tag{161}$$

For FedEBA+, the variance can be formulated with a similar form, only different in client's loss $L_i(\tilde{x})$, abbreviated as $\tilde{L}_i$. Then, the variance of FedEBA+ with $N$ clients can be formulated as:

$$V_N^{EBA+} = \frac{1}{N}\sum_{i=1}^{N} \tilde{L}_i^2 - \left(\frac{1}{N}\sum_{i=1}^{N} \tilde{L}_i\right)^2. \tag{162}$$

When client number is $N+1$, abbreviate FedAvg's loss $L_i(x)$ as $L_i$, we conclude

$$V_N^{Avg} \tag{163}$$

$$= \frac{1}{N+1} \sum_{i=1}^{N+1} L_i^2 - \left( \frac{1}{N+1} \sum_{i=1}^{N+1} L_i \right)^2 \tag{164}$$

$$= \frac{N}{N+1} \frac{1}{N} \left( L_1^2 + L_2^2 + \cdots + L_{N+1}^2 \right) - \left[ \frac{N}{N+1} \frac{1}{N} \left( L_1 + L_2 + \cdots + L_{N+1} \right) \right]^2 \tag{165}$$

$$= \frac{N}{N+1} \frac{1}{N} \left[ \left( L_1^2 + L_2^2 + \cdots + L_N^2 \right) + L_{N+1}^2 \right]$$
$$- \left[ \frac{N}{N+1} \left( \frac{L_1 + L_2 + \cdots + L_N}{N} + \frac{L_{N+1}}{N} \right) \right]^2 \tag{166}$$

$$= \left( \frac{N}{N+1} \right)^2 \left[ \frac{N+1}{N} \frac{\sum_{i=1}^N L_i^2}{N} - \left( \frac{1}{N} \sum_{i=1}^N L_i \right)^2 \right]$$
$$+ \frac{1}{N+1} L_{N+1}^2 - \frac{L_{N+1}^2}{(N+1)^2} - 2 (\frac{N}{N+1})^2 \frac{\sum_{i=1}^N L_i}{N} \frac{L_{N+1}}{N} \tag{167}$$

$$= (\frac{N}{N+1})^2 \frac{1}{N} \frac{\sum_{i=1}^N L_i^2}{N} + \frac{N}{N+1} V_N + \frac{1}{N+1} L_{N+1}^2$$
$$- \frac{1}{(N+1)^2} L_{N+1}^2 - 2 (\frac{N}{N+1})^2 \frac{\sum_{i=1}^N L_i}{N} \frac{L_{N+1}}{N} \tag{168}$$

$$= \frac{N}{N+1} V_N + \frac{L_1^2 + \cdots + L_N^2}{(N+1)^2} + + \frac{N L_{N+1}^2}{(N+1)^2} - \frac{2 (L_1 + \cdots + _N) L_{N+1}}{(N+1)^2} \tag{169}$$

$$= \frac{N}{N+1} V_N + \frac{\sum_{i=1}^N (L_i - L_{N+1})^2}{(N+1)^2} . \tag{170}$$

We start proving $V_N^{Avg} \geq V_N^{EBA+}, \forall N$ by considering a special case with two clients: There are two clients, Client 1 and Client 2, each with local model $x_1, x_2$ and training loss $F_1(x_1)$ and $F_2(x_2)$.

In this analysis, we assume Client 2 to be the *outlier*, which means the client's optimal parameter and model parameter distribution is far away from Client 1. In particular, $\mu_2 >> L_{smooth}^1$.

The global model starts with $x = 0$, and after enough local training updates, the model $x_1, x_2$ will converge to their personal optimum $x_1^*, x_2^*$. W.l.o.g, we let Client 1 with $F_1(x_1^*) = 0$, Client 2 with $F_2(x_2^*) = a > 0$. Let $x_1^* < x_2^*$ (relative position, which does not affect the analysis).

Based on the proposed aggregation $p_i \propto \exp \frac{F_i(x)}{\tau}$, we can derive the aggregated global model $\tilde{x}$ of FedEBA+ to be:

$$\tilde{x} = p_1 x_1^* + p_2 x_2^* = \frac{x_1^* + e^a x_2^*}{e^a + 1} . \tag{171}$$

While for FedAvg, the aggregated global model $\overline{x}$ is:

$$\overline{x} = \frac{x_1^* + x_2^*}{2} . \tag{172}$$

For FedEBA+, the test loss of Client 1 and Client 2 are $\tilde{L}_1 = L_1(\tilde{x}), \tilde{L}_2 = L_2(\tilde{x})$ respectively. The corresponding variance is $V_2^{EBA+} = \frac{1}{2} (\tilde{L}_1 - \tilde{L}_2)^2$.

For FedAvg, the test loss of Client 1 and Client 2 is $\overline{L}_1 = L_1(\overline{x}), \overline{L}_2 = L_2(\overline{x})$ respectively. The corresponding variance is $V_2^{AVG} = \frac{1}{2} (\overline{L}_1 - \overline{L}_2)^2$.

Since Client 2 is a outlier with $F_2(x_2^*) > 0$ and $x_1^* < x_2^*$, we can easily conclude $F_2(x)$ is monotonically decreasing on $(x_1^*, x_2^*)$, $F_1(x)$ is monotonically increasing on $(x_1^*, x_2^*)$. Besides, w.l.o.g, since $\nabla F_1(x) \leq L_{smooth} << \mu_2$, we can let $\mu = \frac{a}{x_2^* - x_1^*}$.

Thus, we promise $\frac{a}{x_2^* - x_1^*} > \nabla F_1(x_2^*)$. According to the property of calculus, we can easily check that $F_2(x) - F_1(x) > 0$ is monotonically decreasing on $(x_1^*, x_2^*)$.

Since

$$x_2^* - \tilde{x} = \frac{x_2^* - x_1^*}{e^a + 1} \leq x_2^* - \overline{x} = \frac{x_2^* - x_1^*}{2} \, , \tag{173}$$

thus we have $(F_2(\tilde{x}) - F_1(\tilde{x}))^2 \leq (F_2(\overline{x}) - F_1(\overline{x}))^2$ .

So far, we have prove $V_2^{EBA+} \leq V_2^{AVG}$.

To extend the analysis to arbitrary $N$, we utilize the mathematical induction:

Assume $V_N^{EBA+} \leq V_N^{AVG}$, we need to derive $V_{N+1}^{EBA+} \leq V_{N+1}^{AVG}$.

Consider a similar scenario as we analyze with two clients. We assume Client N+1 to be an outlier, which means the client's optimal value and parameter distribution are far away from other clients. In particular, $\mu_{N+1} >> L_{smooth}^{others}$. W.l.o.g, let the optimal value $F(x_{N+1}^*)$ for Client N+1 be $a$, others to be zero.

Again, the global model starts with $x = 0$, and after enough local training updates, the models will converge to their personal optimum $x_1^*, x_2^*, \ldots, x_{N+1}^*$ and $x_{N+1}^* > x_{others}^*$.

By (170), we have:

$$V_{N+1}^{Avg} = \frac{N}{N+1} V_N^{AVG} + \frac{\sum_{i=1}^{N} (\overline{L}_i - \overline{L}_{N+1})^2}{(N+1)^2} \, , \tag{174}$$

where $\overline{L}_i$ is the test loss of client $i$ after average and

$$V_{N+1}^{EBA+} = \frac{N}{N+1} V_N^{EBA+} + \frac{\sum_{i=1}^{N} (\tilde{L}_i - \tilde{L}_{N+1})^2}{(N+1)^2} \, . \tag{175}$$

Since we know $V_N^{EBA+} \leq V_N^{AVG}$, thus as long as we promise $\frac{\sum_{i=1}^{N} (\tilde{L}_i - \tilde{L}_{N+1})^2}{(N+1)^2} \leq \frac{\sum_{i=1}^{N} (\overline{L}_i - \overline{L}_{N+1})^2}{(N+1)^2}$, we can finish the proof.

Consider an arbitrary client $i \in [1, N]$, since we already know $F_{N+1}(x_{N+1}^*) = a > F_i(x_i^*) = 0$, the expression for $\tilde{x}$ is

$$\tilde{x} = \sum_{i=1}^{N+1} p_i x_i^* = \frac{1}{N + e^a} \sum_{i=1}^{N} x_i^* + \frac{e^a}{N + e^a} x_{N+1}^* \, , \tag{176}$$

While for FedAvg,

$$\overline{x} = \sum_{i=1}^{N+1} \frac{1}{N+1} x_i^* \, . \tag{177}$$

Following the exact analysis on Client $i$ and Client $N+1$, we can conclude that $F_{N+1}(x) - F_i(x) > 0$ is monotonically decreasing on $(x_i^*, x_{N+1}^*)$.

Since

$$x_{N+1}^* - \tilde{x} = \frac{N x_{N+1}^* - \sum_{i=1}^{N} x_i^*}{e^a + N} \leq x_{N+1}^* - \overline{x} = \frac{N x_{N+1}^* - \sum_{i=1}^{N} x_i^*}{e^a + 1} \, , \tag{178}$$

thus we have $(F_{N+1}(\tilde{x}) - F_i(\tilde{x}))^2 \leq (F_{N+1}(\overline{x}) - F_i(\overline{x}))^2 \; \forall i \in [1, \ldots, N]$.

Therefore, we promise $\frac{\sum_{i=1}^N (\tilde{L}_i - \tilde{L}_{N+1})^2}{(N+1)^2} \leq \frac{\sum_{i=1}^N (\overline{L}_i - \overline{L}_{N+1})^2}{(N+1)^2}$.

So far, we have prove $V_{N+1}^{EBA+} \leq V_{N+1}^{AVG}$.

According to the mathematical induction, we prove $V_N^{EBA+} \leq V_N^{AVG}$ for arbitrary client number $N$ under smooth and strongly convex setting.

## J   Pareto-optimality Analysis

In addition to variance, *Pareto-optimality* can serve as another metric to assess fairness, as suggested by several studies (Wei & Niethammer, 2022; Hu et al., 2022). This metric achieves equilibrium by reaching each client's optimal performance without hindering others (Guardieiro et al., 2023). We prove that FedEBA+ achieves Pareto optimality through the entropy-based aggregation strategy.

**Definition J.1** (Pareto optimality)**.** *Suppose we have a group of $m$ clients in FL, and each client $i$ has a performance score $f_i$. Pareto optimality happens when we can't improve one client's performance without making someone else's worse: $\forall i \in [1, m], \exists j \in [1, m], j \neq i$ such that $f_i \leq f'_i$ and $f_j > f'_j$, where $f'_i$ and $f'_j$ represent the improved performance measures of participants $i$ and $j$, respectively.*

In the following proposition, we show that FedEBA+ satisfies Pareto optimality. '

**Proposition J.2** (Pareto optimality.)**.** *The proposed maximum entropy model $\mathbb{H}(p_i)$ is proven to be monotonically increasing under the given constraints, ensuring that the aggregation strategy $\varphi(p) = \arg\max_{p \in \mathcal{P}} h(p(f))$ is Pareto optimal. Here, $p(f)$ is the aggregation weights $p = [p_1, p_2, \ldots, p_m]$ of the loss function $f = [f_1, f_2, \ldots, f_m]$, and $h(\cdot)$ represents the entropy function. The proof can be found in Appendix J.*

In this following, we demonstrate the Proposition J.2. In particular, we consider the degenerate setting of FedEBA+ where the parameter $\alpha = 0$. We first provide the following lemma that illustrates the correlation between Pareto optimality and monotonicity.

**Lemma J.3** (Property 1 in (Sampat & Zavala, 2019).)**.** *The allocation strategy $\varphi(p) = \arg\max_{p \in \mathcal{P}} h(p(f))$ is Pareto optimal if $h$ is a strictly monotonically increasing function.*

In order for this paper to be self-contained, we restate the proof of Property 1 in (Sampat & Zavala, 2019) here:

**Proof Sketch:** We prove the result by contradiction. Consider that $p^* = \varphi(\mathcal{P})$ is not Pareto optimal; thus, there exists an alternative $p \in \mathcal{P}$ such that

$$\sum_i p_i f_i = \frac{\sum_i p_i \log p_i}{Z} \geq \sum_i p_i^* f_i = \frac{\sum_i p_i^* \log p_i^*}{Z}, \tag{179}$$

where $Z > 0$ is a constant. Since $h(p)$ is a strictly monotonically increasing function, we have $h(p) > h(p^*)$. This is a contradiction because $h^*$ maximizes $h(\cdot)$.

According to the above lemma, to show our algorithm achieves Pareto-optimal, we only need to show it is monotonically increasing.

Recall the objective of maximum entropy:

$$\mathbb{H}(p) = -\sum p(x) log(p(x)), \tag{180}$$

subject to certain constraints on the probabilities $p(x)$.

To show that the proposed aggregation strategy is monotonically increasing, we need to prove that if the constraints on the probabilities $p(x)$ are relaxed, then the maximum entropy of the aggregation probability increases.

One way to do this is to use the properties of the logarithm function. The logarithm function is strictly monotonically increasing. This means that for any positive real numbers a and b, if $a \leq b$, then $\log(a) \leq \log(b)$.

Now, suppose that we have two sets of constraints on the probabilities $p(x)$, and that the second set of constraints is a relaxation of the first set. This means that the second set of constraints allows for a larger set of probability distributions than the first set of constraints.

If we maximize the entropy subject to the first set of constraints, we get some probability distribution $p(x)$. If we then maximize the entropy subject to the second set of constraints, we get some probability distribution $q(x)$ such that $p(x) \leq q(x)$ for all x.

Using the properties of the logarithm function and the definition of the entropy, we have:

$$H(p(x)) = -\sum (p(x) \log(p(x))) \tag{181}$$

$$\leq -\sum (p(x) \log(q(x))) \tag{182}$$

$$= -\sum ((p(x)/q(x))q(x) \log(q(x))) \tag{183}$$

$$= H(q(x)) - \sum ((\frac{p(x)}{q(x)} q(x) \log(p(x)/q(x))) \tag{184}$$

$$\leq H(q(x)). \tag{185}$$

This means that the entropy $H(q(x))$ is greater or equal to $H(p(x))$ when the second set of constraints is a relaxation of the first set of constraints. As the entropy increases when the constraints are relaxed, the maximum entropy-based aggregation strategy is monotonically increasing.

Up to this point, we proved that our proposed aggregation strategy is monotonically increasing. Combined with the Lemma J.3, we can prove that equation (4) is Pareto optimal.

## K   Uniqueness of our Aggregation Strategy

In this section, we prove the proposed entropy-based aggregation strategy is unique.

Recall our optimization objective of constrained maximum entropy:

$$H(p(x)) = -\sum (p(x) \log(p(x))), \tag{186}$$

subject to certain contains, which is $\sum_i p_i = 1, p_i \geq 0, \sum_i p_i F_i = \tilde{f}$.

Based on equation 4, and writing the entropy in matrix form, we have:

$$H_{i,j}(p) = \begin{cases} p_i(\frac{F_i}{\tau} - \log \sum e^{F_i/\tau}) = -ap_i & \text{for } i = j \\ 0 & \text{otherwise} \end{cases}, \tag{187}$$

where $a$ is some positive constant.

For every non-zero vector $v$ we have that:

$$v^T H(p)v = \sum_{j \in \mathcal{N}} -ap_i v_j^2 < 0. \tag{188}$$

The Hessian is thus negative definite.

Furthermore, since the constraints are linear, both convex and concave, the constrained maximum entropy function is strictly concave and thus has a unique global maximum.

## L   Experiment Details

### L.1   Experimental Environment

For all experiments, we use NVIDIA GeForce RTX 3090 GPUs. Each simulation trail with 2000 communication rounds and three random seeds.

**Federated Datasets and Models.** We tested the performance of FedEBA+ on five public datasets: MNIST, Fashion MNIST, CIFAR-10, CIFAR-100, and Tiny-ImageNet. We use two methods to split the real datasets into non-iid datasets: (1) following the setting of (Wang et al., 2021), where 100 clients participate in the federated system, and according to the labels, we divide all the data of MNIST, FashionMNIST, CIFAR-10, CIFAR-100 and Tiny-ImageNet into 200 shards separately, and each user randomly picks up 2 shards for local training. (2) we leverage Latent Dirichlet Allocation (LDA) to control the distribution drift with the Dirichlet parameter $\alpha = 0.1$.

As for the model, we use an MLP model with 2 hidden layers on MNIST and Fashion-MNIST, and a CNN model with 2 convolution layers on CIFAR-10, ResNet-18 on CIFAR-100, and MobileNet-v2 on TinyImageNet.

**Baselines** We compared several advanced FL fairness algorithms with FedEBA+, including FedAvg (McMahan et al., 2017), FedSGD (McMahan et al., 2016), AFL (Mohri et al., 2019),q-FFL (Li et al., 2019a),FedMGDA+(Hu et al., 2022),PropFair (Zhang et al., 2023), TERM (Li et al., 2020a), FOCUS (Chu et al., 2023), Ditto (Li et al., 2021),FedFV (Wang et al., 2021), and lp-proj (Lin et al., 2022).

**Hyper-parameters** As shown in Table 3, we tuned some hyper-parameters of baselines to ensure the performance in line with the previous studies and listed parameters used in FedEBA+. All experiments are running over 2000 rounds for the local epoch ($K = 10$) with local batch size $B = 50$ for MNIST and $B = 64$ for CIFAR datasets. The learning rate remains the same for different methods, that is $\eta = 0.1$ on MNIST, Fasion-MNIST, CIFAR-10, $\eta = 0.05$ on Tiny-ImageNet and $\eta = 0.01$ on CIFAR-100 with decay rate $d = 0.999$.

Table 6: Hyperparameters of baselines.

| Algorithm | Hyper-parameters |
|-----------|------------------|
| q-FFL | $q \in \{0.001, 0.01, 0.1, 0.5, 10, 15\}$ |
| PropFair | $M \in \{0.2, 2.0 5.0\}, \epsilon = 0.2$ |
| AFL | $\lambda \in \{0.1, 0.5, 0.7\}$ |
| TERM | $T \in \{0.1, 0.5, 0.8, 1, 5\}$ |
| FedMGDA+ | $\epsilon \in \{0, 0.03, 0.08, 0.1, 1.0\}$ |
| FedProx | $q = \{0.001, 0.001, 0.1, 0.5, 10.0, 15.0\}$ |
| Ditto | $\lambda = \{0.0, 0.5\}$ |
| FOCUS | $\beta = 0.5, cluster = 2$ |
| lp-proj | $localmodeldim = 60, \lambda = 15, p = 1.0$ |
| FedFV | $\alpha \in \{0.1, 0.2, 0.5\}, \tau \in \{0, 1, 10\}$ |
| FedEBA+ | $\tau \in \{0.1, 0.5, 1.0, 5.0, 10.0, 20.0\}, \alpha \in \{0.0, 0.1, 0.5, 0.9\}$ |

## M  ADDITIONAL EXPERIMENT RESULTS

### M.1  FAIRNESS EVALUATION OF FEDEBA+

In this section, we provide additional experimental results to illustrate that FedEBA+ is superior to other baselines.

Figure 4 illustrates that, on the MNIST dataset, FedEBA+ demonstrates faster convergence, increased stability, and superior results in comparison to baselines. As for the CIFAR-10 dataset, its complexity causes some instability for all methods, however, FedEBA+ still concludes the training with the most favorable fairness results.

**Table 9 shows FedEBA+ outperforms other baselines on CIFAR-10 using MLP model.** The results in Table 9 demonstrate that 1) FedEBA+ consistently achieves a smaller variance of accuracy compared to other baselines, thus is fairer. 2) FedEBA+ significantly improves the performance of the worst 5% clients and 3) FedEBA+ performances steady in terms of best 5% clients. A significant improvement in worst 5% is achieved with relatively no compromise in best 5 %, thus is fairer.

Table 7: **Performance of algorithms on FashionMNIST and CIFAR-10.** We report the accuracy of global model, variance fairness, worst 5%, and best 5% accuracy. The data is divided into 100 clients, with 10 clients sampled in each round. All experiments are running over 2000 rounds for a single local epoch ($K = 10$) with local batch size $= 50$, and learning rate $\eta = 0.1$. The reported results are averaged over 5 runs with different random seeds. We highlight the best and the second-best results by using **bold font** and blue text.

| Algorithm | FashionMNIST (MLP) | | | | CIFAR-10 (CNN) | | | |
|---|---|---|---|---|---|---|---|---|
| | Global Acc. | Var. | Worst 5% | Best 5% | Global Acc. | Var. | Worst 5% | Best 5% |
| FedAvg | $86.49 \pm 0.09$ | $62.44\pm4.55$ | $71.27\pm1.14$ | $95.84\pm 0.35$ | $67.79\pm0.35$ | $103.83\pm10.46$ | $45.00\pm2.83$ | $85.13\pm0.82$ |
| q-FFL$|_{q=0.001}$ | $87.05\pm 0.25$ | $66.67\pm 1.39$ | $72.11\pm 0.03$ | $95.09\pm 0.71$ | $68.53\pm 0.18$ | $97.42\pm 0.79$ | $48.40\pm 0.60$ | $84.70\pm 1.31$ |
| q-FFL$|_{q=0.01}$ | $86.62\pm 0.03$ | $58.11\pm 3.21$ | $71.36\pm 1.98$ | $95.29\pm0.27$ | $68.85\pm 0.03$ | $95.17\pm 1.85$ | $48.20\pm0.80$ | $84.10\pm0.10$ |
| q-FFL$|_{q=0.5}$ | $86.57\pm 0.19$ | $54.91\pm 2.82$ | $70.88\pm 0.98$ | $95.06\pm0.17$ | $68.76\pm 0.22$ | $97.81\pm 2.18$ | $48.33\pm0.84$ | $84.51\pm1.33$ |
| q-FFL$|_{q=10.0}$ | $77.29\pm 0.20$ | $47.20\pm 0.82$ | $61.99\pm 0.48$ | $92.25\pm0.57$ | $40.78\pm 0.06$ | $85.93\pm 1.48$ | $22.70\pm0.10$ | $56.40\pm0.21$ |
| q-FFL$|_{q=15.0}$ | $75.77\pm0.42$ | $46.58\pm0.75$ | $61.63\pm0.46$ | $89.60\pm0.42$ | $36.89\pm0.14$ | $79.65\pm5.17$ | $19.30\pm0.70$ | $51.30\pm0.09$ |
| FedMGDA+$|_{\epsilon=0.0}$ | $86.01\pm0.31$ | $58.87\pm3.23$ | $71.49\pm0.16$ | $95.45\pm0.43$ | $67.16\pm0.33$ | $97.33\pm1.68$ | $46.00\pm0.79$ | $83.30\pm0.10$ |
| FedMGDA+$|_{\epsilon=0.03}$ | $84.64\pm0.25$ | $57.89\pm6.21$ | $73.49\pm1.17$ | $93.22\pm0.20$ | $65.19\pm0.87$ | $89.78\pm5.87$ | $48.84\pm1.12$ | $81.94\pm0.67$ |
| FedMGDA+$|_{\epsilon=0.08}$ | $84.90\pm0.34$ | $61.55\pm5.87$ | $\mathbf{73.64\pm0.85}$ | $92.78\pm0.12$ | $65.06\pm0.69$ | $93.70\pm14.10$ | $48.23\pm0.82$ | $82.01\pm0.09$ |
| AFL$|_{\lambda=0.7}$ | $85.14\pm0.18$ | $57.39\pm6.13$ | $70.09\pm0.69$ | $95.94\pm0.09$ | $66.21\pm1.21$ | $79.75\pm1.25$ | $47.54\pm0.61$ | $82.08\pm0.77$ |
| AFL$|_{\lambda=0.5}$ | $84.14\pm0.18$ | $90.76\pm3.33$ | $60.11\pm0.58$ | $96.00\pm0.09$ | $65.11\pm2.44$ | $86.19\pm9.46$ | $44.73\pm3.90$ | $82.10\pm0.62$ |
| AFL$|_{\lambda=0.1}$ | $84.91\pm0.71$ | $69.39\pm6.50$ | $69.24\pm0.35$ | $95.39\pm0.72$ | $65.63\pm0.54$ | $88.74\pm3.39$ | $47.29\pm0.30$ | $82.33\pm0.41$ |
| PropFair$|_{M=0.2,thres=0.2}$ | $85.51\pm0.28$ | $75.27\pm5.38$ | $63.60\pm0.53$ | $97.60\pm0.19$ | $65.79\pm0.53$ | $79.67\pm5.71$ | $49.88\pm0.93$ | $82.40\pm0.40$ |
| PropFair$|_{M=5.0,thres=0.2}$ | $84.59\pm1.01$ | $85.31\pm8.62$ | $61.40\pm0.55$ | $96.40\pm0.29$ | $66.91\pm1.43$ | $78.90\pm6.48$ | $50.16\pm0.56$ | $85.40\pm0.34$ |
| TERM$|_{T=0.1}$ | $84.31\pm0.38$ | $73.46\pm2.06$ | $68.23\pm0.10$ | $94.16\pm0.16$ | $65.41\pm0.37$ | $91.99\pm2.69$ | $49.08\pm0.66$ | $81.98\pm0.19$ |
| TERM$|_{T=0.5}$ | $82.19\pm1.41$ | $87.82\pm2.62$ | $62.11\pm0.71$ | $93.25\pm0.39$ | $61.04\pm1.96$ | $96.78\pm7.67$ | $42.45\pm1.73$ | $80.06\pm0.62$ |
| TERM$|_{T=0.8}$ | $81.33\pm1.21$ | $95.65\pm9.56$ | $56.41\pm0.56$ | $92.88\pm0.70$ | $59.21\pm1.45$ | $82.63\pm3.64$ | $41.33\pm0.68$ | $77.39\pm1.04$ |
| FedFV$|_{\alpha=0.1,\tau_{fv}=1}$ | $86.51\pm0.28$ | $49.73\pm2.26$ | $71.33\pm1.16$ | $95.89\pm0.23$ | $68.94\pm0.27$ | $90.84\pm2.67$ | $50.53\pm4.33$ | $86.00\pm1.23$ |
| FedFV$|_{\alpha=0.2,\tau_{fv}=0}$ | $86.42\pm0.38$ | $52.41\pm5.94$ | $71.22\pm1.35$ | $95.47\pm0.43$ | $68.89\pm0.15$ | $82.99\pm3.10$ | $50.08\pm0.40$ | $86.24\pm1.17$ |
| FedFV$|_{\alpha=0.5,\tau_{fv}=10}$ | $86.88\pm0.26$ | $47.63\pm1.79$ | $71.49\pm0.39$ | $95.62\pm0.29$ | $69.42\pm0.60$ | $78.10\pm3.62$ | $52.80\pm0.34$ | $85.76\pm0.80$ |
| FedFV$|_{\alpha=0.1,\tau_{fv}=10}$ | $86.98\pm0.45$ | $56.63\pm1.85$ | $66.40\pm0.57$ | $\mathbf{98.80\pm0.12}$ | $71.10\pm0.44$ | $86.50\pm7.36$ | $49.80\pm0.72$ | $\mathbf{88.42\pm0.25}$ |
| FedEBA+$|_{\alpha=0,\tau=0.1}$ | $86.70\pm0.11$ | $50.27\pm5.60$ | $71.13\pm0.69$ | $95.47\pm0.27$ | $69.38\pm0.52$ | $89.49\pm10.95$ | $50.40\pm1.72$ | $86.07\pm0.90$ |
| FedEBA+$|_{\alpha=0.5,\tau=0.1}$ | $87.21\pm0.06$ | $\mathbf{40.02\pm1.58}$ | $73.07\pm1.03$ | $95.81\pm0.14$ | $72.39\pm0.47$ | $70.60\pm3.19$ | $55.27\pm1.18$ | $86.27\pm1.16$ |
| FedEBA+$|_{\alpha=0.9,\tau=0.1}$ | $\mathbf{87.50\pm0.19}$ | $43.41\pm4.34$ | $72.07\pm1.47$ | $95.91\pm0.19$ | $\mathbf{72.75\pm0.25}$ | $\mathbf{68.71\pm4.39}$ | $\mathbf{55.80\pm1.28}$ | $86.93\pm0.52$ |

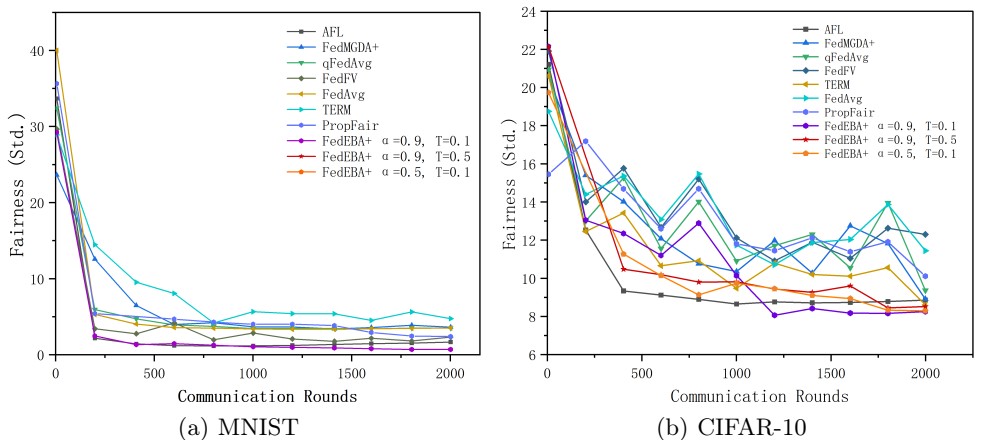

(a) MNIST

(b) CIFAR-10

Figure 4: **Performance of all the methods in terms of Fairness (Var.).**

## M.2 Fairness Evaluation in Different Non-i.i.d. Cases

We adopt two kinds of data splitation strategies to change the degree of non-i.i.d., which are data devided by labels mentioned in the main text, and the data partitioning in deference to the Latent Dirichlet Allocation (LDA) with the Dirichlet parameter . Based on FedAvg, we have experimented with various data segmentation strategies for FedEBA+ to verify the performance of FedEBA+ for scenarios with different kinds of data held by clients.

Table 8: **Ablation study for $\theta$ of FedEBA+.** This table shows our schedule of using the fair angle $\theta$ to control the gradient alignment times is effective, as it largely reduces the communication rounds with larger angles. In addition, compared with the results of baseline in Table 1, the results illustrate that our algorithm remains effective when we increase the fair angle. The additional cost is computed by Additional communication/total communications, the communication cost of communicating the MLP model is 7.8MB/round, the CNN model is 30.4MB/round.

| Algorithm | FashionMNIST (MLP) | | | CIFAR-10 (CNN) | | |
|---|---|---|---|---|---|---|
| | Global Acc. | Var. | Additional cost | Global Acc. | Var. | Additional cost |
| FedAvg | $86.49 \pm 0.09$ | $62.44 \pm 4.55$ | - | $67.79 \pm 0.35$ | $103.83 \pm 10.46$ | - |
| q-FFL | $87.05 \pm 0.25$ | $66.67 \pm 1.39$ | - | $68.53 \pm 0.18$ | $97.42 \pm 0.79$ | - |
| FedMGDA+ | $84.64 \pm 0.25$ | $57.89 \pm 6.21$ | - | $67.16 \pm 0.33$ | $97.33 \pm 1.68$ | - |
| AFL | $85.14 \pm 0.18$ | $57.39 \pm 6.13$ | - | $66.21 \pm 1.21$ | $79.75 \pm 1.25$ | - |
| PropFair | $85.51 \pm 0.28$ | $75.27 \pm 5.38$ | - | $65.79 \pm 0.53$ | $79.67 \pm 5.71$ | - |
| TERM | $84.31 \pm 0.38$ | $73.46 \pm 2.06$ | - | $65.41 \pm 0.37$ | $91.99 \pm 2.69$ | - |
| FedFV | $86.98 \pm 0.45$ | $56.63 \pm 1.85$ | - | $71.10 \pm 0.44$ | $86.50 \pm 7.36$ | - |
| FedEBA+ | | | | | | |
| $\theta = 0°$ | $87.50 \pm 0.19$ | $43.41 \pm 4.34$ | 50.0% | $72.75 \pm 0.25$ | $68.71 \pm 4.39$ | 50.0% |
| $\theta = 15°$ | $87.14 \pm 0.12$ | $43.95 \pm 5.12$ | 48.6% | $71.92 \pm 0.33$ | $75.95 \pm 4.72$ | 26.2% |
| $\theta = 30°$ | $86.96 \pm 0.06$ | $46.82 \pm 1.21$ | 37.7% | $70.91 \pm 0.46$ | $70.97 \pm 4.88$ | 12.7% |
| $\theta = 45°$ | $86.94 \pm 0.26$ | $46.63 \pm 4.38$ | 4.2% | $70.24 \pm 0.08$ | $79.51 \pm 2.88$ | 0.2% |
| $\theta = 90°$ | $86.78 \pm 0.47$ | $48.91 \pm 3.62$ | 0% | $70.14 \pm 0.27$ | $79.43 \pm 1.45$ | 0% |

Table 9: Performance of algorithms on CIFAR-10 using MLP. We report the global model's accuracy, fairness of accuracy, worst 5% and best 5% accuracy. All experiments are running over 2000 rounds for a single local epoch ($K = 10$) with local batch size $= 50$, and learning rate $\eta = 0.1$. The reported results are averaged over 5 runs with different random seeds. We highlight the best and the second-best results by using bold font and blue text.

| Method | Global Acc. | Std. | Worst 5% | Best 5% |
|---|---|---|---|---|
| FedAvg | $46.85 \pm 0.65$ | $12.57 \pm 1.50$ | $19.84 \pm 6.55$ | $69.28 \pm 1.17$ |
| q-FFL$|_{q=0.1}$ | $47.02 \pm 0.89$ | $13.16 \pm 1.84$ | $18.72 \pm 6.94$ | $70.16 \pm 2.06$ |
| q-FFL$|_{q=0.2}$ | $46.91 \pm 0.90$ | $13.09 \pm 1.84$ | $18.88 \pm 7.00$ | $70.16 \pm 2.10$ |
| q-FFL$|_{q=1.0}$ | $46.79 \pm 0.73$ | $11.72 \pm 1.00$ | $22.80 \pm 3.39$ | $68.00 \pm 1.60$ |
| q-FFL$|_{q=2.0}$ | $46.36 \pm 0.38$ | $10.85 \pm 0.76$ | $24.64 \pm 2.17$ | $66.80 \pm 2.02$ |
| q-FFL$|_{q=5.0}$ | $45.25 \pm 0.42$ | $9.59 \pm 0.36$ | $26.56 \pm 1.03$ | $63.60 \pm 1.13$ |
| Ditto$|_{\lambda=0.0}$ | $52.78 \pm 1.23$ | $10.17 \pm 0.24$ | $31.80 \pm 2.27$ | $71.47 \pm 1.20$ |
| Ditto$|_{\lambda=0.5}$ | $53.77 \pm 1.02$ | $8.89 \pm 0.32$ | $36.27 \pm 2.81$ | $71.27 \pm 0.52$ |
| AFL$|_{\lambda=0.01}$ | $52.69 \pm 0.19$ | $10.57 \pm 0.37$ | $34.00 \pm 1.30$ | $71.33 \pm 0.57$ |
| AFL$|_{\lambda=0.1}$ | $52.68 \pm 0.46$ | $10.64 \pm 0.14$ | $33.27 \pm 1.75$ | $\mathbf{71.53 \pm 0.52}$ |
| TERM$|_{T=1.0}$ | $45.14 \pm 2.25$ | $9.12 \pm 0.35$ | $27.07 \pm 3.49$ | $62.73 \pm 1.37$ |
| FedMGDA+$|_{\epsilon=0.01}$ | $45.65 \pm 0.21$ | $10.94 \pm 0.87$ | $25.12 \pm 2.34$ | $67.44 \pm 1.20$ |
| FedMGDA+$|_{\epsilon=0.05}$ | $45.58 \pm 0.21$ | $10.98 \pm 0.81$ | $25.12 \pm 1.87$ | $67.76 \pm 2.27$ |
| FedMGDA+$|_{\epsilon=0.1}$ | $45.52 \pm 0.17$ | $11.32 \pm 0.86$ | $24.32 \pm 2.24$ | $68.48 \pm 2.68$ |
| FedMGDA+$|_{\epsilon=0.5}$ | $45.34 \pm 0.21$ | $11.63 \pm 0.69$ | $24.00 \pm 1.93$ | $68.64 \pm 3.11$ |
| FedMGDA+$|_{\epsilon=1.0}$ | $45.34 \pm 0.22$ | $11.64 \pm 0.66$ | $24.00 \pm 1.93$ | $68.64 \pm 3.11$ |
| FedFV$|_{\alpha=0.1,\tau_{fv}=1}$ | $54.28 \pm 0.37$ | $9.25 \pm 0.42$ | $35.25 \pm 1.01$ | $71.13 \pm 1.37$ |
| FedEBA$|_{\alpha=0.9,\tau=0.1}$ | $53.94 \pm 0.13$ | $9.25 \pm 0.95$ | $35.87 \pm 1.80$ | $69.93 \pm 1.00$ |
| FedEBA+$|_{\alpha=0.5,\tau=0.1}$ | $53.14 \pm 0.05$ | $8.48 \pm 0.32$ | $36.03 \pm 2.08$ | $69.20 \pm 0.75$ |
| FedEBA+$|_{\alpha=0.9,\tau=0.1}$ | $\mathbf{54.43 \pm 0.24}$ | $\mathbf{8.10 \pm 0.17}$ | $\mathbf{40.07 \pm 0.57}$ | $69.80 \pm 0.16$ |

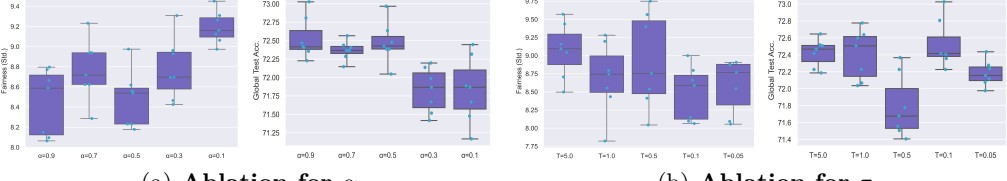

(a) **Ablation for $\alpha$**  (b) **Ablation for $\tau$**

Figure 5: Ablation study for hyperparameters

Table 10: Performance of algorithms+momentum on Fashion-MNIST to show that FedEBA+ is orthogonal to advance optimization methods like momentum (Karimireddy et al., 2020a), allowing seamless integration. All experiments are running over 2000 rounds on the MLP model for a single local epoch ($K = 10$) with local batch size = 50, global momentum = 0.9 and learning rate $\eta = 0.1$. The reported results are averaged over 5 runs with different random seeds. We highlight the best and the second-best results by using bold font and blue text.

| Method | Global Acc. | Var. | Worst 5% | Best 5% |
|--------|-------------|------|----------|---------|
| FedAvg | $86.68\pm 0.37$ | $66.15\pm 3.23$ | $72.18\pm 0.22$ | $96.04\pm\pm 0.35$ |
| $\text{AFL}|_{\lambda=0.05}$ | $79.68\pm 0.91$ | $55.00\pm 3.34$ | $66.67\pm 0.12$ | $94.00\pm 0.08$ |
| $\text{AFL}|_{\lambda=0.7}$ | $85.41\pm 0.30$ | $63.42\pm\pm 1.55$ | $\mathbf{73.83\pm 0.37}$ | $96.46\pm 0.12$ |
| $\text{q-FFL}|_{q=0.01}$ | $86.82\pm 0.20$ | $64.11\pm 2.17$ | $71.08\pm 0.16$ | $96.29\pm 0.08$ |
| $\text{q-FFL}|_{q=15}$ | $79.59\pm 0.48$ | $62.26\pm 2.88$ | $66.33\pm 1.14$ | $90.07\pm 0.98$ |
| $\text{FedMGDA+}|_{\epsilon=0.0}$ | $82.69\pm 0.52$ | $65.26\pm 3.81$ | $69.63\pm 1.20$ | $92.67\pm 0.54$ |
| $\text{PropFair}|_{M=5,thres=0.2}$ | $85.67\pm 0.19$ | $73.44\pm 2.44$ | $64.59\pm 0.42$ | $97.47\pm 0.11$ |
| $\text{FedProx}|_{\mu=0.1}$ | $86.76\pm 0.26$ | $60.69\pm 3.07$ | $72.67\pm 0.29$ | $95.96\pm 0.14$ |
| $\text{TERM}|_{T=0.1}$ | $84.58\pm 0.28$ | $76.44\pm 2.50$ | $69.52\pm 0.36$ | $94.04\pm 0.50$ |
| $\text{FedFV}|_{\alpha=0.1,\tau=10}$ | $87.46\pm 0.18$ | $58.35\pm 1.89$ | $67.71\pm 0.56$ | $\mathbf{97.79\pm 0.18}$ |
| $\text{FedEBA+}|_{\alpha=0.9,T=0.1}$ | $\mathbf{87.67\pm 0.28}$ | $\mathbf{46.67\pm 1.09}$ | $71.90\pm 0.70$ | $96.26\pm 0.03$ |

Table 11: Performance of algorithms+VARP on Fashion-MNIST to show that FedEBA+ is orthogonal to advance optimization methods like VARP (Jhunjhunwala et al., 2022), allowing seamless integration. All experiments are running over 2000 rounds on the MLP model for a single local epoch ($K = 10$) with local batch size = 50, global learning rate = 1.0 and client learning rate = 0.1. The reported results are averaged over 5 runs with different random seeds. We highlight the best and the second-best results by using bold font and blue text.

| Method | Global Acc. | Var. | Worst 5% | Best 5% |
|--------|-------------|------|----------|---------|
| FedAvg (FedVARP) | $87.12\pm 0.08$ | $59.96\pm 2.48$ | $72.45\pm 0.26$ | $96.09\pm\pm 0.27$ |
| $\text{q-FFL}|_{q=0.01}$ | $86.73\pm 0.31$ | $62.89\pm 2.67$ | $73.55\pm 0.11$ | $95.54\pm 0.14$ |
| $\text{q-FFL}|_{q=15}$ | $78.98\pm 0.63$ | $58.28\pm 1.95$ | $67.12\pm 0.97$ | $88.42\pm 0.67$ |
| $\text{FedFV}|_{\alpha=0.1,\tau=10}$ | $87.28\pm 0.10$ | $57.90\pm 1.77$ | $67.41\pm 0.30$ | $\mathbf{97.66\pm 0.06}$ |
| $\text{FedEBA+}|_{\alpha=0.9,T=0.1}$ | $\mathbf{87.45\pm 0.18}$ | $\mathbf{49.91\pm 2.38}$ | $71.44\pm 0.64$ | $95.94\pm 0.09$ |

Table 12: **Ablation study for Dirichlet parameter $\alpha$.** Performance comparison between FedAvg and FedEBA+ on CIFAR-100 using ResNet18 (devided by Dirichlet Distribution with $\alpha \in \{0.1, 0.5, 1.0\}$). We report the global model's accuracy, fairness of accuracy, worst 5% and best 5% accuracy. All experiments are running over 2000 rounds for a single local epoch ($K = 10$) with local batch size = 64, and learning rate $\eta = 0.01$. The reported results are averaged over 5 runs with different random seeds.

| Algorithm | Global Acc. | | | Var. | | | Worst 5% | | | Best 5% | | |
|-----------|-------------|---|---|------|---|---|----------|---|---|---------|---|---|
| | $\alpha = 0.1$ | $\alpha = 0.5$ | $\alpha = 1.0$ | $\alpha = 0.1$ | $\alpha = 0.5$ | $\alpha = 1.0$ | $\alpha = 0.1$ | $\alpha = 0.5$ | $\alpha = 1.0$ | $\alpha = 0.1$ | $\alpha = 0.5$ | $\alpha = 1.0$ |
| FedAvg | $30.94\pm0.04$ | $54.69\pm0.25$ | $64.91\pm0.02$ | $17.24\pm0.08$ | $7.92\pm0.03$ | $5.18\pm0.06$ | $0.20\pm0.00$ | $38.79\pm0.24$ | $54.36\pm0.11$ | $65.90\pm1.48$ | $70.10\pm0.25$ | $75.43\pm0.39$ |
| FedEBA+ | $33.39\pm0.22$ | $58.55\pm0.41$ | $65.98\pm0.04$ | $16.92\pm0.04$ | $7.71\pm0.08$ | $4.44\pm0.10$ | $0.95\pm0.15$ | $41.63\pm0.16$ | $58.20\pm0.17$ | $68.51\pm0.21$ | $74.03\pm0.07$ | $74.96\pm0.16$ |

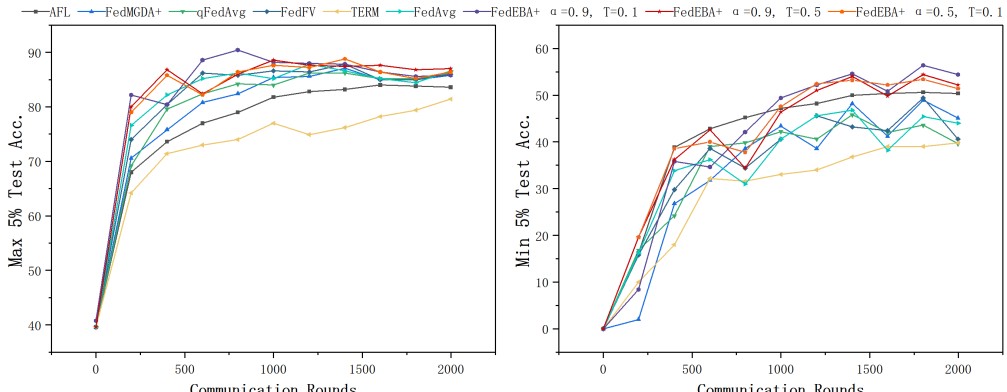

Figure 6: **The maximum and minimum 5% performance of all baselines and FedEBA+ on CIFAR-10.**

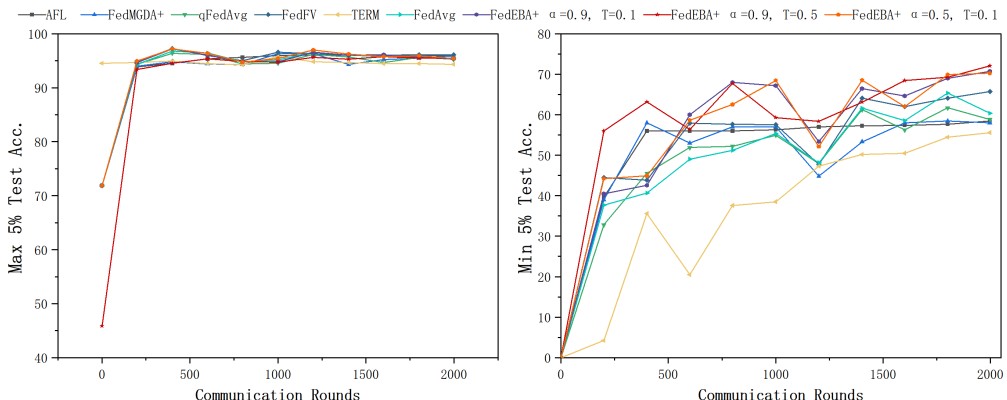

Figure 7: **The maximum and minimum 5% performance of all baselines and FedEBA+ on FashionMNSIT.**

### M.3    GLOBAL ACCURACY EVALUATION OF FedEBA+

We run all methods on the CNN model, regarding the CIFAR-10 figure. Under different hyper-parameters, FedEBA+ can reach a stable high performance of worst 5% while guaranteeing best 5%, as shown in Figure 6. As for FashionMNIST using MLP model, the worst 5% and best 5% performance of FedEBA+ are similar to that of CIFAR-10. We can see that FedEBA+ has a more significant lead in worst 5% with almost no loss in best 5%, as shown in Figure 7.

### M.4    ROBUSTNESS EVALUATION TO NOISY LABEL SCENARIO

The local noisy label follows the symmetric flipping approach introduced in Jiang et al. (2022); Fang & Ye (2022), with a noise ratio of $\epsilon$ set to 0.5. All the other settings like the learning rate keep the same. Specifically, we employ the MLP model for Fashion-MNIST and the CNN model for CIFAR-10.

The results of Table 13 reveal that (1) FedEBA+ maintains its superiority in accuracy and fairness even when there are local noisy labels; (2) FedEBA+ can be integrated with established approaches for addressing local noisy labels, consistently outperforming other algorithms combined with existing methods in terms of both fairness and accuracy.

Table 13: **Performance of algorithms on local noisy label scenario.** We evaluate the effectiveness of FedEBA+ when incorporating local noisy labels on both the FashionMNIST dataset with an MLP model and the CIFAR-10 dataset with a CNN model, using a noise ratio of $\epsilon = 0.5$.

| Algorithm | FashionMNIST | | | | CIFAR-10 | | | |
|---|---|---|---|---|---|---|---|---|
| | Global Acc. ↑ | Std. ↓ | Worst 5% ↑ | Best 5% ↑ | Global Acc. ↑ | Std. ↓ | Worst 5% ↑ | Best 5% ↑ |
| FedAvg | 80.59±0.42 | 57.34±2.98 | 65.40±0.43 | 94.87±0.25 | 33.45±0.89 | 38.03±2.30 | 21.67±0.96 | 46.27±1.65 |
| q-FFL | 79.85±0.31 | 68.00±4.34 | 64.13±0.75 | 95.47±0.19 | 30.83±0.76 | 44.46±2.76 | 17.21±1.03 | 44.33±0.19 |
| AFL | 80.34±0.35 | 57.35±6.06 | 65.60±2.01 | 95.00±0.91 | 32.64±0.33 | 35.58±3.17 | 20.47±0.82 | 44.80±1.61 |
| FedFV | 63.08±0.88 | 88.95±3.06 | 46.13±0.77 | 83.13±1.52 | 34.28±0.39 | 41.07±0.77 | 21.13±0.90 | 46.60±0.33 |
| FOCUS | 80.79±0.27 | 58.61±3.61 | 64.40±1.85 | 94.80±0.62 | 26.81±1.22 | 14.04±0.68 | 6.84±1.58 | 56.69±1.22 |
| FedEBA+ | 82.03±0.42 | 49.23±7.21 | 67.67±1.06 | 95.27±0.81 | 35.04±0.21 | 34.60±3.69 | 23.07±1.24 | 47.80±1.23 |
| FedAvg + LSR | 84.36±0.07 | 57.80±5.71 | 69.20±0.75 | 96.87±0.34 | 58.90±0.42 | 80.80±8.73 | 40.80±0.75 | 76.93±1.24 |
| q-FFL + LSR | 84.23±0.08 | 63.69±1.62 | 64.73±0.09 | 96.87±0.41 | 58.91±0.75 | 86.32±10.20 | 41.33±0.90 | 77.60±2.73 |
| FedEBA+ + LSR | 85.30±0.12 | 54.10±4.13 | 67.93±0.62 | 96.80±0.28 | 61.21±0.88 | 64.73±0.97 | 43.40±1.72 | 75.53±2.05 |

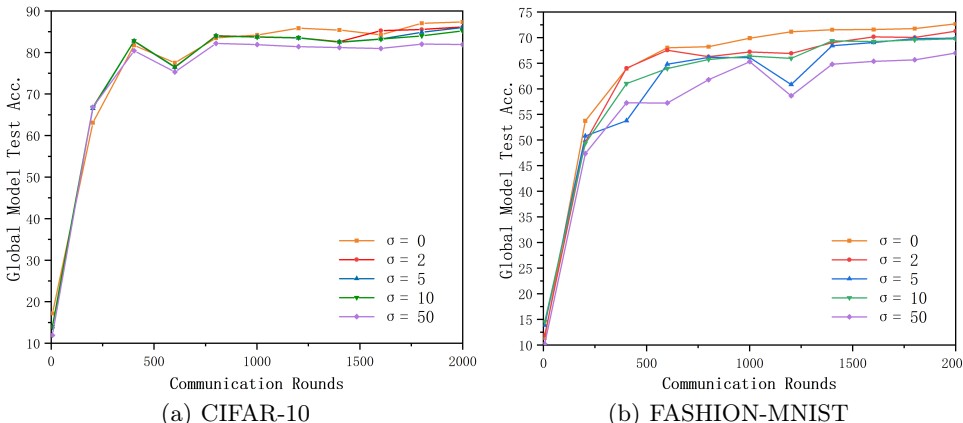

(a) CIFAR-10      (b) FASHION-MNIST

Figure 8: Privacy Evaluation of FedEBA+.

## M.5 PRIVACY EVALUATION.

We also evaluate FedEBA+ under privacy preservation. Following Abadi et al. (2016), we insert Gaussian noise into the intermediate regularization variable $\delta$ with noise standard deviation $\sigma_2 : \tilde{\sigma}_i \leftarrow \sigma_i + \frac{1}{L}\mathcal{N}(0, \sigma_2^2 C_0^2 I)$, where $L$ is the batch size, $\sigma_2$ is the noise parameter, $C_2$ is the clipping constant. The result is shown in Figure 8. With $\sigma_2 \leq 5$, the curves show only marginal reductions without significant performance degradation. However, higher values of $\sigma_2$ risk compromising performance. This suggests that our approach is compatible with a specific threshold of privacy preservation. In addition, Table 14 shows that compared to other fairness baselines, FedEBA+ maintains its fairness and performance advantage when using differential privacy.

## M.6 ABLATION STUDY

**Remark M.1** (The annealing manner for $\tau$). *While we set $\tau$ as a constant in our algorithm, we demonstrate that utilizing an annealing schedule for $\tau$ can further enhance performance. The linear annealing schedule is defined below:*

$$\tau^T = \tau^0/(1 + \kappa(T - 1)), \tag{189}$$

*where $T$ is the total communication rounds and hyperparameter $\kappa$ controls the decay rate. There are also concave schedule $\tau^k = \tau^0/(1 + \kappa(T - 1))^{\frac{1}{2}}$ and convex schedule $\tau^k = \tau^0/(1 + \kappa(T - 1))^3$. We experiment with different annealing strategies for $\tau$ in Figure 9.*

For the annealing schedule of $\tau$ mentioned above, Figure 9 shows that the annealing schedule has advantages in reducing the variance compared with constant $\tau$. Besides, the global accuracy is robust to the annealing strategy, and the annealing strategy is robust to the initial temperature $T_0$.

Table 14: Performance of fairness algorithms under different differential privacy noise $\sigma$.

| noise $\sigma_2$ | Fashion-MNIST | | | | CIFAR10 | | | |
|---|---|---|---|---|---|---|---|---|
| | Global Acc. | Var. | Worst 5% | Best 5% | Global Acc. | Var. | Worst 5% | Best 5% |
| | FedEBA+ | | | | | | | |
| 0 | 87.50±0.19 | 43.41±4.34 | 72.07±1.47 | 95.91±0.19 | 72.75±0.25 | 68.71±4.39 | 55.80±1.28 | 86.93±0.52 |
| 2 | 86.24±0.14 | 75.67±3.40 | 63.67±0.74 | 97.9±0.22 | 70.69±0.40 | 76.25±3.56 | 51.87±0.25 | 86.5±0.24 |
| 5 | 86.01±0.08 | 73.11±2.62 | 64.90±0.94 | 98.0±0.16 | 69.86±0.14 | 76.4±2.38 | 51.20±0.11 | 85.15±0.45 |
| 10 | 85.96±0.08 | 71.52±2.45 | 64.8±1.85 | 97.53±0.34 | 69.48±0.32 | 85.53±2.10 | 49.93±0.77 | 84.53±0.62 |
| 50 | 83.43±0.14 | 79.7±1.18 | 61.37±1.52 | 97.00±0.59 | 67.57±0.68 | 120.83±2.80 | 45.40±0.99 | 86.17±0.33 |
| | FedAvg | | | | | | | |
| 0 | 86.49±0.09 | 62.44±4.55 | 71.27±1.14 | 95.84±0.35 | 67.79±0.35 | 103.83±10.46 | 45.00±2.83 | 85.13±0.82 |
| 2 | 64.20±0.22 | 534.40±1.24 | 7.4±0.2 | 93.2±0 | 45.29±0.81 | 101.04±9.70 | 23.4±0.10 | 68.2±0.33 |
| 5 | 64.14±0.02 | 536.57±2.72 | 7.4±0 | 93.1±0.13 | 45.01±0.33 | 98.38±5.24 | 26.4±1.5 | 66.2±1.2 |
| 10 | 64.10±0.13 | 533.34±4.26 | 7.2±0 | 93.0±0 | 45.45±0.62 | 97.50±4.93 | 26.6±2.2 | 68.0±1.4 |
| 50 | 64.06±0.05 | 533.61±2.40 | 7.55±0.16 | 93.1±0.10 | 45.27±0.92 | 100.54±6.23 | 26.5±1.33 | 66.4±1.4 |
| | qFedAvg | | | | | | | |
| 0 | 86.57±0.19 | 54.91±2.82 | 70.88±0.98 | 95.06±0.17 | 68.76± 0.22 | 97.81±2.18 | 48.33±0.84 | 84.51±1.33 |
| 2 | 64.17±0.02 | 529.99±0.92 | 7.8±0 | 93.2±0 | 43.79±0.70 | 187.79±2.03 | 16.8±0 | 76.14±2.32 |
| 5 | 64.16±0.04 | 530.55±1.17 | 7.6±0 | 93.2±0 | 44.50±0.78 | 191.12±1.70 | 15.4±1.14 | 73.8±1.28 |
| 10 | 64.15±0.03 | 526.82±0.67 | 7.6±0 | 93.2±0 | 43.42±0.80 | 200.31±2.80 | 14.33±1.24 | 73.8±1.14 |
| 50 | 64.21±0.07 | 529.58±0.50 | 7.6±0 | 93.2±0 | 43.92±0.92 | 195.69±3.07 | 15.88±1.30 | 74.2±0.84 |
| | FedMGDA+ | | | | | | | |
| 0 | 84.64±0.25 | 57.89±6.21 | 73.49±1.17 | 93.22±0.20 | 65.19±0.87 | 89.78±5.87 | 48.84±1.12 | 81.94±0.67 |
| 2 | 79.34±0.06 | 112.12±1.49 | 56.67±0.25 | 95.13±0.09 | 43.84±0.22 | 183.39±3.17 | 14.60±1.2 | 70.40±0.4 |
| 5 | 77.13±0.15 | 136.19±1.20 | 51.8±0.40 | 95.00±0.22 | 41.39±0.63 | 96.67±2.88 | 23.2±0.6 | 62.00±0.2 |
| 10 | 71.02±0.01 | 248.45±2.18 | 36.7±0.7 | 93.2±0.13 | 36.75±0.45 | 107.94±4.10 | 16.2±0.34 | 57.00±4.0 |
| 50 | 57.04±0.03 | 754.46±0.81 | 0.2±0 | 93.9±0.1 | 23.08±0.05 | 203.65±3.6 | 0.40±0 | 56.4±0.43 |

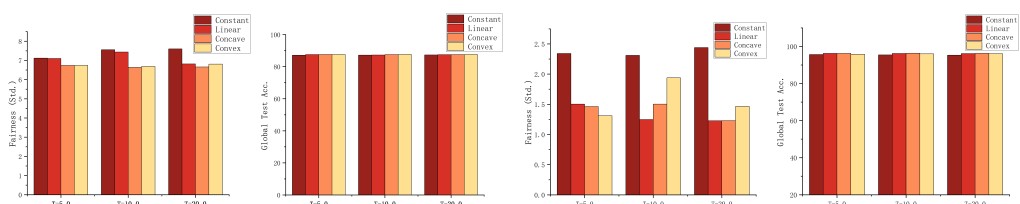

(a) **Fairness and Accuracy on Fashion-MNIST**  (b) **Fairness and Accuracy on MNIST**

Figure 9: Ablation study for Annealing schedule $\tau$

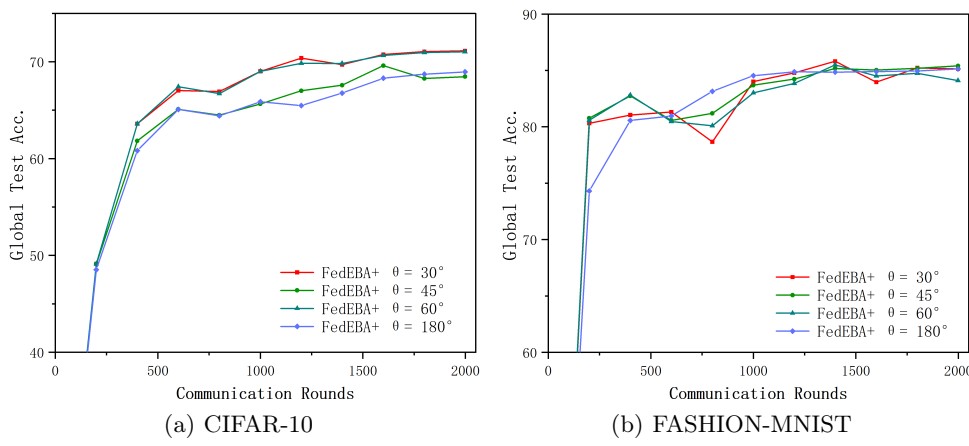

(a) CIFAR-10  (b) FASHION-MNIST

Figure 10: **Performance of $FedEBA+$ under different $\theta$ in terms of global accuracy.**

For the tolerable fair angle, we also provide the ablation studies of $\theta$. The results in Figure 10 11 12 show our algorithm is relatively robust to the tolerable fair angle $\theta$, though the choice of $\theta = 45$ may slow the performance slightly on global accuracy and min 5% accuracy over CIFAR-10.

For different fairness evaluation metrics, Table 17 demonstrates that in our setting, FedEBA+ exhibits competitive performance under FAA metrics. Instead, FOCUS exhibits a relatively

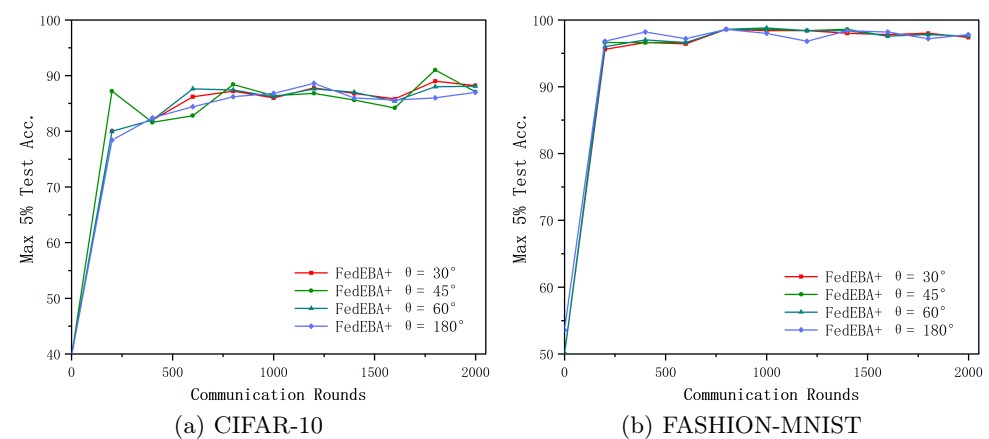

(a) CIFAR-10          (b) FASHION-MNIST

Figure 11: **Performance of $FedEBA+$ under different $\theta$ in terms of Max $5\%$ test accuracy.**

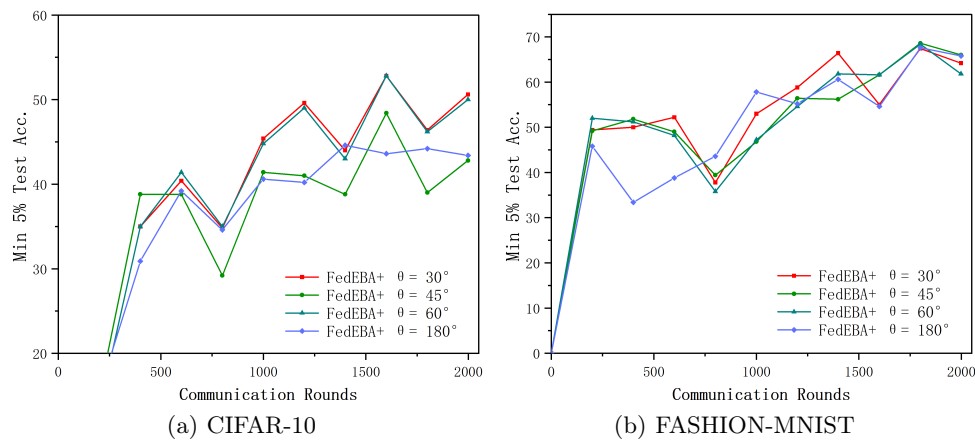

(a) CIFAR-10          (b) FASHION-MNIST

Figure 12: **Performance of $FedEBA+$ under different $\theta$ in terms of Min $5\%$ test accuracy.**

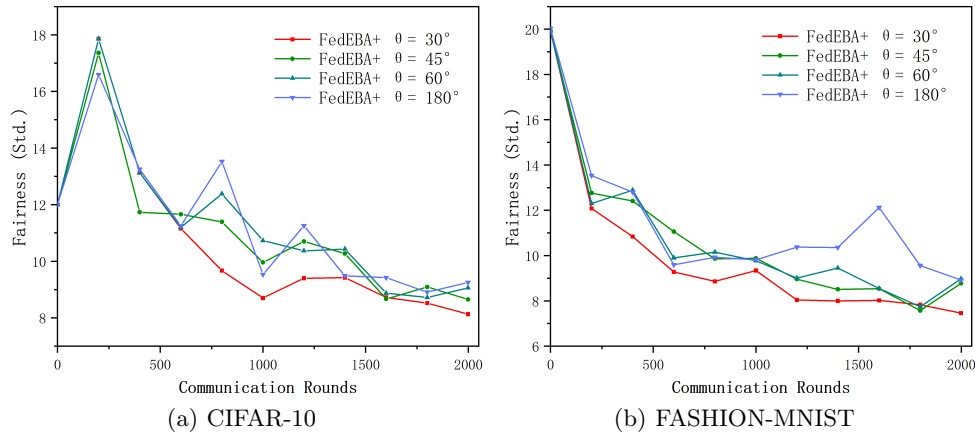

(a) CIFAR-10          (b) FASHION-MNIST

Figure 13: **Performance of $FedEBA+$ under different $\theta$ in terms of Fairness (Std).**

large FAA. This discrepancy arises from the differing settings between ours and FOCUS's. In our scenario, only a subset of clients undergoes training, contrasting with FOCUS's full client participation, consequently leading to subpar clustering performance.

Table 15: **Ablation study for FedEBA+ on four datasets.** We test the effectiveness of FedEBA+ when decomposing each proposed step, i.e., entropy-based aggregation and alignment update, on different datasets. FedEBA differs from FedAvg only in the aggregation method, and FedEBA+ incorporates the alignment into FedEBA. FedAvg serves as the backbone, FedAvg+① is employed to demonstrate the individual effectiveness of our proposed aggregation step, FedAvg+② is utilized to showcase the individual effectiveness of our proposed alignment step, and FedAvg + ① + ② is used to show the effectiveness of our proposed algorithm, FedEBA+.

| Algorithm | CIFAR-10 (CNN) | | | | FashionMNIST (MLP) | | | |
|---|---|---|---|---|---|---|---|---|
| | Global Acc. ↑ | Var. ↓ | Worst 5% ↑ | Best 5% ↑ | Global Acc.↑ | Var. ↓ | Worst 5% ↑ | Best 5% ↑ |
| FedAvg | 67.79±0.35 | 103.83±10.46 | 45.00±2.83 | 85.13±0.82 | 86.49±0.09 | 62.44±4.55 | 71.27±1.14 | 95.84±0.35 |
| FedAvg+① | 69.38±0.52 | 89.49±10.95 | 50.40±1.72 | 86.07±0.90 | 86.70±0.11 | 50.27±5.60 | 71.13±0.69 | 95.47±0.27 |
| FedAvg+② | 72.04±0.51 | 75.73±4.27 | 53.45±1.25 | 87.33±0.23 | 87.42± 0.09 | 60.08±7.30 | 69.12±1.23 | 97.8±0.19 |
| FedAvg+①+② | 72.75±0.25 | 68.71±4.39 | 55.80±1.28 | 86.93±0.52 | 87.50±0.19 | 43.41±4.34 | 72.07±1.47 | 95.91±0.19 |

| Algorithm | CIFAR-100 (Resnet-18) | | | | Tiny-ImageNet (MobileNet-2) | | | |
|---|---|---|---|---|---|---|---|---|
| | Global Acc. ↑ | Var. ↓ | Worst 5% ↑ | Best 5% ↑ | Global Acc.↑ | Var. ↓ | Worst 5% ↑ | Best 5% ↑ |
| FedAvg | 30.94±0.04 | 17.24±0.08 | 0.20±0.00 | 65.90±1.48 | 61.99±0.17 | 19.62±1.12 | 53.60±0.06 | 71.18±0.13 |
| FedAvg+① | 32.38±0.13 | 17.09±0.06 | 0.75±0.22 | 66.40±0.47 | 63.34±0.25 | 15.29±1.36 | 54.17±0.04 | 70.98±0.10 |
| FedAvg+② | 31.93±0.39 | 17.15±0.05 | 0.39±0.01 | 66.04±0.16 | 63.46±0.04 | 14.52±0.21 | 54.36±0.03 | 71.13±0.03 |
| FedAvg+①+② | 33.39±0.22 | 16.92±0.04 | 0.95±0.15 | 68.51±0.21 | 64.05±0.09 | 14.91±1.85 | 54.32±0.09 | 71.27±0.04 |

Table 16: **Performance of FedEBA+ with different $\tau$ and $\alpha$ choices.** The performance of different hyper-parameter choices of FedEBA+ shows better performance than baselines.

| Algorithm | FashionMNIST (MLP) | | CIFAR-10 (CNN) | |
|---|---|---|---|---|
| | Global Acc. | Var. | Global Acc. | Var. |
| FedAvg | 86.49 ± 0.09 | 62.44 ± 4.55 | 67.79 ± 0.35 | 103.83 ± 10.46 |
| q-FFL$|_{q=0.001}$ | 87.05 ± 0.25 | 66.67 ± 1.39 | 68.53 ± 0.18 | 97.42 ± 0.79 |
| q-FFL$|_{q=0.5}$ | 86.57 ± 0.19 | 54.91 ± 2.82 | 68.76 ± 0.22 | 97.81 ± 2.18 |
| q-FFL$|_{q=10.0}$ | 77.29 ± 0.20 | 47.20 ± 0.82 | 40.78 ± 0.06 | 85.93 ± 1.48 |
| PropFair$|_{M=0.2,thres=0.2}$ | 85.51 ± 0.28 | 75.27 ± 5.38 | 65.79 ± 0.53 | 79.67 ± 5.71 |
| PropFair$|_{M=5.0,thres=0.2}$ | 84.59 ± 1.01 | 85.31 ± 8.62 | 66.91 ± 1.43 | 78.90 ± 6.48 |
| FedFV$|_{\alpha=0.1,\tau fv=10}$ | 86.98 ± 0.45 | 56.63 ± 1.85 | 71.10 ± 0.44 | 86.50 ± 7.36 |
| FedFV$|_{\alpha=0.2,\tau fv=0}$ | 86.42 ± 0.38 | 52.41 ± 5.94 | 68.89 ± 0.15 | 82.99 ± 3.10 |
| FedEBA+$|_{\alpha=0.1,\tau=0.1}$ | 86.98±0.10 | 53.26±1.00 | 71.82±0.54 | 83.18±3.44 |
| FedEBA+$|_{\alpha=0.3,\tau=0.1}$ | 87.01±0.06 | 51.878±1.56 | 71.79±0.35 | 77.74±6.54 |
| FedEBA+$|_{\alpha=0.7,\tau=0.1}$ | 87.23±0.07 | 40.456±1.45 | 72.36±0.15 | 77.61±6.31 |
| FedEBA+$|_{\alpha=0.9,\tau=0.05}$ | 87.42±0.10 | 50.46±2.37 | 72.19±0.16 | 71.79±6.37 |
| FedEBA+$|_{\alpha=0.9,\tau=0.5}$ | 87.26±0.06 | 52.65±4.03 | 71.89±0.39 | 75.29±9.01 |
| FedEBA+$|_{\alpha=0.9,\tau=1.0}$ | 87.14±0.07 | 52.71±1.45 | 72.30±0.26 | 73.79±9.11 |
| FedEBA+$|_{\alpha=0.9,\tau=5.0}$ | 87.10±0.14 | 55.52±2.15 | 72.43±0.11 | 82.08±8.31 |

Table 17: **Performance of Fair FL Algorithms under FAA:** We present results under the FAA metric, as utilized in Chu et al. (2023), where FAA represents the discrepancy in excess loss across clients. The algorithms are tested on the FashionMNIST and CIFAR-10 datasets, with 10 out of 100 clients participating in each round. Specifically, for FOCUS, we adhere to the settings in Chu et al. (2023) and set the cluster number to 2. The smaller the FAA, the better.

| | FedAvg | AFL | q-FFL | FedFV | FOCUS | FedEBA+ |
|---|---|---|---|---|---|---|
| FashionMNIST | 0.7262±0.010 | 0.4500±0.006 | 0.4624±0.008 | 0.3749±0.017 | 1.16±0.161 | 0.4048±0.011 |
| CIFAR-10 | 2.296±0.031 | 0.8104±0.009 | 0.8465±0.013 | 0.7733±0.017 | 2.6448±0.061 | 0.6846±0.035 |

Table 18: Comparison of Algorithms with metric *coefficient of variation* ($C_V$) The $C_V$ improvement shows the improvement of algorithms over FedAvg. The result is calculated by global accuracy and variance of Table 1.

| Algorithm | FashionMNIST | | CIFAR-10 | |
|---|---|---|---|---|
| | $C_v = \frac{\text{std}}{\text{acc}}$ | $C_v$ improvement | $C_v = \frac{\text{std}}{\text{acc}}$ | $C_v$ improvement |
| FedAvg | 0.09136199 | 0% | 0.150312741 | 0% |
| q-FFL | 0.112432356 | -23% | 0.144026806 | 4.2% |
| FedMGDA+ | 0.089893051 | 1.3% | 0.146896915 | 2.4% |
| AFL | 0.088978374 | 2.6% | 0.134878199 | 10.1% |
| PropFair | 0.101459812 | -11.3% | 0.135671155 | 10.9% |
| TERM | 0.101659126 | -10.1% | 0.146631123 | 2.7% |
| FedFV | 0.086517483 | 4.8% | 0.130809249 | 13.3% |
| FedEBA+ | 0.072539115 | 21.8% | 0.1139402 | 27.8% |

Table 19: **Performance of Algorithms with Various Metrics.** We provide the results under cosine similarity and entropy metrics, as used in (Li et al., 2019a), the geometric angle corresponds to cosine similarity metric, and KL divergence between the normalized accuracy vector **a** and uniform distribution **u** that can be directly translated to the entropy of **a**. We test the algorithms on the FashionMNIST dataset, with fine-tuned hyperparameters.

| Algorithm | Global Acc. | Var. | Angle (∘) | KL $(a||u)$ |
|---|---|---|---|---|
| FedAvg | 86.49 ± 0.09 | 62.44±4.55 | 8.70±1.71 | 0.0145±0.002 |
| q-FFL | 87.05± 0.25 | 66.67± 1.39 | 7.97±0.06 | 0.0127±0.001 |
| FedMGDA+ | 84.64±0.25 | 57.89±6.21 | 8.21±1.71 | 0.0132±0.0004 |
| AFL | 85.14±0.18 | 57.39±6.13 | 7.28±0.45 | 0.0124±0.0002 |
| PropFair | 85.51±0.28 | 75.27±5.38 | 8.61±2.29 | 0.0139±0.002 |
| TERM | 84.31±0.38 | 73.46±2.06 | 9.04±0.45 | 0.0137±0.004 |
| FedFV | 86.98±0.45 | 56.63±1.85 | 8.01±1.14 | 0.0111±0.0002 |
| FedEBA+ | 87.50±0.19 | 43.41±4.34 | 6.46±0.65 | 0.0063±0.0009 |

Table 20: **Performance of algorithms on Fashion-MNIST and CIFAR-10.** Based on the same experimental setup as Table 1 in the main text, we introduce additional baselines that focus on designing aggregation algorithm suitable for the heterogeneous characteristics under the federated systems, namely, FedwAvg (Hong et al., 2022) and FedDISCO (Ye et al., 2023) to compare the performance. Specifically, FedwAvg assesses the number of forgettable samples of the global model on different clients' local data every $t$ global communication rounds and assigns higher aggregation weights to local update parameters with higher forgetting degrees; FedDISCO assigns different weights to the client update parameters based on the offset of the local data label distribution from the global data label distribution, with clients more aligned with the global data label distribution being assigned higher aggregation weights. Based on their original experimental section, we set appropriate hyper-parameters for the two added baselines, where $\alpha = 0.3$ for FedwAvg, $a = 0.1, b = 0.1$ for FedDISCO, and the distribution difference is calculated by L2 norm.

| Algorithm | Fashion-MNIST | | | | CIFAR-10 | | | |
|---|---|---|---|---|---|---|---|---|
| | Global Acc. ↑ | Var. ↓ | Worst 5% ↑ | Best 5% ↑ | Global Acc.↑ | Var. ↓ | Worst 5% ↑ | Best 5% ↑ |
| FedAvg | 86.49±0.09 | 62.44±4.55 | 71.27±1.14 | 95.84±0.35 | 67.79±0.35 | 103.83±10.46 | 45.00±2.83 | 85.13±0.82 |
| FedwAvg | 86.23±0.05 | 63.26±1.45 | 68.07±0.57 | 98.00±0.16 | 68.71±0.31 | 82.21±2.89 | 49.20±0.00 | 82.73±0.98 |
| FedDISCO | 85.74±0.34 | 57.61±5.17 | 68.00±3.00 | 98.07±0.09 | 69.27±0.45 | 86.39±6.35 | 48.43±1.50 | 83.67±0.82 |
| FedEBA | 86.70±0.11 | 50.27±5.60 | 71.13±0.69 | 95.47±0.27 | 69.38±0.52 | 89.49±10.95 | 50.40±1.72 | 86.07±0.90 |
| FedEBA+ | **87.50**±0.19 | **43.41**±4.34 | **72.07**±1.47 | 95.91±0.19 | **72.75**±0.25 | **68.71**±4.39 | **55.80**±1.28 | **86.93**±0.52 |

Table 21: **The impact of neural networks scalability of different widths on algorithms.** To test scalability, we set up experiments with CNNs that are narrower and wider than the main paper, and provided the running time required for each communication round. Specifically, the narrower CNN includes two convolutional layers (channel 3-32-32), and three linear layers (dimension 800-128-64-10). The wider CNN includes two convolutional layers (channel 3-128-128), and three linear layers (dimension 1600-384-192-10), with all other experimental settings being the same as the default.

| Algorithm | Narrower CNN | | | | Wider CNN | | | |
|---|---|---|---|---|---|---|---|---|
| | Global Acc. ↑ | Var. ↓ | Worst 5% ↑ | Best 5% ↑ | Global Acc.↑ | Var. ↓ | Worst 5% ↑ | Best 5% ↑ |
| FedAvg | 65.37±0.27 | 116.91±1.02 | 41.60±0.86 | 84.73±1.75 | 69.93±0.46 | 79.28±3.02 | 50.61±0.50 | 85.20±0.65 |
| q-FFL | 65.22±0.71 | 106.98±1.76 | 42.33±0.52 | 84.33±1.16 | 69.60±0.48 | 74.00±3.35 | 50.27±1.52 | 83.33±0.94 |
| FedEBA+ | **70.59**±0.61 | **58.95**±6.49 | **54.67**±2.65 | 84.13±0.52 | **74.14**±0.07 | **57.35**±5.74 | **56.47**±1.04 | **85.47**±0.25 |

Table 22: **The impact of neural networks scalability of different depths on algorithms.** To test scalability, we set up experiments with CNNs that are shallower and deeper than the main paper, and provided the running time required for each communication round. Specifically, the shallower CNN includes only one convolutional layer (channel 3-64), and three linear layers (dimension 64-384-192-10). The deeper CNN includes three convolutional layers (channel 3-64-128-128), and three linear layers (dimension 512-384-192-10), with all other experimental settings being the same as the default.

| Algorithm | Shallower CNN | | | | Deeper CNN | | | |
|---|---|---|---|---|---|---|---|---|
| | Global Acc. ↑ | Var. ↓ | Worst 5% ↑ | Best 5% ↑ | Global Acc.↑ | Var. ↓ | Worst 5% ↑ | Best 5% ↑ |
| FedAvg | 45.10±0.86 | 119.56±17.13 | 25.53±2.66 | 67.93±2.75 | 67.71±0.45 | 82.11±5.09 | 48.40±0.33 | 83.53±1.11 |
| q-FFL | 44.82±0.82 | 108.05±7.40 | 26.33±2.22 | 66.07±0.25 | 65.75±0.42 | 77.13±8.44 | 48.81±1.39 | 81.60±0.16 |
| FedEBA+ | **46.91**±1.28 | 113.30±20.19 | 25.80±2.90 | **68.60**±1.73 | **69.67**±0.42 | **69.95**±5.55 | **51.53**±1.62 | **83.80**±0.99 |

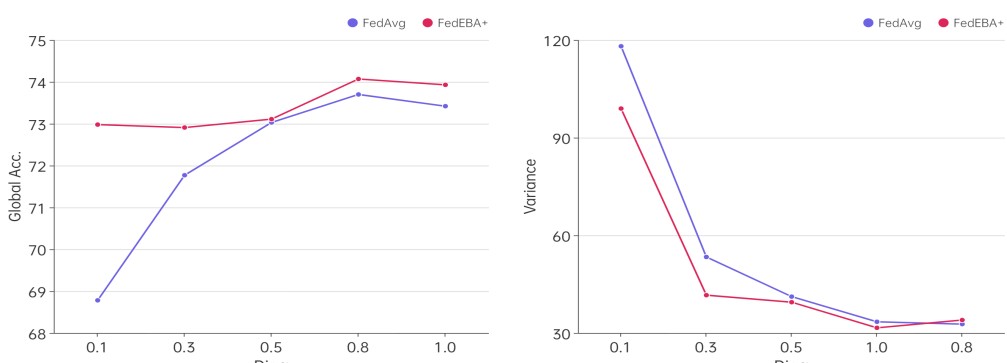

Figure 14: **Comparison of performance on CIFAR-10 under different degrees of Non-IID.** We performed different degrees of Non-IID partitioning on the CIFAR-10 dataset using Latent Dirichlet Allocation (LDA). Specifically, according to the degree of Non-IID from high to low, we set Dirichlet $\alpha \in \{0.1, 0.3, 0.5, 0.8, 1.0\}$. Combined with the different Non-IID partitions discussed in the main paper, this comprehensively demonstrates the performance of FedEBA+ under various scenarios.

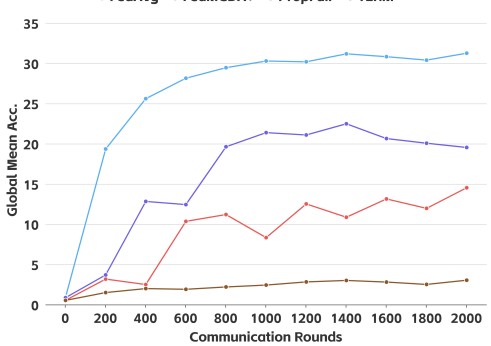

Figure 15: **Case of relatively low performance of FedMGDA+, PropFair, and TERM on the CIFAR-100 dataset with seed=1234.** In this setting, the accuracy of these algorithms is relatively poor, and the convergence is abnormal. However, with fine-tuned parameters and different seed setups, they can perform normally, and the relatively good performance of these algorithms is reported in Table 2.