# OpenReview forum: "Entropy-Based Aggregation for Fair and Effective Federated Learning"
_ICLR.cc/2025/Conference — Submitted to ICLR 2025_

### Official Review · Reviewer_8Wxv · 2024-10-27

**Soundness:** 4
**Presentation:** 4
**Contribution:** 4
**Rating:** 8
**Confidence:** 4

**Summary:**

Existing federated fairness algorithms strive to enhance fairness but often fall short in maintaining the overall performance of the global model, typically measured by the average accuracy of all clients. To address this issue, this paper proposes a novel algorithm that leverages entropy-based aggregation combined with model and gradient alignment to simultaneously optimize fairness and global model performance. The method presented in this paper employs a two-layer optimization framework and derives an analytical solution for the inner loop aggregation probabilities, making the optimization process highly computationally efficient. Furthermore, the paper introduces innovative alignment updates and adaptive strategies in the outer loop to further balance the performance of the global model and fairness. Additionally, the paper conducts a convergence analysis and numerous experiments to demonstrate the novelty and effectiveness of the proposed method. The content is very detailed, making it an excellent paper.

**Strengths:**

- Content is solid and comprehensive.
- The paper is well-structured and advances step by step.
- The theory is clear and solid.
- The experimental section is well-developed and extensive.

**Weaknesses:**

- Please explain how the convergence rate range mentioned in lines 396 to 397 of the main text is derived.

**Questions:**

see the weakness.

---

> ### Author Response · Authors · 2024-11-21
> **Response to Reviewer 8Wxv**
>
> Dear reviewer 8Wxv, Thank you very much for your thoughtful review and for recognizing the contributions of our paper. Below is our response to your question.
>
> > Q1. Please explain how the convergence rate range mentioned in lines 396 to 397 of the main text is derived.
> >
>
> Thank you for your constructive comment.
>
> From Theorem 5.1, we know that
>
> $\min _{t \in[T]}\mathbb{E}\left\|\nabla f\left(x_t\right)\right\|^2      \leq  \mathcal{O}\left(\frac{(f^0-f^*)}{\sqrt{mKT}} \right)+\mathcal{O}\left(\frac{\sqrt{m}\sum_iw_i^2\sigma_L^2}{2\sqrt{KT}} \right) + \mathcal{O}\left(\frac{5(\sigma_L^2+4K\sigma_G^2)}{2KT} \right)   \+ \mathcal{O}\left(\frac{20(A^2+1) \chi ^2\sigma_G^2 }{T} \right)  \, .$
>
> According to the properties of unified probability, we have $\frac{1}{m} \leq \sum_{i=1}^m w_i^2 \leq 1$. The upper bound arises from $\sum_i w_i^2 \leq \sum_i w_i$, and the lower bound follows from the Cauchy-Schwarz inequality.
>
> When $w_i = \frac{1}{m}$, $\sum_{i=1}^m w_i^2 = \frac{1}{m}$, and substituting this into the convergence rate gives an upper bound of $\mathcal{O}\left(\frac{1}{\sqrt{mKT}} + \frac{1}{T}\right)$.
>
> On the other hand, when $\sum_{i=1}^m w_i^2 = 1$, the convergence rate becomes $\mathcal{O}\left(\frac{\sqrt{m}}{\sqrt{KT}} + \frac{1}{T}\right)$. Therefore, we conclude that the upper bound of the convergence rate lies between $\mathcal{O}\left(\frac{1}{\sqrt{mKT}} + \frac{1}{T}\right)$ and $\mathcal{O}\left(\frac{\sqrt{m}}{\sqrt{KT}} + \frac{1}{T}\right)$.

---

### Official Review · Reviewer_Tsda · 2024-11-04

**Soundness:** 4
**Presentation:** 3
**Contribution:** 3
**Rating:** 6
**Confidence:** 4

**Summary:**

This paper introduces FedEBA+, a novel federated learning algorithm designed to enhance both fairness and global model performance through a computationally efficient bi-level optimization framework. The authors propose an innovative entropy-based fair aggregation method for the inner loop and develop adaptive alignment strategies to optimize global performance and fairness in the outer loop. The paper provides theoretical analysis confirming the convergence of FedEBA+ under non-convex settings and demonstrates its superiority over state-of-the-art fairness algorithms through empirical results on various datasets.

**Strengths:**

- The work addresses a critical issue in federated learning and provides a practical solution that could have some impact on the field.
- The paper is well-researched, with a robust theoretical foundation and comprehensive empirical validation.
- The authors provide insightful convergence analysis and fairness guarantees.
- The objective function and the proposed algorithm are novel contributions.

**Weaknesses:**

- The paper would benefit from a more intuitive explanation of the connection between entropy and fairness.
- The explanation of the entropy-based aggregation method could be more detailed, particularly in terms of how it differs from and improves upon existing methods.
- The concept of "ideal loss" appears somewhat confusing. Since $\tilde{f}(x)$ represents the ideal loss, which signifies the global model’s performance under an ideal training setting, it is unclear why the gradient of the ideal loss can be estimated by averaging local one-step gradients during the model alignment procedure, yet it should be estimated by Equation.10 during the gradient alignment procedure. Could the authors give more intuition to clarify this point?

**Questions:**

See Weaknesses.

---

> ### Author Response · Authors · 2024-11-21
> **Response to Reviewer Tsda (1/2)**
>
> Dear reviewer Tsda, we would like to thank you for your time spent reviewing our paper and for providing constructive comments. Please kindly find our responses to your raised questions below:
>
> > Q1. The paper would benefit from a more ****intuitive explanation of the connection between entropy and fairness.
> >
>
> Thank you for your suggestion.
>
> Entropy is widely used to promote fairness in resource allocation [1] and data processing [2].
>
> **A motivation for its use in these scenarios is contained in the entropy concentration theorem** [3], which posits that maximum entropy provides the least biased solution for determining prior probabilities.
>
> However, the **existing** **entropy concentration theorem cannot be directly applied to federated learning settings.** In existing fairness applications, maximizing entropy without considering constraints will lead to a uniform distribution, where all users share equal resources. This conflicts with the natural heterogeneity in federated learning; aggregation with equal weights of all users cannot achieve the goal of fairness, i.e., equal performance across users. We address this challenge by introducing constraints to adjust weights, aligning aggregated performance with that of the ideal fair model. This constrained entropy approach focuses more on underperforming clients, assigning them higher weights and reducing performance unfairness.
>
> [Toy example]
>
> We present a toy example to demonstrate how entropy and fairness work. FedAvg uses uniform aggregation, which is equivalent to entropy without constraints.
>
> In particular, consider two clients with loss function $f_1\left(x_t\right)=2(x-2)^2, f_2\left(x_t\right)=\frac{1}{2}(x+4)^2$ respectively.
>
> Assuming $x_t=0$, w.l.o.g, we consider one single-step gradient decent with stepsize=1/4. Then $x_1^{t+1}=x_t-\lambda \nabla f_1\left(x_t\right)=2, x_2^{t+1}=x_t-\lambda \nabla f_2\left(x_t\right)=-1$.
>
> Thus, the aggregation probabilities for FedAvg, q-FFL, and FedEBA will be:
>
> ($\frac{1}{2}, \frac{1}{2}$), ($\frac{4}{13},\frac{4}{13}$), ($\frac{1}{1+e^{9/2}}, \frac{e^{9/2}}{1+e^{9/2}}$).
>
> Then the variance of $x_{t+1}$ will be:
>
> $Var_{AVG} = 2*(2.81)^2, Var_{q-FFL}=2*(2.52)^2, Var_{EBA+}=2*(0.6)^2$
>
> The corresponding variance rank is $Var_{EBA+}<Var_{q-FFL}<Var_{AVG}$.
>
> In this case, the normalized performance's entropy, after maxing the constrained entropy of aggregation probability, exhibits a relationship akin to variance (greater entropy corresponds to improved fairness).
> $\operatorname{Entropy}\left(f\left(x_{E B A+}^{t+1}\right)\right)=-\sum_{i=1}^2 \frac{f_i\left(x_{E B A+}^{t+1}\right)}{\sum_{j=1}^2 f_j\left(x_{E B A+}^{t+1}\right)} \log \left(\frac{f_j\left(x_{E B A+}^{t+1}\right)}{\sum_{i=j}^2 f_i\left(x_{E B A+}^{t+1}\right)}\right) \approx 0.996$, where $f_1\left(x_{E B A+}^{t+1}\right)=2 *(2.1)^2, f_2\left(x_{E B A+}^{t+1}\right)=\frac{1}{2} *(3.9)^2$.
> $\operatorname{Entropy}\left(f\left(x_{q-FFL}^{t+1}\right)\right)=-\sum_{i=1}^2 \frac{f_i\left(x_{q-FFL}^{t+1}\right)}{\sum_{j=1}^2 f_j\left(x_{q-FFL}^{t+1}\right)} \log \left(\frac{f_j\left(x_{q-FFL}^{t+1}\right)}{\sum_{i=j}^2 f_i\left(x_{q-FFL}^{t+1}\right)}\right) \approx 0.942$, where $f_1\left(x_{q-FFL}^{t+1}\right)=2 *(2.4)^2, f_2\left(x_{q-FFL}^{t+1}\right)=\frac{1}{2} *(3.6)^2$.
> $\operatorname{Entropy}\left(f\left(x_{AVG}^{t+1}\right)\right)=-\sum_{i=1}^2 \frac{f_i\left(x_{\text {avg}}^{t+1}\right)}{\sum_{j=1}^2 f_j\left(x_{\text {AVG}}^{t+1}\right)} \log \left(\frac{f_j\left(x_{\text {AVG}}^{t+1}\right)}{\sum_{i=j}^2 f_i\left(x_{\text {AVG}}^{t+1}\right)}\right) \approx 0.890$, where $f_1\left(x_{AVG}^{t+1}\right)=2 *(1.5)^2, f_2\left(x_{AVG}^{t+1}\right)=\frac{1}{2} *(4.5)^2$.
>
> Therefore, $Entropy(f(x_{EBA+}^{t+1}))>Entropy(f(x_{q-FFL}^{t+1}))>Entropy(f(x_{AVG}^{t+1}))$ and $Var_{EBA+}<Var_{q-FFL}<Var_{AVG}$.
>
> We have mentioned this in Appendix I.1 of the revised manuscript.
>
> [1]Johansson M, Sternad M. Resource allocation under uncertainty using the maximum entropy principle[J]. IEEE Transactions on Information Theory, 2005, 51(12): 4103-4117.
>
> [2]Celis L E, Keswani V, Vishnoi N. Data preprocessing to mitigate bias: A maximum entropy based approach[C]//International conference on machine learning. PMLR, 2020: 1349-1359.
>
> [3]Jaynes E T. On the rationale of maximum-entropy methods[J]. Proceedings of the IEEE, 1982, 70(9): 939-952.

---

> ### Author Response · Authors · 2024-11-21
> **Response to Reviewer Tsda (2/2)**
>
> > Q2. The explanation of the entropy-based aggregation method could be more detailed, particularly in terms of how it differs from and improves upon existing methods.
> >
>
> Thank you for your suggestion.
>
> We clarify the differences compared to existing aggregation methods by:
>
> - **The aggregation formulation is different**. Our aggregation probability $p_i = e^{\frac{F_i(x)/\tau}{Z}}$ is proportional to the exponential term of loss with a controllable parameter $\tau$. Unlike heuristic fair approaches that assign weights proportional to client loss $p_i \propto F_i(x)$[1, 5,6], our method is derived from a well-defined constrained optimization problem.
> - **The goal is different.** Existing entropy-based aggregation methods [3,4] and softmax-based reweighting approaches [5,6] have entirely different objectives from fair FL. These methods focus on improving model accuracy but lack the ability to enhance fairness.
> - **Novelty in the Entropy Model for FL:** To extend entropy to FL fairness, we introduce a novel constrained entropy model, the first of its kind in the FL fairness community. This model incorporates a constraint that adjusts the distribution to prioritize underperforming clients, resulting in a weighted fair aggregation.
>
> The advantages of our approach over existing methods lie in:
>
> - **The exponent form and control term $\tau$ benefit the fairness level and enhance existing fair aggregation methods.** For our model, $p_i = e^{\frac{F_i(x)/\tau}{Z}}$, the exponent effectively addresses extreme loss (extreme unfairness), since the exponent grows faster with increasing loss. Additionally, the controllable term $\tau$ regulates the fairness level by determining the spread of weights assigned to each client. As Remark 4.2 shows, by choosing an appropriate expression for $\tau$, we can recover existing fair aggregation algorithms like FedAvg, AFL, and q-FFL [2].
> - **Empirically more effective in fairness than others.** Compared with existing fair aggregation methods like q-FFL and TERM, our entropy aggregation (FedEBA+ with $\alpha=0$) shows better performance in both accuracy and fairness, as shown in Tables 1 and 2 of the submission.
>
> [1]Fairness-aware loss history based federated learning heuristic algorithm[J]. Knowledge-Based Systems, 2024, 288: 111467.
>
> [2]Fair Resource Allocation in Federated Learning[C]//International Conference on Learning Representations.
>
> [3]Self-Driven Entropy Aggregation for Byzantine-Robust Heterogeneous Federated Learning. ICML 2024.
>
> [4]Enhancing Federated Learning Convergence with Dynamic Data Queue and Data Entropy-driven Participant Selection[J]. IEEE Internet of Things Journal, 2024.
>
> [5]A dynamic reweighting strategy for fair federated learning[C]//ICASSP 2022.
>
> [6]Fair federated learning for heterogeneous data[C]//9th ACM IKDD CODS and 27th COMAD 2022.
>
> > Q3. The concept of "ideal loss" appears somewhat confusing. Since f(x) represents the ideal loss, which signifies the global model’s performance under an ideal training setting, it is unclear why the gradient of the ideal loss can be estimated by averaging local one-step gradients during the model alignment procedure, yet it should be estimated by Equation.10 during the gradient alignment procedure. Could the authors give more intuition to clarify this point?
> >
>
> Thank you for your comment. We would like to clarify that the ideal loss$\tilde{f}(x$) has two different estimations depending on the improvement goals:
>
> - **For global accuracy improvement**, it is estimated by Eq. (9) instead of Eq. (10). Eq. (9) represents the averaging of local one-step gradients. The rationale for using Eq. (9) as a good estimation for global updates is as follows:
>     1. A single local update corresponds to an unshifted update on local data, whereas multiple local updates introduce model bias in heterogeneous FL~[1,2].
>     2. The expectation of sampled clients’ data over rounds represents the global data due to unbiased random sampling, as stated in submission L340–343.
> - **For fairness improvement**, it is estimated by Eq. (10), which reweights the current model’s gradient. First, it uses the global model’s loss to calculate the fair weights for each participating client, based on the proposed entropy-based aggregation strategy. Then, using the global model’s gradient ensures an unshifted update on local data, capturing the ideal fair update.
>
> For example, ideally, if each client performs full gradient descent for one local epoch:
>
> - Eq. (9) will represent the true update averaged over all data, similar to centralized learning.
> - Eq. (10) will represent the true fair update of the current model $x_t$ across these clients, using the entropy-based aggregation probabilities derived in our paper.
>
> [1]Scaffold: Stochastic controlled averaging for federated learning[C]//ICML 2020.
>
> [2]Local learning matters: Rethinking data heterogeneity in federated learning[C]//CVPR 2022.

---

> > ### Comment · Reviewer_Tsda · 2024-11-26
> >
> > Thank you for the response. You have addressed my concerns. Given other reviews, I will keep my score as is.

---

> > > ### Author Response · Authors · 2024-11-28
> > > **Thank you for maintaining the positive score.**
> > >
> > > Dear Reviewer Tsda,
> > >
> > > We appreciate your continued support and the positive score you’ve given our paper. In light of your valuable feedback, we have revised the manuscript accordingly.

---

### Official Review · Reviewer_591F · 2024-11-04

**Soundness:** 2
**Presentation:** 2
**Contribution:** 2
**Rating:** 5
**Confidence:** 4

**Summary:**

The paper titled "Entropy-Based Aggregation for Fair and Effective Federated Learning" proposes a novel algorithm called FedEBA+ to address the fairness issues in Federated Learning (FL) while maintaining the global model's performance. The authors leverage an entropy-based aggregation mechanism, combined with model and gradient alignment, to optimize both fairness and global performance. A bi-level optimization framework is introduced to derive an efficient closed-form solution for aggregation probabilities, which is claimed to enhance fairness by adjusting weights for underperforming clients. The paper provides theoretical guarantees for the algorithm's convergence and fairness, and empirical results on several datasets demonstrate that the proposed approach outperforms existing fairness-oriented FL algorithms.

**Strengths:**

- **Clear Problem Motivation**: The paper addresses an important problem in federated learning—how to achieve fairness across clients while ensuring that the global model's performance does not degrade. The motivation behind balancing fairness and performance in a heterogeneous FL environment is well articulated.

- **Theoretical Guarantees**: The paper provides theoretical analysis for the convergence of FedEBA+ in non-convex settings, which adds credibility to the proposed approach. The fairness improvements are also supported by performance variance analysis.

- **Bi-level Optimization Framework**: The introduction of a bi-level optimization framework is interesting and provides a structured way to balance fairness and performance. The authors also derive a closed-form solution for the aggregation probability, which improves computational efficiency.

**Weaknesses:**

- **Lack of Novelty in Aggregation Strategy**: The core novelty of the paper—using an entropy-based aggregation mechanism—essentially boils down to applying an exponential normalization based on client losses to determine aggregation weights. This approach is not particularly novel, as similar strategies have been used in various other contexts (e.g., softmax-based weighting).

- **Overemphasis on Entropy**: The paper heavily emphasizes entropy, but in practice, the core idea is simply a re-weighting based on client loss. The connection between entropy and fairness, while valid, feels somewhat forced in its application here. The novelty of applying maximum entropy principles in such a straightforward manner might be overstated.

- **Limited Innovation in Fairness-Performance Trade-off**: Although the paper claims to balance fairness and performance, the approach primarily adjusts aggregation weights based on client loss. Many existing algorithms adjust client weights in some form, and the use of entropy as a fairness mechanism does not seem to introduce substantial improvements beyond what is already available in the literature.

**Questions:**

1. **Novelty of Exponential Normalization**: The core aggregation strategy is based on an exponential normalization of client losses. Could the authors clarify how this approach fundamentally differs from existing techniques that also adjust aggregation weights based on client performance? How does the use of entropy provide a significant advantage over these existing methods?

---

> ### Author Response · Authors · 2024-11-21
> **Response to Reviewer 591F**
>
> Dear reviewer 591F, we would like to thank you for your time spent reviewing our paper and for providing constructive comments. Please kindly find our responses to your raised questions below:
>
> **Weaknesses:**
>
> > W1. **Lack of Novelty in Aggregation Strategy**
> >
>
> See reply to Questions.
>
> > W2. **Overemphasis on Entropy**
> >
>
> See reply to Questions.
>
> > W3. **Limited Innovation in Fairness-Performance Trade-off**: Although the paper claims to balance fairness and performance, the approach primarily adjusts aggregation weights based on client loss. Many existing algorithms adjust client weights in some form, and the use of entropy as a fairness mechanism does not seem to introduce substantial improvements beyond what is already available in the literature.
> >
> - The approach consists of two main parts: aggregation and alignment updates.
> - For aggregation alone, the benefits of using the proposed aggregation probability are twofold:
>     - It generalizes many existing aggregation methods and has a clear theoretical foundation, as opposed to heuristic approaches that are simply proportional to loss, as seen in many works [1,5,6].
>     - The improvement over existing aggregation-only methods is significant. Compared to methods like q-FFL and TERM, our entropy-based aggregation (FedEBA+ with α=0) demonstrates better performance in terms of both accuracy and fairness, as shown in Table 1 and Table 2 of the submission.
>
>
>
> **Questions:**
>
> > Q1. **Novelty of Exponential Normalization**: The core aggregation strategy is based on an exponential normalization of client losses. Could the authors clarify how this approach fundamentally differs from existing techniques that also adjust aggregation weights based on client performance? How does the use of entropy provide a significant advantage over these existing methods?
> >
>
> Thank you for your suggestion.
>
> We clarify the differences compared to existing aggregation methods by:
>
> - **The aggregation formulation is different**. Our aggregation probability $p_i = e^{\frac{F_i(x)/\tau}{Z}}$ is proportional to the exponential term of loss with a controllable parameter $\tau$. Unlike heuristic fair approaches that assign weights proportional to client loss $p_i \propto F_i(x)$[1, 5,6], our method is derived from a well-defined constrained optimization problem.
> - **The goal is different.** Existing entropy-based aggregation methods [3,4] and softmax-based reweighting approaches [5,6] have entirely different objectives from fair FL. These methods focus on improving model accuracy but lack the ability to enhance fairness.
> - **Novelty in the Entropy Model for FL:** To extend entropy to FL fairness, we introduce a novel constrained entropy model, the first of its kind in the FL fairness community. This model incorporates a constraint that adjusts the distribution to prioritize underperforming clients, resulting in a weighted fair aggregation.
>
> The advantages of our approach over existing methods lie in:
>
> - **The exponent form and control term $\tau$ benefit the fairness level and enhance existing fair aggregation methods.** For our model, $p_i = e^{\frac{F_i(x)/\tau}{Z}}$, the exponent effectively addresses extreme loss (extreme unfairness), since the exponent grows faster with increasing loss. Additionally, the controllable term $\tau$ regulates the fairness level by determining the spread of weights assigned to each client. As Remark 4.2 shows, by choosing an appropriate expression for $\tau$, we can recover existing fair aggregation algorithms like FedAvg, AFL, and q-FFL [2].
> - **Empirically more effective in fairness than others.** Compared with existing fair aggregation methods like q-FFL and TERM, our entropy aggregation (FedEBA+ with $\alpha=0$) shows better performance in both accuracy and fairness, as shown in Tables 1 and 2 of the submission.
>
> [1]Mollanejad A, Navin A H, Ghanbari S. Fairness-aware loss history based federated learning heuristic algorithm[J]. Knowledge-Based Systems, 2024, 288: 111467.
>
> [2]Li T, Sanjabi M, Beirami A, et al. Fair Resource Allocation in Federated Learning[C]//International Conference on Learning Representations.
>
> [3] Huang, W., Shi, Z., Ye, M., Li, H. & Du, B. Self-Driven Entropy Aggregation for Byzantine-Robust Heterogeneous Federated Learning. ICML 2024
>
> [4]Herath C, Liu X, Lambotharan S, et al. Enhancing Federated Learning Convergence with Dynamic Data Queue and Data Entropy-driven Participant Selection[J]. IEEE Internet of Things Journal, 2024.
>
> [5]Zhao Z, Joshi G. A dynamic reweighting strategy for fair federated learning[C]//ICASSP 2022-2022 IEEE International Conference on Acoustics, Speech and Signal Processing (ICASSP). IEEE, 2022: 8772-8776.
>
> [6]Kanaparthy S, Padala M, Damle S, et al. Fair federated learning for heterogeneous data[C]//Proceedings of the 5th Joint International Conference on Data Science & Management of Data (9th ACM IKDD CODS and 27th COMAD). 2022: 298-299.

---

> > ### Comment · Reviewer_591F · 2024-12-02
> >
> > Thank you for your detailed response to my comments and concerns. While I appreciate the effort you have made to clarify the novelty and advantages of your approach, I will maintain my original score. Specifically, while you argue that your aggregation probability is derived from a constrained optimization problem rather than heuristic approaches, the core idea of using exponential normalization of client losses appears conceptually similar to existing methods that adjust aggregation weights. The addition of a controllable parameter does not sufficiently distinguish your method from prior work.

---

> > > ### Author Response · Authors · 2024-12-02
> > > **To Reviewer 591F**
> > >
> > > Dear Reviewer 591F,
> > > thank you very much for your response.
> > >
> > > We would like to emphasize that our contributions involve both novel methods and solid theoretical analyses. As for **methodological contribution**, the aggregation strategy is only part of our method; another significant component is the **alignment update strategy**. As for **theoretical contribution**, we are **the first** to provide comprehensive fairness analysis for general convex models and convergence analysis for nonconvex models.
> > >
> > > 1. **[Methodological Contribution]** **The aggregation strategy is only part of our paper's methodological contribution; another significant component is the alignment update strategy.** We believe it is unreasonable to dismiss the paper's contributions solely due to similarities in the form of the aggregation probability formulas.
> > >     - Notably, we are the first to use the exponential loss-based aggregation method to achieve fairness in FL. Simply using the softmax-based method alone leads to poor improvement in fairness, and the controllable parameter is helpful in enhancing fairness. Additionally, the controllable softmax-based aggregation, combined with the alignment update, completes our algorithm and demonstrates significant performance. The detailed results are provided in the Ablation Study of FedEBA+ in **Table 15** of our original submission; we also briefly summarize them here for visualization.
> > >
> > > |  | Acc. on CIFAR-10 | Var. on CIFAR-10 | Acc. on FashionMNIST | Var. on FashionMNIST |
> > > | --- | --- | --- | --- | --- |
> > > | Softmax-based aggregation | 67.78±0.62 | 91.24±7.55 | 86.09±0.11 | 55.64±1.79 |
> > > | Softmax + **controllable parameter**  | 69.38±0.52 |  89.49±10.95 | 86.70±0.11 | 50.27±5.60 |
> > > | Softmax+**controllable parameter**+**alignment update (ours)** | 72.75±0.25 | 68.71±4.39 | 87.50±0.19 | 43.41±4.34 |
> > > |  |   |  | |  |
> > >
> > > 2. **[Theoretical Contribution]**
> > >
> > >     This paper is the first aggregation-based method that provides fairness analysis in both linear and convex models, as well as nonconvex convergence, which includes both the aggregation-only convergence rate and the aggregation-and-alignment update (complete FedEBA+) convergence rate.
> > >
> > >     - FedEBA+ offers both linear and convex model analyses for fairness, which are not provided in prior works (e.g., from q-FFL to lp-proj in the table below), making our fairness analysis a significant contribution to the field.
> > >     - Compared with existing aggregation methods (e.g., from LHFed to FedwAVG in the table below), our FedEBA+ provides not only complete nonconvex convergence but also a fairness analysis guarantee.
> > >
> > >     | Algorithm | Fairness analysis - linear model | Fairness analysis - convex model | Convergence analysis  |
> > >     | --- | --- | --- | --- |
> > >     | **FedEBA**+ | ✔ | ✔ | Nonconvex |
> > >     | q-FFL | ✔ | ✔ | None |
> > >     | FedMGDA+ | ✖ | ✖ | Stongly Convex |
> > >     | TERM | ✔ | ✖ | Stongly Convex |
> > >     | AFL | ✖ | ✖ | Convex |
> > >     | PropFair | ✖ | ✖ | Nonconvex |
> > >     | lp-proj | ✔ | ✖ | Nonconvex |
> > >     | LHFed [1] | ✖ | ✖ | None |
> > >     | SDEA [2] | ✖ | ✖ | None |
> > >     | DDFL [3] | ✖ | ✖ | Stongly Convex |
> > >     | DR-FedAvg [4] | ✖ | ✖ | None |
> > >     | FairAvg [5] | ✖ | ✖ | None |
> > >     | FAIR [6] | ✖ | ✖ | None |
> > >     | FedDisco [7] | ✖ | ✖ | Nonconvex |
> > >     | FedFa [8] | ✖ | ✖ | None |
> > >     | FedwAvg [9] | ✖ | ✖ | None |
> > >
> > >
> > > [1]Mollanejad A, Navin A H, Ghanbari S. Fairness-aware loss history based federated learning heuristic algorithm[J]. Knowledge-Based Systems, 2024.
> > >
> > > [2]Huang, W., Shi, Z., Ye, M., Li, H. & Du, B. Self-Driven Entropy Aggregation for Byzantine-Robust Heterogeneous Federated Learning. ICML 2024
> > >
> > > [3]Herath C, Liu X, Lambotharan S, et al. Enhancing Federated Learning Convergence with Dynamic Data Queue and Data Entropy-driven Participant Selection[J]. IEEE Internet of Things Journal, 2024.
> > >
> > > [4]Zhao Z, Joshi G. A dynamic reweighting strategy for fair federated learning[C]//ICASSP 2022-2022 IEEE International Conference on Acoustics, Speech and Signal Processing (ICASSP). IEEE, 2022: 8772-8776.
> > >
> > > [5]Kanaparthy S, Padala M, Damle S, et al. Fair federated learning for heterogeneous data[C]//Proceedings of the 5th Joint International Conference on Data Science & Management of Data (9th ACM IKDD CODS and 27th COMAD). 2022: 298-299.
> > >
> > > [6]Deng Y, Lyu F, Ren J, et al. Fair: Quality-aware federated learning with precise user incentive and model aggregation[C]//IEEE INFOCOM 2021-IEEE Conference on Computer Communications. IEEE, 2021: 1-10.
> > >
> > > [7]Ye R, Xu M, Wang J, et al. Feddisco: Federated learning with discrepancy-aware collaboration[C]//International Conference on Machine Learning. PMLR, 2023: 39879-39902.
> > >
> > > [8]Huang W, Li T, Wang D, et al. Fairness and accuracy in federated learning[J]. arXiv preprint arXiv:2012.10069, 2020.
> > >
> > > [9]Hong M, Kang S K, Lee J H. Weighted averaging federated learning based on example forgetting events in label imbalanced non-iid[J]. Applied Sciences, 2022, 12(12): 5806.

---

### Official Review · Reviewer_acXk · 2024-11-05

**Soundness:** 2
**Presentation:** 2
**Contribution:** 3
**Rating:** 6
**Confidence:** 3

**Summary:**

Existing algorithm to achieve fairness in FL often do so at the expense of global model accuracy. The paper proposes FedEPA+, an algorithm to improve fairness in FL while maintaining global model performance. The key idea here is to use entropy-based aggregation followed by a novel model alignment technique which adaptively optimizes for either global accuracy or fairness depending on the current performance of the model. Theoretical results are provided showing the convergence of the proposed algorithm in general non-convex FL setups along with guaranteed improvement in fairness in the strongly convex scenario. Empirical results show that the proposed algorithm outperforms existing fair FL baselines in both global model accuracy as well as fairness.

**Strengths:**

1. To the best of my knowledge, the idea of using a constrained entropy model for aggregation in FL to improve fairness is novel and interesting.
2. The idea of adaptively changing the global aggregation to either prioritize global model accuracy or fairness, while heuristic, is again interesting.
3. Theoretical results proving that the proposed FedEBA+ can reduce variance compared to FedAvg for generalized regression and strongly convex models.
4. Experimental results look promising with the proposed FedEBA+ outperforming FedAvg and other baselines both in terms of global model performance and fairness.

**Weaknesses:**

1. The clarity of writing and motivation for doing FedEBA+ can be significantly improved. There were several parts which I found confusing or could not fully comprehend as I discuss below.

* It is a bit surprising to me that the expression for the aggregation probabilities in Eq. (4) does not depend on the desired $\tilde{f}(x)$ in Eq. (3). Is there an intuitive explanation for why this is the case?

* Authors are encouraged to provide an intuitive explanation for why maximizing constrained entropy can improve fairness. Right now, entropy-based aggregation just appears to be a black box to improve fairness

* While I could follow the construction of the proposed bi-level objective in Eq. (6) and the idea to adaptively change the ideal gradient, I was unable to follow the changes made in the local and global aggregation to account for this. In my understanding, we first solve the inner problem. i.e, find $p$ given the current global model. Then given $p$, we update the global model with either the ideal global gradient or ideal fair gradient depending on the requirement. Given this understanding, my questions are as follows:

    * In the alignment to improve global accuracy why are the $p_i$'s computed using the loss of the local models? (Line 312). In my understanding, since we are fixing $x$, when solving the inner problem, the $p$ should be computed using the loss of the global model, i.e, the expression in Line 351?
    * In the alignment to improve fairness why is the local optimization at clients changed  (Eq. 11 and Line 11 in Algorithm 1)? In my understanding only global aggregation should change?

* I don't follow how in Prac-FedEPA+ clients need to only communicate once. If we follow Algorithm 3 then it appears that communication can happen twice within a round: first in Line 6-8 and then again in Lines  18-22. I found this to be very confusing and authors should clarify what is the communication cost in their experiments.

* $A$ is not defined in Theorem 5.1. Similarly $w$ is not defined, although from context it appears to be the prior aggregation weights of client objectives.

* It seems a little strange to see discussion on the lower bound of an upper bound in Remark 5.2. Authors should just state the worst possible convergence rate of FedEPA+ by considering that $\sum_{i=1}^M w_i^2 \leq 1$.

* In Remark 5.3. it appears that to show convergence of $\alpha \neq 0$ we need to assume $k << m$. Is this correct? If so, authors should clearly mention this and explain why this is the case. Also I don't follow the argument that the proposed alignment results in a faster convergence rate than FedAvg since both seem to be achieving $O(1/\sqrt{mKT})$ rate.

* Why is FedMGDA+, PropFair and TERM incompatible in Table 2?

2.  The parameter $\theta$ appears to be crucial to the performance of the algorithm but currently there is very little discussion on how to set this parameter and its impact on convergence. Theoretically, we don't see any effect of $\theta$ in Theorem 1, which is a bit surprising to me. In practice, I would also suggest moving Table 7 to the main text and discuss the trade-off in setting $\theta$ in more detail.

**Questions:**

Please see the questions and comments listed in the Weakness section

---

> ### Author Response · Authors · 2024-11-21
> **Response to Reviewer acXk (1/4)**
>
> Dear reviewer acXK, thank you very much for taking the time to review our paper. We have carefully revised the writing and clarified the motivation in the paper based on your suggestions. Below, you can find our detailed responses to your questions:
>
> > Q1. It is a bit surprising to me that the expression for the aggregation probabilities in Eq. (4) does not depend on the desired $\tilde{f}(x)$ in Eq. (3). Is there an intuitive explanation for why this is the case?
> >
>
> Thank you for your comment.
>
> In derivation of Eq. (3) $p_i=\frac{\left.\exp \left[F_i\left(x_i\right) / \tau\right)\right]}{\sum_{i=1}^N \exp \left[F_i\left(x_i\right) / \tau\right]}$, $\tau$ is related to $\tilde{f}(x)$ according to Eq. (24).
>
> While$\tilde{f}(x)$ influences $\tau$, it is treated as a constant during the derivation and approximated later, making $p_i$ independent of $\tilde{f}(x)$ and the estimation of $\tilde{f}(x)$ is explicitly inflected in $\tau$.
>
> In particular,  by optimizing $L\left(p, \lambda_0, \lambda_1\right):=-\left[\sum_{i=1}^N p_i \log p_i+\lambda_0\left(\sum_{i=1}^N p_i-1\right)+\lambda_1\left(\mu-\sum_{i=1}^N p_i F_i\left(x_i\right)\right)\right]$ over $p_i$, we get that $p_i=\frac{\left.\exp \left[\lambda_1 F_i\left(x_i\right)\right)\right]}{Z}$  (Eq. 22).  Thus the max entropy can be expressed as $\begin{aligned}H_{\max }  =-\sum_{i=1}^N p_i \log p_i  =\lambda_0+1-\lambda_1 \sum_{i=1}^N p_i F_i\left(x_i\right) =\lambda_0+1-\lambda_1 \mu .\end{aligned}$  (Eq. 23) and  $-\frac{\partial H_{\max }}{\partial \mu}=: \frac{1}{\tau}$ (Eq. 24), where $\tilde{f}(x)=\mu$ , as stated below Eq. (8) of the submission.
>
> As a result, the expression in Eq. (4) does not explicitly depend on the estimation $\tilde{f}(x)$ because it is treated as a constant during the derivation. Intuitively, fair aggregation should depend on each client’s performance, i.e., their loss, rather than the ideal estimation term.
>
> We have stated the connection between Eq. (4) and $\tilde{f}(x)$ in the main paper.
>
> > Q.2 Authors are encouraged to provide ****an intuitive explanation for why maximizing constrained entropy can improve fairness**.** Right now, entropy-based aggregation just appears to be a black box to improve fairness
> >
>
> Thank you for your suggestion.
>
> Entropy is widely used to promote fairness in resource allocation [1] and data processing [2].
>
> **A motivation for its use in these scenarios is contained in the entropy concentration theorem** [3], which posits that maximum entropy provides the least biased solution for determining prior probabilities.
>
> However, the **existing** **entropy concentration theorem cannot be directly applied to federated learning settings.** In existing fairness applications, maximizing entropy without considering constraints will lead to a uniform distribution, where all users share equal resources. This conflicts with the natural heterogeneity in federated learning; aggregation with equal weights of all users cannot achieve the goal of fairness, i.e., equal performance across users. We address this challenge by introducing constraints to adjust weights, aligning aggregated performance with that of the ideal fair model. This constrained entropy approach focuses more on underperforming clients, assigning them higher weights and reducing performance unfairness.
>
> [Toy example]
>
> We present a toy example to demonstrate how entropy and fairness work. FedAvg uses uniform aggregation, which is equivalent to entropy without constraints.
>
> In particular, consider two clients with loss function $f_1\left(x_t\right)=2(x-2)^2, f_2\left(x_t\right)=\frac{1}{2}(x+4)^2$ respectively.
>
> Assuming $x_t=0$, w.l.o.g, we consider one single-step gradient decent with stepsize=1/4. Then $x_1^{t+1}=x_t-\lambda \nabla f_1\left(x_t\right)=2, x_2^{t+1}=x_t-\lambda \nabla f_2\left(x_t\right)=-1$.
>
> Thus, the aggregation probabilities for FedAvg, q-FFL, and FedEBA will be:
>
> ($\frac{1}{2}, \frac{1}{2}$), ($\frac{4}{13},\frac{4}{13}$), ($\frac{1}{1+e^{9/2}}, \frac{e^{9/2}}{1+e^{9/2}}$).
>
> Then the variance of $x_{t+1}$ will be:
>
> $Var_{AVG} = 2*(2.81)^2, Var_{q-FFL}=2*(2.52)^2, Var_{EBA+}=2*(0.6)^2$
>
> The corresponding variance rank is $Var_{EBA+}<Var_{q-FFL}<Var_{AVG}$.

---

> ### Author Response · Authors · 2024-11-21
> **Response to Reviewer acXk (2/4)**
>
> In this case, the normalized performance's entropy, after maxing the constrained entropy of aggregation probability, exhibits a relationship akin to variance (greater entropy corresponds to improved fairness).
> $\operatorname{Entropy}\left(f\left(x_{E B A+}^{t+1}\right)\right)=-\sum_{i=1}^2 \frac{f_i\left(x_{E B A+}^{t+1}\right)}{\sum_{j=1}^2 f_j\left(x_{E B A+}^{t+1}\right)} \log \left(\frac{f_j\left(x_{E B A+}^{t+1}\right)}{\sum_{i=j}^2 f_i\left(x_{E B A+}^{t+1}\right)}\right) \approx 0.996$, where $f_1\left(x_{E B A+}^{t+1}\right)=2 *(2.1)^2, f_2\left(x_{E B A+}^{t+1}\right)=\frac{1}{2} *(3.9)^2$.
> $\operatorname{Entropy}\left(f\left(x_{q-FFL}^{t+1}\right)\right)=-\sum_{i=1}^2 \frac{f_i\left(x_{q-FFL}^{t+1}\right)}{\sum_{j=1}^2 f_j\left(x_{q-FFL}^{t+1}\right)} \log \left(\frac{f_j\left(x_{q-FFL}^{t+1}\right)}{\sum_{i=j}^2 f_i\left(x_{q-FFL}^{t+1}\right)}\right) \approx 0.942$, where $f_1\left(x_{q-FFL}^{t+1}\right)=2 *(2.4)^2, f_2\left(x_{q-FFL}^{t+1}\right)=\frac{1}{2} *(3.6)^2$.
> $\operatorname{Entropy}\left(f\left(x_{AVG}^{t+1}\right)\right)=-\sum_{i=1}^2 \frac{f_i\left(x_{\text {avg}}^{t+1}\right)}{\sum_{j=1}^2 f_j\left(x_{\text {AVG}}^{t+1}\right)} \log \left(\frac{f_j\left(x_{\text {AVG}}^{t+1}\right)}{\sum_{i=j}^2 f_i\left(x_{\text {AVG}}^{t+1}\right)}\right) \approx 0.890$, where $f_1\left(x_{AVG}^{t+1}\right)=2 *(1.5)^2, f_2\left(x_{AVG}^{t+1}\right)=\frac{1}{2} *(4.5)^2$.
>
> Therefore, $Entropy(f(x_{EBA+}^{t+1}))>Entropy(f(x_{q-FFL}^{t+1}))>Entropy(f(x_{AVG}^{t+1}))$ and $Var_{EBA+}<Var_{q-FFL}<Var_{AVG}$.
>
> We have mentioned this in Appendix I.1 of the revised manuscript.
>
> [1]Johansson M, Sternad M. Resource allocation under uncertainty using the maximum entropy principle[J]. IEEE Transactions on Information Theory, 2005, 51(12): 4103-4117.
>
> [2]Celis L E, Keswani V, Vishnoi N. Data preprocessing to mitigate bias: A maximum entropy based approach[C]//International conference on machine learning. PMLR, 2020: 1349-1359.
>
> [3]Jaynes E T. On the rationale of maximum-entropy methods[J]. Proceedings of the IEEE, 1982, 70(9): 939-952.
>
> > Q3-1. In the alignment to improve global accuracy why are the pi's computed using the loss of the local models? (Line 312). In my understanding, since we are fixing x , when solving the inner problem, the p  should be computed using the loss of the global model, i.e, the expression in Line 351?
> >
>
> We believe you are referring to Line 320, where $p_i$ is computed based on the loss of the model after local updates.
>
> The difference between Line 320 and Line 351 lies in their objectives. In Line 320, the focus is on improving global accuracy, so the alignment update is calculated as the average of local updates. Here, $p_i$, representing client aggregation weights, is used to ensure that the aggregated model for each client closely aligns with the ideal fair gradient. Thus, it is computed using the loss of the local models.
>
> In contrast, in Line 351, $p_i$ is employed in the alignment update to approximate the ideal fair gradient. Since the ideal local update corresponds to a one-epoch update, the associated weights should be computed using the loss of the global model.
>
> To summarize:
>
> - When $p_i$ is used in standard model update aggregation, it is computed based on the loss of the local model.
> - When $p_i$ is used in the alignment term, it is computed based on the loss of the global model.
>
> > Q3-2. In the alignment to improve fairness why is the local optimization at clients changed (Eq. 11 and Line 11 in Algorithm 1)? In my understanding only global aggregation should change?
> >
>
> The aggregation and local alignment update (Eq.11) are both to improve fairness.
>
> If only the global aggregation is modified, it corresponds to FedEBA without alignment update. Specifically, Eq. (11) is derived from Eq. (7), which provides the outer-loop solution for the model parameters. Additionally, the alignment process is executed locally to refine each epoch’s local update direction, ensuring it does not deviate significantly from the fair direction, as illustrated in Figure 2.
>
> As a result, given the formulation of the model parameter updates (Eq. 7) and the alignment process performed during each local epoch, the local optimization process must be changed accordingly.

---

> ### Author Response · Authors · 2024-11-21
> **Response to Reviewer acXk (3/4)**
>
> > Q4. I don't follow how in Prac-FedEPA+ clients need to only communicate once. If we follow Algorithm 3 then it appears that communication can happen twice within a round: first in Line 6-8 and then again in Lines 18-22. I found this to be very confusing and authors should clarify what is the communication cost in their experiments.
> >
>
> Apologies for the unclear description of Algorithm 3. We would like to clarify the following:
>
> - In Algorithm 3, only **Line 5** (where the server broadcasts to clients) and **Line 12** (where clients upload to the server) involve communication. Therefore, the communication occurs only once per round.
> - For **Lines 13-25**, including **Lines 18-22**, all computations are performed on the server, as it has already received all the necessary information. Specifically, for **Lines 18-22**, the gradient information has already been obtained, and the aggregation can be directly performed on the server side using Eq. (8).
>
>  Thanks for pointing out this issue. The server-side gradient aggregation does not require additional communication. We have carefully revised the descriptions in the revised manuscript.
>
> > Q5. A is not defined in Theorem 5.1. Similarly w is not defined, although from context it appears to be the prior aggregation weights of client objectives.
> >
>
> Thank you for your suggestion. $A\geq0$ is a constant defined in Assumption 3, and $\mathbb{w}$ is the prior aggregation defined in Lemma H.1.
>
> Due to the page limit, we moved the Assumptions and Lemmas to the Appendix in our original submission. We have clarified these details in the revised manuscript.
>
> > Q6. It seems a little strange to see discussion on the lower bound of an upper bound in Remark 5.2. Authors should just state the worst possible convergence rate of FedEBA+ by considering that $\sum_{i=1}^m w_i^2\leq 1$.
> >
>
> Thank you for your suggestion. In the original submission, we provide the lower bound to help understand the performance of FedEBA+ ($\alpha=0$) when $w_i = \frac{1}{m}$, i.e., uniform aggregation. In this case, the bound is the same as FedAvg.
>
> We appreciate your suggestion and have stated the case when $\sum_{i=1}^m w_i^2\leq 1$ in Remark 5.2.
>
> > Q7-1. In Remark 5.3. it appears that to show convergence of α≠0 we need to assume k<<m. Is this correct? If so, authors should clearly mention this and explain why this is the case.
> >
>
> Yes, it is correct.
>
> In this paper, $K$ represents the local epoch times (in each communication round) and $m$ represents the client numbers, usually client numbers is larger than local epoch in the cross-device FL. For example, in our paper,$m=100, K=10$.
>
> We have revised the manuscript and mention this in our paper.
>
> > Q7-2. Also I don't follow the argument that the proposed alignment results in a faster convergence rate than FedAvg since both seem to be achieving O(1/mKT) rate.
> >
>
> The rate order of the proposed alignment remains the same as $\frac{1}{\sqrt{mKT}}$, and we state a faster convergence rate due to the constant term improvement.
>
> As for showing the improvement of alignment compared with FedAvg by setting $w_i=\frac{1}{m}$, when using the alignment update further ($\alpha \neq 0$),  term $\mathcal{O}(\frac{\sigma_L^2}{\sqrt{mKT}})$ changes into $\mathcal{O}(\frac{(1-\alpha)^2 \sigma_L^2+\alpha^2 \sqrt{K / m} \rho^2}{\sqrt{m K T}})$, since $0<\alpha<1,\frac{K}{m}<1,\rho \sim \sigma_L$, the constant term becomes smaller in the rate order, where a larger $\alpha$ leads to a faster convergence rate by degrading the influence of $\frac{1}{\sqrt{mKT}}$.
>
> > Q8. Why is FedMGDA+, PropFair and TERM incompatible in Table 2?
> >
>
> Thank you for your suggestion. We have provided the results of these three algorithms after carefully fine-tuning their hyperparameters and running them multiple times. These algorithms are not stable and were excluded from the original submission because they performed poorly with commonly used random seeds, such as 12345, that are used for other baselines in the paper.
>
> In the table below, we report the best 3 seeds out of 10 random trials, which demonstrates FedEBA+'s advantage over the other algorithms.
>
> |  | CIFAR-100 |  |  |  |
> | --- | --- | --- | --- | --- |
> | Algorithm | Global Acc. | Std. | Worst 5% | Best 5% |
> | FedMGDA+ | 31.34±0.12 | 16.61±0.29  | 0.74±0.12 | 65.21±1.15  |
> | PropFair | 30.85±0.07 | 16.52±0.24 | 0.29±0.04 | 64.33±0.71 |
> | TERM | 28.98±0.45 | 17.19 ±0.13 | 0.37±0.02 | 63.85±0.40 |
> | FedEBA+ | **31.98 ±0.30** | **13.75 ±0.16** | **1.12 ±0.05** | **67.94 ±0.54** |
>
> |  | Tiny-ImageNet |  |  |  |
> | --- | --- | --- | --- | --- |
> | Algorithm | Global Acc. | Std. | Worst 5% | Best 5% |
> | FedMGDA+ | 62.33±0.26 | 17.49±0.31 | 53.77±0.16 | 70.04±0.30 |
> | PropFair | 62.01±0.17 | 16.81±0.28 | 53.83±0.42 | 69.95±0.18 |
> | TERM | 61.29±0.37 | 19.36±0.94 | 52.92±0.65 | 69.82±0.44 |
> | FedEBA+ | **63.75 ±0.09** | **13.89 ±0.72** | **55.64 ±0.18** | **70.93 ±0.22** |

---

> ### Author Response · Authors · 2024-11-21
> **Response to Reviewer acXk (4/4)**
>
> The experimental setups are: FedMGDA+ with $\eta=1.0 , \epsilon=0.1$; PropFair with $M=5.0, thres=0.2$; TERM with $t=0.1$. The other setups, like learning rate and batch size, are the same as the submission.
>
> > Q9-1. Theoretically, we don't see **any effect of θ in Theorem 1**, which is a bit surprising to me.
> >
>
> Thank you for your insightful suggestion.
>
> In our Theorem 1, $\theta$  shows no theoretical impact on convergence since the gradient of ideal loss is uniformly bounded regardless of the choice of  $\theta$ .
>
> In particular, Assumption 4 bounds the gradient of the ideal loss and the practical aggregated gradient. It applies to both the ideal fair loss and the ideal global loss, allowing their gradients to be uniformly bounded and analyzed. While $\theta$ determines whether to estimate the ideal fair or global loss, both are uniformly bounded in the derivation, which is why $\theta$ doesn’t explicitly appear in the convergence rate.
>
> > Q9-2. In practice, I would also suggest moving Table 7 to the main text and **discuss the trade-off in setting θ** in more detail.
> >
>
> For the setting of $\theta$, it balances the communication cost and performance improvement.
>
> - A smaller $\theta$ increases communication costs. This happens because a smaller $\theta$ requires more frequent fair alignment updates, needing more communication rounds. However, it also leads to greater performance improvements.
> - According to Table 7, if the communication cost is affordable,  $\theta = 0$ should be chosen for optimal performance. Otherwise, we recommend using the Prac-FedEBA+ algorithm with the default $\theta = 15^\circ$, which requires no additional communication cost but with better performance than SOTA baselines.
>
> We have moved Table 7 into the main text and discussed it in detail according to the reviewer’s suggestion.

---

> ### Comment · Reviewer_acXk · 2024-11-23
> **Response to Rebuttal**
>
> Thank you for the rebuttal. Some points regarding the theory and algorithm are clearer now. However, I am still unsure about some aspects which I discuss below.
>
> 1. **Expression for aggregation probabilities:** This is still unclear to me. From Eq. (21) we have a relation between $\lambda_0$ and $\lambda_1$, i.e. $\lambda_0 = \log \sum_{i=1}^N \exp(\lambda_1 F_i(x_i)) - 1$. We can substitute this expression for $\lambda_0$ in the expression for entropy $H_\max = \lambda_0 + 1 - \lambda_1\sum_{i=1}^N p_iF_i(x_i)$ and find the value of $\lambda_1$ which maximizes entropy. Therefore why do we need to introduce the additional hyperparameter $\tau$ ?
>
> 2. **Changes made to the algorithm for ideal fair gradient vs ideal global gradient:** I would suggest to the authors to explicitly write the objective that they are solving in each case and then derive the gradient and explain any approximations they are making. Right now the authors are just writing down the direct gradient (Eq. 8 and Eq. 10) for the two cases which is making it hard to understand what exactly is the objective they are trying to solve in each case.
>
> 3. **Instability of FedMGDA+, PropFair and TERM:** Have other works also observed a similar phenomenon? In any case it would be good to mention how the hyperparameters of these algorithms were tuned and include plots where the algorithms diverged.
>
> 4. Why do we need $K \ll m$? Although it may be satisfied in practice, it is still an unnatural condition which other FL convergence works don't need.
>
> 5. **Effect of $\theta$:** I'm not sure what the authors mean by the gradient of ideal loss is uniformly bounded. Could you explain this in more detail or point me where in the proof this property is being used? Intuitively it seems like if you set $\theta$ to be too small or too large then you will just be doing ideal fair aggregation or ideal global aggregation always, which should have some impact on the convergence.
>
> Also please highlight any new edits to the paper in a different color so it becomes easier to see what exactly has changed.

---

> > ### Author Response · Authors · 2024-11-24
> > **Response to Reviewer acXK (1/3)**
> >
> > Dear Reviewer acXK, thank you very much for your prompt response and we appreciate the opportunity to further clarify your concerns.
> >
> > > Q1. **Expression for aggregation probabilities.** This is still unclear to me. From Eq. (21) we have a relation between $\lambda_0$ and $\lambda_1$, i.e. $\lambda_0 +1= log \sum_{i=1}^N exp(\lambda_1F_i(x))$ . We can substitute this expression for $\lambda_0$ in the expression for entropy $H_{\max } =\lambda_0+1-\lambda_1 \sum_{i=1}^N p_i F_i(x_i)$ and find the value of $\lambda_1$ which maximizes entropy. Therefore why do we need to introduce the additional hyperparameter $\tau$?
> > >
> >
> > We would like to clarify that the introduction of $\tau$ is intended to avoid notation confusion, as $\lambda_1$ represents the Lagrange multiplier and $\tau$ represents the inverse temperature, a hyperparameter.
> >
> > In particular, Eq. (21) is given by:
> > $\lambda_0 + 1 = \log \sum_{i=1}^N \exp \left(\lambda_1 F_i(x_i)\right) =: \log Z$.
> >
> > Substituting the expression for $\lambda_0$ into the equation $H_{\max} = \lambda_0 + 1 - \lambda_1 \sum_{i=1}^N p_i F_i(x_i)$, we get:
> >
> > $$
> > H_{\max } =\lambda_0+1-\lambda_1 \sum_{i=1}^N p_i F_i(x_i) \\
> > = \log \sum_{i=1}^N \exp \left(\lambda_1 F_i(x_i)\right)-\lambda_1\sum_{i=1}^N p_i F_i(x_i) \\
> > = \log \sum_{i=1}^N \exp \left(\lambda_1 F_i(x_i)\right)-\lambda_1\mu
> > $$
> >
> > To find the value of $\lambda_1$ that maximizes the entropy, we set $\lambda_1 = -\frac{\partial H_{\max}}{\partial \mu}$ for the maximum entropy.
> >
> > Since $\lambda_1$ is the Lagrange multiplier, we substitute it back into Eq. (22):
> > $p_i = \frac{\exp \left[\lambda_1 F_i(x_i)\right]}{Z},$ and obtain the solution for $p_i$ that maximizes the entropy.
> >
> > However, in practice, we cannot directly compute $-\frac{\partial H_{\max}}{\partial \mu}$. Therefore, we define the inverse temperature $\tau$ by:
> > $\lambda_1 = -\frac{\partial H_{\max}}{\partial \mu} := \frac{1}{\tau}.$
> >
> > Thus, the final expression for $p_i$ becomes:
> > $p_i = \frac{\exp \left(F_i(x_i)/\tau \right)}{Z}.$
> >
> > Since $\lambda_1$ represents the Lagrange multiplier in the derivation, it should be replaced back into Eq. (22) in a closed form. To avoid confusion, we define an additional hyperparameter, $\tau$, to get the final expression for $p_i$.
> >
> > This type of inverse temperature is commonly used in Energy-based Models (EBMs) as shown in works [1, 2]. The derivation closely follows the approach in [2, 3], which uses similar operations to express the Lagrange multiplier’s solution as a defined temperature, similar to the derivation in Eq. (3-1) of [3] and Eq. (25) in [2].
> >
> > [1]Yoon S, Hwang H, Kwon D, et al. Maximum Entropy Inverse Reinforcement Learning of Diffusion Models with Energy-Based Models[C]//NeurIPS 2024.
> >
> > [2]Bian Y, Rong Y, Xu T, et al. Energy-Based Learning for Cooperative Games, with Applications to Valuation Problems in Machine Learning[C]//International Conference on Learning Representations.
> >
> > [3]Jaynes E T. Information theory and statistical mechanics[J]. Physical review, 1957, 106(4): 620.

---

> > ### Author Response · Authors · 2024-11-24
> > **Response to Reviewer acXK (2/3)**
> >
> > > Q2. **Changes made to the algorithm for ideal fair gradient vs ideal global gradient:** I would suggest to the authors to explicitly write the objective that they are solving in each case and then derive the gradient and explain any approximations they are making. Right now the authors are just writing down the direct gradient (Eq. 8 and Eq. 10) for the two cases which is making it hard to understand what exactly is the objective they are trying to solve in each case.
> > >
> >
> > Thank you for your suggestion.
> >
> > In both cases, we are approximating the formula from Eq (7) $\frac{\partial L\left(x,p_i, \lambda_0, \lambda_1\right)}{\partial x} = (1-\alpha)\sum_i p_i \nabla F_i(x) + \alpha \nabla \tilde{f}(x)$ with a particular focus on finding ways to approximate $\nabla \tilde{f}(x)$.
> >
> > In more detail, in traditional Federated Learning (FL), the global model update is computed as: $x_{t+1} = x_t-\eta\Delta_t$,
> >
> > whereas in Gradient Descent (GD), the update rule is: $x_{t+1}=x_t-\mu\frac{\partial L(x)}{\partial x}$.
> >
> > Therefore, by substituting Eq. (7) into the GD method, we obtain:
> >
> > $\Delta_t = (1-\alpha)\sum_i p_i \nabla F_i(x) + \alpha \nabla \tilde{f}(x)$.
> >
> > Furthermore, in FL, each client $i$ performs multiple local epochs $K$ during each communication round $t$, thus
> >
> > $$
> > \Delta_t=(1-\alpha)\sum_ip_i\sum_{k=0}^{K-1}\nabla F_i(x_{t,k}^i;\xi_{t,k}^i) + \alpha \nabla \tilde{f}(x) = \sum_ip_i\left[(1-\alpha)\sum_{k=0}^{K-1}\nabla F_i(x_{t,k}^i;\xi_{t,k}^i)+\alpha \nabla \tilde{f}(x)\right] \qquad (1-1)
> > $$
> >
> >  Suppose $\Delta_i^t=x_{t,K-1}^t-x_{t,0}^t=\sum_{k=0}^{K-1}\nabla F_i(x_{t,k}^i;\xi_{t,k}^i)$, we have
> >
> > $$
> > \Delta_t = (1-\alpha)\sum_ip_i\Delta_i^t + \alpha \nabla \tilde{f}(x). \qquad (1-2)
> > $$
> >
> > - In the global model alignment update, $\nabla \tilde{f}(x_t)$ is estimated as $\nabla \tilde{f}(x_t) :=\nabla \tilde{f}^a(x_t)=\tilde{\Delta}^a_t= \frac{1}{|S_t|}\sum\nolimits_{i\in S_t}(x_{t,1}^i - x_{t,0}^i) ,$
> >
> >     which approximates the ideal global model’s update. Thus, according to Eq. (1-2), we can perform the global model update as:
> >
> >     $$
> >     \Delta_t = (1-\alpha) \sum\nolimits_{i\in S_t}p_i\Delta_t^i +\alpha \tilde{\Delta}^a_t,
> >     $$
> >
> >     as given by Eq.(8).
> >
> > - In the fair alignment update, $\nabla \tilde{f}(x_t)$ is estimated as
> >  $\nabla\tilde{f}(x_t):=\nabla \tilde{f}^b(x_t) = \sum\nolimits_{i\in S_t}p_i\sum_{k=0}^{K-1}\nabla\tilde{f}^b(x_{t,k}^i)$ ,
> >
> >     where $\nabla\tilde{f}^b(x_{t,k}^i)$  is estimated by
> >
> >      $\sum\nolimits_{i\in S_t}\tilde{p_i} \nabla F_i(x_t)$
> >     as given by Eq. (10), which represents the ideal fair gradient. This gradient is the same for all participating clients.
> >
> >     Thus, according to Eq. (1-1), we can perform the fair model update as:
> >
> >     $$
> >     \Delta_t= \sum_ip_i\left[(1-\alpha)\sum_{k=0}^{K-1}\nabla F_i(x_{t,k}^i;\xi_{t,k}^i)+\alpha \nabla \tilde{f}(x)\right] = \sum_ip_i\sum_{k=0}^{K-1}\left[(1-\alpha)\nabla F_i(x_{t,k}^i;\xi_{t,k}^i) + \alpha \nabla \tilde{f}^b(x_{t,k}^i)  \right] =\sum_ip_i\sum_{k=0}^{K-1}h_{i,k}^t,
> >     $$
> >
> >     where $h_{i,k}^t=(1-\alpha)\nabla F_i(x_{t,k}^i;\xi_{t,k}^i) + \alpha \nabla \tilde{f}^b(x_{t,k}^i)$ , as given by Eq.(11).
> >
> >
> > Motivated by the reviewer’s suggestion, we have included these details (marked by red color) in the main text on Page 6 and Page 7 to make it easier to understand.

---

> > ### Author Response · Authors · 2024-11-24
> > **Response to Reviewer acXK (3/3)**
> >
> > > Q3. Instability of FedMGDA+, PropFair and TERM: Have other works also observed a similar phenomenon? In any case it would be good to mention how the hyperparameters of these algorithms were tuned and include plots where the algorithms diverged.
> > >
> >
> > Other works have not explicitly stated the instability problem but have shown that the algorithms sometimes achieve poor performance, as observed in [1, 2].
> >
> > Some studies have noted that FedMGDA+ performs relatively poorly, as shown in Table 1 of [1], and TERM shows relatively low performance in Table 2 and Figure 3 of [2], although neither directly mentions the instability phenomenon. We find that performance instability is especially noticeable in the CIFAR-100 and Tiny-ImageNet datasets. However, in CIFAR-10, as reported in [1, 2], the performance is also suboptimal.
> >
> > Hyperparameters tuning:
> >
> > - For FedMGDA+, we tune the hyperparameter $\epsilon=\[0, 0.03, 0.08, 0.1, 1.0]$, and report the best-performing value of $\epsilon=0.1$ in the setting.
> > - For PropFair, we tune the hyperparameter $M=\[0.2, 1.0, 5.0\]$, and report the best-performing value of $M=5.0$ in the setting.
> > - For TERM, we tune the hyperparameter $\tau=\[0.1, 0.5, 0.8, 1, 5\]$, and report the best-performing value of $\tau=0.1$ in the setting.
> >
> > For these baselines, the seeds used were seed=1234 and seed=12345, which achieved relatively low performance. However, seeds {1, 119, 512} achieved relatively good performance according to our results.
> >
> > The plot in Figure 15 of the updated submission shows that the algorithms achieve unsatisfactory accuracy in this setting.
> >
> > [1]Pan Z, Li C, Yu F, et al. FedLF: Layer-Wise Fair Federated Learning[C]//AAAI 2024.
> >
> > [2]Pan Z, Wang S, Li C, et al. Fedmdfg: Federated learning with multi-gradient descent and fair guidance[C]//AAAI 2023.
> >
> > > Q4. Why do we need $K≪m$? Although it may be satisfied in practice, it is still an unnatural condition which other FL convergence works don't need.
> > >
> >
> > Sorry for the misunderstanding.
> >
> > We would like to clarify that the FL algorithm converges in order $\mathcal{O}(\frac{1}{\sqrt{mKT}})$  without the need of condition $K≪m$. The convergence rate is $\mathcal{O}(\frac{1}{\sqrt{mKT}})$  no matter the relation between $K$ and $m$, because the term $\sqrt{\frac{K}{m}}$ is just a constant term in the order.
> >
> > We use this condition to show the advantage of alignment over aggregation-only methods like FedAvg. In particular, when the condition $K≪m$ is satisfied, the constant term in the order will be smaller than that of FedAvg, thus achieving a faster convergence rate than FedAvg.
> >
> > > Q5. Effect of θ: Could you explain this in more detail or point me where in the proof this property is being used? Intuitively it seems like if you set θ to be too small or too large then you will just be doing ideal fair aggregation or ideal global aggregation always, which should have some impact on the convergence.
> > >
> >
> > Thank you for your insightful suggestion. We would like to explain this in more detail and point out where the proof for this property is used.
> >
> > - **[Explanation]** We use Assumption 4 to bound both the ideal fair update and the ideal global update, since both the aggregation and update methods can be derived from Eq. (7)
> >
> >     $\frac{\partial L\left(x,p_i, \lambda_0, \lambda_1\right)}{\partial x} = (1-\alpha)\sum_{i=1}^m p_i \nabla F_i(x) + \alpha \nabla \tilde{f}(x)$.
> >
> >     In detail, $\nabla \tilde{f}(x)$ is approximated by either $\nabla \tilde{f}^a(x)$ (ideal global update) or $\nabla \tilde{f}^b(x)$ (ideal fair gradient). In particular,
> >     $\nabla \tilde{f}(x) =
> >     \begin{cases}
> >     \nabla \tilde{f}^a(x), & \text{if } \arccos\left( \frac{\mathbf{L}, \mathbf{r}}{\|\mathbf{L}\|\|\mathbf{r}\|} \right) \leq \theta \newline
> >     \nabla \tilde{f}^b(x). & \text{if } \arccos\left( \frac{\mathbf{L}, \mathbf{r}}{\|\mathbf{L}\|\|\mathbf{r}\|} \right) > \theta
> >     \end{cases}$
> >
> >     In the derivation, we use Assumption 4 to bound $E\| \nabla \tilde{f}(x) - \nabla f(x)\|^2 \leq \rho^2$, uniformly bounding both cases. Regardless of the approximation for $\nabla \tilde{f}(x)$, it is always bounded by  $\rho$, ensuring uniformity for all approximations of $\nabla \tilde{f}(x)$.
> >
> >
> > - **[Proof]** The proof of this property is provided in the derivation of Lemma H.5, from Eq (89) to Eq (90) , and in the derivation of $A_1$ from Eq (105) to Eq (107).
> >
> > In addition, as the reviewer implied, one might find a tighter bound for the different approximation errors, which would lead to varying results depending on the approximation method. In other words, the choice of $\theta$ will affect the convergence bound. However, in our paper, we found that the uniform bound from Assumption 4 is sufficient to guarantee the convergence of the algorithm.
> >
> > > Highlight any new edits in a different color.
> > >
> >
> > Thank you very much for your constructive suggestion. We have updated the submission and highlighted the modifications in red for the reviewers' convenience.

---

> ### Comment · Reviewer_acXk · 2024-11-25
> **Rebuttal response continued**
>
> Thank you for your response. Some parts are still unclear as I discuss below:
>
> 1. **Expression for aggregation probabilities**
>
> The authors write
> > To find the value of  that maximizes the entropy, we set $\lambda_1 = - \frac{\partial H_{\max}}{\partial \mu}$ for the maximum entropy.
>
> My question is why is this step necessary? Since $\sum_{i}p_iF_i(x_i) = \mu$ we know that
>
> $$ \frac{\sum_{i=1}^N \exp(\lambda_1 F_i(x_i))F_i(x_i)}{\sum_{i=1}^N \exp(\lambda_1 F_i(x_i))} = \mu. $$
>
> So we can solve the above equation to find $\lambda_1$ in terms of $\mu$ and $\\{F_i(x_i)\\}_{i=1}^N$ . Why are we not doing this?
>
> 2 . **Changes made to the algorithm for ideal fair gradient vs ideal global gradient:**
>
> Thank you for the clarification, but my question still remains.
>
> In the global model alignment case, clients are performing the following optimization:
> $$ x_{t,k+1}^i = x_{t,k}^i - \eta_l \nabla F_i(x_{t,k}^i, \xi_i)$$
> and in the fair alignment case, clients are performing:
> $$ x_{t,k+1}^{i} = x_{t,k}^{i} - \eta_l h_{t,k}^{i}$$
> where $h_{t,k}^{i} = (1-\alpha) \nabla F_i(x_{t,k}^i, \xi_i) + \alpha \tilde{g}^{b,t}$
>
> My question is **why is the local optimization changing in the fair alignment case?**
>
> The local optimization that occurs for fair alignment (i.e, $h_{t,k}^{i} = (1-\alpha) \nabla F_i(x_{t,k}^i, \xi_i) + \alpha \tilde{g}^{b,t}$) looks similar to the idea of using a control variate that has appeared in several works starting from SCAFFOLD [a] (see Equation 10 in Algorithm 1) to deal with the problem of client-drift. But in the context of this work, I don't see the theoretical motivation for using such a control-variate based idea.
>
> In fact, the Prac-FedEBA+ algorithm where local optimization remains the same for both fair and global alignment is the one which I expect to see and is the one which is more intuitive to me.
>
> 3. **$K \ll m$ condition**
>
> The authors write
> > The convergence rate is $O(1/\sqrt{mKT})$ no matter the relation between $K$ and $m$ because the term $\sqrt{K/m}$ is just a constant term in the order
>
> Why is it a constant order (unless authors assume so)? Other FL works for instance SCAFFOLD, do not place any restriction on $K$ so $K$ could be any polynomial of $m$ for instance, $K = \mathcal{O}(m^3)$. In this case $\sqrt{K/m} = \mathcal{O}(m)$ which is not constant order.
>
>
>
> **References**
>
> [a] Karimireddy, Sai Praneeth, et al. "Scaffold: Stochastic controlled averaging for on-device federated learning." arXiv preprint arXiv:1910.06378 2.6 (2019).

---

> > ### Author Response · Authors · 2024-11-28
> > **Response to Reviewer acXK (1/2)**
> >
> > Dear Reviewer acXK, thank you for your insightful suggestions. We have carefully revised the manuscript based on your feedback, and the changes have been highlighted in red. Please find our responses to your remaining concerns below.
> >
> > > **Q1. Expression for aggregation probabilities**
> > The authors write
> > ”To find the value of that maximizes the entropy, we set $\lambda_1 = -\frac{\partial H_{\max}}{\partial \mu}$ for the maximum entropy.”
> > My question is why is this step necessary? Since $\sum_ip_iF_i(x_i)=\mu$ we know that$\frac{\sum_i\exp(\lambda_1F_i(x_i))F_i(x_i)}{\sum_i\exp(\lambda_1F_i(x_i))} = \mu$
> > So we can solve the above equation to find $\lambda_1$ in terms of $\mu$ and $\{F_i(x_i)\}_{i=1}^N$ . Why are we not doing this?
> > >
> >
> > Thank you very much for your constructive suggestion. Sorry for the misleading, $\lambda_1$ is used as a hyperparameter in this paper. We cannot use this equation $\frac{\sum_i\exp(\lambda_1F_i(x_i))F_i(x_i)}{\sum_i\exp(\lambda_1F_i(x_i))} = \mu$ to directly obtain $\lambda_1$ since $\mu$ involves the variable $p_i$. However, by taking $\lambda_1$ as hyperparameter, we can directly conclude the aggregation probability $p_i=\frac{\exp \left[ F_i(x_i)/\tau \right]}{\sum_{j=1}^N \exp(F_j(x_j)/\tau)}$.
> > To make it clearer, **we have simplified the expression during the derivation.** In particular,
> >
> > - we directly use the hyperparameter $\tau$ (introduced in Eq (4) of the main paper) to replace the redundant hyperparameter $\lambda_1$ into the optimization problem. In detail, we change the Eq. (18) into
> >
> >     $$
> >     L\left(p, \lambda_0; \frac{1}{\tau}\right):=-\left[\sum_{i=1 }^N p_i \log p_i   +\lambda_0\left(\sum_{i = 1}^N p_i-1\right)+\frac{1}{\tau}\left(\mu-\sum_{i = 1}^N p_i F_i(x_i)\right)\right]
> >     $$
> >
> >
> > **Corresponding changes made in the updated version:**
> >
> > - Compared with the derivation in the original submission, Eq (19) to Eq. (22) remains unchanged, and Eq. (22) will directly get the proposed aggregation probability.
> > - Eq (23) and Eq (24) of the original submission have been removed.
> >
> > We have updated the derivation into a simplified version on Page 19 of the updated submission and modified Eq.(6) on Page 5 correspondingly in the updated version.
> >
> > > **Q2. Changes made to the algorithm for ideal fair gradient vs ideal global gradient:**
> > Thank you for the clarification, but my question still remains.In the global model alignment case, clients are performing the following optimization: $x_{t,k+1}^i=x_{t,k}-\eta_l\nabla F_i(x_{t,k}^i;\xi_i)$, and in the fair alignment case, clients are performing: $x_{t,k+1}^i=x_{t,k}-\eta_lh_{i,k}^t$, where $h_{i,k}^t=(1-\alpha)\nabla F_i(x_{t,k}^i;\xi_{t,k}^i) + \alpha \nabla g^{b,t}$ .\
> > My question is **why is the local optimization changing in the fair alignment case?**
> > The local optimization that occurs for fair alignment (i.e,$h_{i,k}^t=(1-\alpha)\nabla F_i(x_{t,k}^i;\xi_{t,k}^i) + \alpha \nabla g^{b,t}$ ) looks similar to the idea of using a control variate that has appeared in several works starting from SCAFFOLD [a] (see Equation 10 in Algorithm 1) to deal with the problem of client-drift. But in the context of this work, I don't see the theoretical motivation for using such a control-variate based idea.
> > In fact, the Prac-FedEBA+ algorithm where local optimization remains the same for both fair and global alignment is the one which I expect to see and is the one which is more intuitive to me.
> > >
> >
> > Thank you very much for recognizing the proposed Prac-FedEBA+. We would like to show 1) how the proposed FedEBA+ differs from SCAFFOLD and 2) the theoretical motivation for local optimization in the fair alignment case.
> >
> > - ***[Difference between FedEBA+ and SCAFFOLD]:***
> >     - The local update of FedEBA+ is $x_i \leftarrow x_i - \eta_l \left((1 - \alpha) \nabla F_i(x_{t,k}^i; \xi_i) + \alpha \tilde{g}^{b,t}\right)$, where $\tilde{g}^{b,t} = \sum_{i \in S_t} \tilde{p_i} \nabla F_i(x_t)$ and $\tilde{p_i}=\frac{\exp [F_i(x_t) / \tau)]}{\sum_{j\in S_t} \exp [F_j(x_t) / \tau]}$ is the **proposed fair reweighted aggregation.** Thus FedEBA+ corrects the local update **towards the fair update direction**.
> >     - The local update of SCAFFOLD is $x_i \leftarrow x_i-\eta_l\left(\nabla F_i(x_{t,k}^i;\xi_i)+(c-c_i)\right)$, where $\Delta c=\frac{1}{|S_t|}\sum_{i\in S_t}\Delta c_i$ is the **uniform aggregation** of control variable’s updates, Thus SCAFFOLD corrects the local update **towards the global model’ unshifted update direction**.

---

> > ### Author Response · Authors · 2024-11-28
> > **Response to Reviewer acXK (2/2)**
> >
> > - ***[Theoretical motivation for local optimization changing in the fair alignment case]:***
> >     - **[Motivation]** Refining each local update is more effective for correcting the update direction than correcting on server, since multiple local updates will aggravate the update shift~[1,2].
> >     - **[Technical Motivation]** The changes in local optimization for the fair alignment case arise from extending the centralized GD method to the FL setting.
> >         - For traditional FL, the extension of the update rule from centralized GD to FL is
> >
> >             $$
> >             x_{t+1}= x_t-\frac{\partial L\left(x\right)}{\partial x}=x_t-\eta\sum_ip_i\nabla F_i(x_i) =x_t-\eta\sum_ip_i\sum_{k=0}^{K-1}\nabla F_i(x_k^i;\xi_i)
> >             $$
> >
> >             Thus, the local update rule is
> >
> >             $$
> >             x_{t,k}^i=x_{t,k-1}^{i}-\eta_l \nabla F_i(x_{t,k}^i;\xi_i)
> >             $$
> >
> >         - Similarly, for FedEBA+, based on  Eq. (7) $\frac{\partial L\left(x,p_i\right)}{\partial x} = (1-\alpha)\sum_i p_i \nabla F_i(x) + \alpha \nabla \tilde{f}(x)$ , the extension of the update rule from centralized GD to FL is
> >
> >             $$
> >             x_{t+1}=x_t-\eta\sum_ip_i\left[(1-\alpha)\sum_{k=0}^{K-1}\nabla F_i(x_{t,k}^i;\xi_{t,k}^i) + \alpha \nabla \tilde{f}^b(x)\right] \\
> >             =x_t-\eta\sum_ip_i\sum_{k=0}^{K-1}\left[(1-\alpha)\nabla F_i(x_{t,k}^i;\xi_{t,k}^i)+\alpha \nabla \tilde{f}^b(x_{t,k}^i)\right]
> >             $$
> >
> >             Thus, the local update rule is
> >
> >             $$
> >             x_{t,k}^i=x_{t,k-1}^{i}-\eta_l h_{t,k}^i =  x_{t,k-1}^{i}-\eta_l \left[(1-\alpha)\nabla F_i(x_{t,k}^i;\xi_{t,k}^i)+\alpha \nabla \tilde{f}^b(x_{t,k}^i)\right]
> >             $$
> >
> >
> >     In summary, the theoretical motivation stems from generalizing Eq. (7) to the FL setting, where the ideal gradient $\nabla \tilde{f}(x)$ is also expressed by multiple local updates. Thus the local optimization is changed.
> >
> >
> > [1]Karimireddy S P, Rebjock Q, Stich S, et al. Error feedback fixes signsgd and other gradient compression schemes[C]//International Conference on Machine Learning. PMLR, 2019: 3252-3261.
> >
> > [2]Karimireddy S P, Kale S, Mohri M, et al. Scaffold: Stochastic controlled averaging for federated learning[C]//International conference on machine learning. PMLR, 2020: 5132-5143.
> >
> > > **Q3. K≪m condition**
> > The authors write “The convergence rate is O(1/mKT) no matter the relation between K and m because the term K/m is just a constant term in the order”.
> > Why is it a constant order (unless authors assume so)? Other FL works for instance SCAFFOLD, do not place any restriction on K so K could be any polynomial of m for instance, K=O(m3). In this case K/m=O(m) which is not constant order.
> > >
> >
> > Thank you for your comment, and we apologize for any confusion caused by our earlier expression.
> >
> > To clarify, the convergence rate is independent of any specific relationship between $K$ and $m$, and no restrictions are placed on $K$ for the convergence guarantee.
> >
> > Regarding the imprecise statement about $K/m$ being a constant, we intended to convey that for typical values of $K$ (e.g., {5, 10}) and $m$ (e.g., {100, 500}) in cross-device FL, the convergence rate primarily depends on the total number of communication round $T$, which dominates the order of complexity in the analysis.
> >
> > Based on the reviewer's suggestion, we have carefully revised the discussion in Remark 5.3 of the updated manuscript to clarify the expression.

---

> > ### Author Response · Authors · 2024-12-02
> > **Looking forward to your reply**
> >
> > Dear Reviewer acXk,
> >
> > We are truly grateful for your insightful review, which has been instrumental in improving the quality of our manuscript. We have taken your feedback into account and revised our paper accordingly. As the discussion phase draws to a close, we want to ensure that we have addressed all your concerns comprehensively. If there are any additional points that you believe would contribute to the enhancement of our manuscript, please do not hesitate to let us know.
> > Additionally, we would deeply appreciate your reconsideration of the review score, should you find the revisions satisfactory.
> >
> > Warmest regards,
> >
> > Authors

---

> > > ### Comment · Reviewer_acXk · 2024-12-03
> > >
> > > Thank you for your last clarification. Based on all the discussion during the rebuttal, most of my concerns have been addressed to some extent so I am raising my score to 6.

---

> > > > ### Author Response · Authors · 2024-12-03
> > > > **Thank you for raising the score**
> > > >
> > > > Dear Reviewer acXk,
> > > >
> > > > Thank you very much for raising the score. We have incorporated the discussion from the rebuttal into the manuscript. We sincerely appreciate your time, effort, and thoughtful insights.

---

### Official Review · Reviewer_oeex · 2024-11-09

**Soundness:** 3
**Presentation:** 3
**Contribution:** 3
**Rating:** 5
**Confidence:** 3

**Summary:**

This paper studies an important problem, i.e,. to ensure fairness of federated learning while striving to maintain the accuracy, for scenarios where heterogeneity hurts the performance of FL. Their approach was based on an entropy-based formulation and they show theoretical results such as convergence as well as experiments which to an extent verifies the effectiveness of the proposed algorithm.

The paper appears to be fairly well-written and well motivated. The main concern is the the limited novelty and significance. The experimental results are not impressive either.

**Strengths:**

1. The significance of federated learning is well presented and the motivation for ensuring fairness is well-motivated.

2. The theoretical analysis appears correct.

3. The experiments do confirm the improvement, to an extent.

**Weaknesses:**

1. While the exact formulation has not been  published in the literature, the formulation appears to be fairly straightforward and does not seem to be a significant contribution to the field. The analysis follows the standard analysis.

2. The experimental results, while they show some improvement, do not appear impressive. The actual significance of the proposed framework is not convincing.

**Questions:**

1. Could you summarize the (non-trivial) significance of technical contributions, compared with existing literature?

2. Could you elaborate on the significance of the experimental results? The numbers (such as accuracy) appears low and gives the doubt that the paper did not use strong baselines.

3. This paper severely lacks references from ICML, NeurIPS, ICLR, AISTATS, especially from recent years. The authors are encouraged to provide a thorough literature study from the mainstream ML venues especially from 2022-2024, to help convince readers on the novelty and significance of this submission.

---

> ### Author Response · Authors · 2024-11-21
> **Response to Reviewer oeex (1/2)**
>
> Dear reviewer oeex, we would like to thank you for your time spent reviewing our paper and for providing constructive comments. Please kindly find our responses to your raised questions below:
>
> > Q1. Could you summarize the (non-trivial) significance of technical contributions, compared with existing literature?
> >
>
> Thank you for your suggestion. The technical contributions can be elaborated in the following three aspects:
>
> 1. **[The first algorithm that improves both fairness and accuracy]**
>
>     Our algorithm is **the first** to explicitly model the influence of fairness and global accuracy and improve them **simultaneously**.
>
> 2. **[Non-trivial objective formulation and solution]**
>
>     In the FL fairness objective, the general objective (2) is non-trivial to explicitly formulate, and **no previous work has explicitly formulated it**. We address this by modeling it as objective (6), a bi-level optimization objective with an analytic inner-loop solution. This approach makes it more practical compared to traditional bi-level optimization methods, which require computationally expensive inner- and outer-loop solutions.
>
> 3. **[Effectiveness of fair aggregation]**
>
>     While a few aggregation methods aim for fairness [1,2], **existing works have different formulations and are less effective than ours**. For example, when compared with q-FFL, as shown in Table 1 and Table 2, our method demonstrates superior performance.
>
> 4. **[Novel constrained entropy]**
>
>     **Existing entropy-type aggregation methods [3,4] have different objectives from fair FL**, focusing on accuracy rather than fairness. To our knowledge, the constrained entropy approach (Eq. 3) is the first to address fairness in FL.
>
>     **Modeling entropy in fair FL is challenging**. Existing entropy models for fairness, such as in resource allocation, result in uniform aggregation distributions, which fail to address performance fairness for FL. To address this, we introduce the entropy of aggregation with a unique constraint. This constraint shifts the focus toward underperforming clients, enabling *weighted fair aggregation distributions*.
>
>
> [1]Li T, Sanjabi M, Beirami A, et al. Fair Resource Allocation in Federated Learning[C]//International Conference on Learning Representations.
>
> [2]Mollanejad A, Navin A H, Ghanbari S. Fairness-aware loss history based federated learning heuristic algorithm[J]. Knowledge-Based Systems, 2024, 288: 111467.
>
> [3] Huang, W., Shi, Z., Ye, M., Li, H. & Du, B. Self-Driven Entropy Aggregation for Byzantine-Robust Heterogeneous Federated Learning. ICML 2024
>
> [4]Herath C, Liu X, Lambotharan S, et al. Enhancing Federated Learning Convergence with Dynamic Data Queue and Data Entropy-driven Participant Selection[J]. IEEE Internet of Things Journal, 2024.
>
> > Q2. Could you elaborate on the significance of the experimental results? The numbers (such as accuracy) appears low and gives the doubt that the paper did not use strong baselines.
> >
>
> We would like to clarify the following points:
>
> 1. **[SOTA Baselines.]**
> The baselines used in our experiments are widely adopted and considered state-of-the-art (SOTA) in performance-fairness studies. It is important to note that fairness is a broader field encompassing various orthogonal directions, such as *Proportionality* [1] and *Disparity* [3]. Our work primarily focuses on performance fairness via variance. The selected baselines represent SOTA for the experimental metrics and setup.
> Although rare recent algorithms are designed for our setup (performance fairness), we implement the very recent work [1] in our setting to show the advantage of our method, as shown in the table below.
>
>
>     |  | FashionMNIST |  |  |  |
>     | --- | --- | --- | --- | --- |
>     | Algorithm | Global Acc. | Var. | Worst 5% | Best 5% |
>     | Rank-Core-Fed [1] | 85.54 ±0.33 | 58.19±2.83 | 67.80 ±0.55 | 96.60 ±0.40 |
>     | FedEBA+ | 87.50 ±0.19 | 43.41 ±4.34 | 72.07 ±1.47 | 95.91 ±0.19 |
>
>     |  | CIFAR-10 |  |  |  |
>     | --- | --- | --- | --- | --- |
>     | Algorithm | Global Acc. | Var. | Worst 5% | Best 5% |
>     | Rank-Core-Fed [1] | 67.15±1.12 | 87.02 ±2.46 | 45.41 ±0.62 | 85.82 ±0.20 |
>     | FedEBA+ | 72.75 ±0.25 | 68.71 ±4.39 | 55.80 ±1.28 | 86.93 ±0.52 |

---

> ### Author Response · Authors · 2024-11-21
> **Response to Reviewer oeex (2/2)**
>
> 2. **[Significant Fairness and Accuracy Improvement Simultaneously]**
>
>     More importantly, **this paper emphasizes improving both fairness and accuracy** **simultaneously**. When considering fairness and accuracy both, Figure 3(a) clearly demonstrates the advantages of our approach, and Table 17 in Appendix M shows that FedEBA+ achieves nearly 2–4× better performance compared to other methods.
>
>     As a fair FL work, **fairness holds greater importance than accuracy**. Specifically, considering fairness, the variance improvement is 3× on FashionMNIST (11.5 variance improvement) and 1.5× on CIFAR-10 (9.9 variance improvement) compared to the best-performing baseline.
>
>     Regarding accuracy improvement, our method achieves a 3.89% improvement over the best-performing baseline on the CIFAR-10 dataset and a 1.33% improvement on the Tiny-ImageNet dataset. **These are significant improvements for fair FL works, where baselines often show minimal accuracy gains or even sacrifice accuracy to enhance fairness.**
>
> > Q3. This paper severely lacks references from ICML, NeurIPS, ICLR, AISTATS, especially from recent years. The authors are encouraged to provide a thorough literature study from the mainstream ML venues especially from 2022-2024, to help convince readers on the novelty and significance of this submission.
> >
>
> Thank you for your suggestions. In our original submission, we include baselines up to 2023, such as PropFair and FOCUS. In the revised version, we have incorporated fair FL works from 2024 conferences, as shown below.
>
> In particular, recent works address a diverse range of fairness objectives, such as *Proportionality* [1], *Disparity* [3], *Stability* [2], and fairness in vertical FL [4]. In particular:
>
> - [1] provides explainable proportional fairness guarantees to agents in general settings, where the error rates of the agents are proportional to the size of their local data.  It focuses on data-sensitive fairness such as better performance for clients with larger datasets, while our work emphasizes performance fairness aiming to make uniform performance for clients, making the objectives distinct. As shown in the additional experimental results in our reply to your Q2-1, our method consistently outperforms [1].
> - [2] primarily focuses on establishing theoretical bounds to show how clients' altruistic behaviors and the configuration of friend-relationship networks influence achievable egalitarian fairness. [3] offers an information-theoretic perspective on group fairness trade-offs in federated learning, utilizing partial information decomposition to identify unfairness.  These works aim to establish theoretical frameworks for analyzing fairness and trade-offs, using perspectives from information theory and game theory, rather than providing algorithms for fairness as the experiments in these works only support the theoretical findings.
> - [4] discusses fairness in vertical FL by learning fair and unified representations, where feature fields are decentralized across different platforms. In contrast, our work focuses on horizontal FL and compares our results with state-of-the-art horizontal FL fairness algorithms as reported in the literature.
>
> Motivated by your suggestion, we have incorporated this discussion into the Related Work section of our revised manuscript to provide a more comprehensive perspective.
>
> [1]Chaudhury B R, Murhekar A, Yuan Z, et al. Fair federated learning via the proportional veto core[C]//Forty-first International Conference on Machine Learning 2024.
>
> [2]Gao J, Wang Z, Zhao X, et al. Does Egalitarian Fairness Lead to Instability? The Fairness Bounds in Stable Federated Learning Under Altruistic Behaviors[C]//The Thirty-eighth Annual Conference on Neural Information Processing Systems 2024.
>
> [3]Hamman F, Dutta S. Demystifying Local & Global Fairness Trade-offs in Federated Learning Using Partial Information Decomposition[C]//The Twelfth International Conference on Learning Representations 2024.
>
> [4]Fan Z, Fang H, Wang X, et al. Fair and Efficient Contribution Valuation for Vertical Federated Learning[C]//The Twelfth International Conference on Learning Representations 2024.

---

> ### Author Response · Authors · 2024-12-03
> **Looking forward to your reply**
>
> Dear Reviewer oeex,
>
> Thank you for your time and efforts in reviewing our paper. As the discussion period ends, we kindly ask you to review our responses and confirm if they address your concerns. Additionally, we would be sincerely grateful for your reconsideration of the review score.
>
> Warmest regards,
>
> Authors

---

### Author Response · Authors · 2024-11-21
**General Response**

We sincerely thank the reviewers for taking the time to read our paper and for providing detailed feedback and valuable suggestions. We have revised the manuscript accordingly, with all changes highlighted in red in the updated version.

Before summarizing the updates made during the rebuttal, we would like to highlight the key contributions of our work:

- **[Challenge]** Addressing an under-explored challenge: achieving both improved fairness and enhanced global model performance simultaneously.
- **[Method]** Proposing a novel bi-level optimization framework for fair FL, built on a constrained maximum entropy model. This work is the first to extend the concept of entropy to FL for fairness analysis.
- **[Analysis]** Providing theoretical guarantees for both convergence and fairness in FedEBA+.
- **[Experiment]** Extensive experiments have shown that FedEBA+ surpasses state-of-the-art fair FL algorithms, significantly improving fairness and global model accuracy simultaneously across various datasets and model architectures.

For convenience, we list these changes below:

- As suggested by *Reviewer oeex*,
    - we added the recent Rank-Core-Fed [1] algorithm as a baseline in **Table 1** to highlight our method's advantages.
    - we expanded the discussion of recent literature in **Appendix A** to better illustrate the novelty and significance of our work.
- As suggested by *Reviewer acXK*,
    - we included results of FedMGDA+, PropFair, and TERM on CIFAR-100 and Tiny-ImageNet datasets in **Table 2**, supported by extensive randomized seed experiments.
    - we improved writing in **Section 5** and clarified the description of **Algorithm 3** to prevent misunderstandings.
    - we added **Table 3** to the main paper to discuss the trade-off between $\theta$ and communication costs.
- As suggested by *Reviewer 591F and  Reviewer Tsda*,
    - we included a discussion in **Appendix A** highlighting the differences and advantages of our approach compared to existing aggregation methods.

In addition to submitting a revised version, we reply to each reviewer individually to address their questions and concerns. We hope that these responses together with the revised manuscript clear up any confusion and resolve all issues that the reviewers had, and that you will consider increasing your rating for our paper.

[1]Chaudhury B R, Murhekar A, Yuan Z, et al. Fair federated learning via the proportional veto core[C]//Forty-first International Conference on Machine Learning. 2024.

---

### Author Response · Authors · 2024-12-04
**To Reviewers and AC**

Dear Reviewers and AC,

We deeply appreciate your valuable time and effort in reviewing our paper.

As the discussion draws to a close, we would like to provide a concise summary of the updates made to our paper during the rebuttal process for your convenience:

- **Table 1:** Added the recent Rank-Core-Fed algorithm as a baseline to highlight our method's advantages, addressing Reviewer oeex’s concerns.
- **Appendix A:** Expanded the discussion of recent literature to emphasize the novelty and significance of our work, addressing Reviewer oeex’s concerns.
- **Table 2:** Included results for FedMGDA+, PropFair, and TERM on CIFAR-100 and Tiny-ImageNet datasets, supported by extensive randomized seed experiments, addressing Reviewer acXk’s concerns.
- **Paper Modifications:**
    1. Added details to **Section 5** and clarified **Algorithm 3**.
    2. Moved **Table 3** to the main paper to discuss the trade-off between $\theta$ and communication costs.
    3. Simplified the derivation of Eq. (3) in **Appendix C.1**, motivated by Reviewer acXk’s comments.
- **Table 15 and** [Tables in Response](https://openreview.net/forum?id=yqST7JwsCt&noteId=k5M9prb2il)**:** Presented [experimental results](https://www.notion.so/2306dd8db03d496a8c65828a9144d59f?pvs=21) showing the significant improvement of FedEBA+ over existing methods, including the simple exponential loss-based aggregation method (e.g., softmax-based aggregation). Highlighted our theoretical contributions compared to existing aggregation methods through a [comparison table](https://www.notion.so/2306dd8db03d496a8c65828a9144d59f?pvs=21), where we are the first to provide both general fairness and convergence analysis, addressing Reviewer 591F‘s concerns.

We sincerely appreciate Reviewer acXk for raising their score to 6 and thank Reviewer Tsda and Reviewer 8Wxv for maintaining their positive scores of 6 and 8, respectively.

Thank you for your thoughtful feedback and consideration.

Sincerely,

Authors

---

### Meta-Review · Area_Chair_J64U · 2024-12-15

**Metareview:**

The paper proposes FedEPA+, an algorithm designed to improve fairness in federated learning (FL) while maintaining global model accuracy by leveraging entropy-based aggregation and a bi-level optimization approach.

While the idea is novel, the submission suffers from significant weaknesses in theoretical justification, empirical robustness, and, most notably, clarity.

Key aspects, such as the theoretical motivation for modifying local optimization in the fair alignment case and the necessity of additional hyperparameters, remain insufficiently justified. The clarity of writing also falls short, as highlighted by reviewers, with convoluted and unclear explanations for core concepts like aggregation probabilities, even after revisions.

Overall, the paper lacks the rigor and clarity needed to convincingly demonstrate the practicality and significance of FedEPA+, and I recommend rejection.

**Additional Comments On Reviewer Discussion:**

Overall reviewers provided mixed reviews with no clear consensus.

---

### Decision · Program_Chairs · 2025-01-22

Reject